# Can RLHF be More Efficient with Imperfect Reward Models?
# A Policy Coverage Perspective

**Jiawei Huang** [1]   **Bingcong Li** [1]   **Christoph Dann** [2]   **Niao He** [1]

## Abstract

Sample efficiency is critical for online Reinforcement Learning from Human Feedback (RLHF). While existing works investigate sample-efficient online exploration strategies, the potential of utilizing misspecified yet relevant reward models to accelerate learning remains underexplored. This paper studies how to transfer knowledge from those imperfect reward models in online RLHF. We start by identifying a novel property due to KL-regularization in the RLHF objective: *a policy's coverability of the optimal policy is captured by its sub-optimality*. Building on this insight, we propose novel transfer learning principles and a theoretical algorithm—***Transfer Policy Optimization (TPO)***—with provable benefits compared to standard online learning. Empirically, inspired by our theoretical findings, we develop a win-rate-based transfer policy selection strategy with improved computational efficiency. Moreover, our empirical transfer learning technique is modular and can be integrated with various policy optimization methods, such as DPO, IPO and XPO, to further enhance their performance. We validate the effectiveness of our method through experiments on summarization tasks.

## 1. Introduction

Reinforcement Learning from Human Feedback (RLHF) has achieved remarkable success in fine-tuning Large-Language Models (LLMs) to align with human preferences (Bai et al., 2022; Christiano et al., 2017; Ouyang et al., 2022). Using datasets annotated with human preferences reflecting human intrinsic reward model, RLHF optimizes LLM policies with reinforcement learning (RL) techniques. Due to the high cost of collecting large amounts of human

preference labels, there has been significant attention in reducing the *sample complexity*—the amount of data required for training—of online RLHF through efficient exploration strategies (Cen et al., 2024; Wang et al., 2023; Xie et al., 2024; Zhang et al., 2024). However, a largely overlooked opportunity is to additionally leverage existing reward models for annotation, which have already aligned partially with the human preferences. The growing number of available open-source and high-quality reward models trained on diverse tasks provide a rich pool of candidates for transfer learning. Harnessing guidance embedded in such reward models holds great potential for improving sample efficiency.

There are a variety of practical scenarios where such source reward models can be effectively utilized. Firstly, reward models trained on relevant tasks often prove to be valuable in similar tasks. A notable example is cross-lingual reward transfer (Hong et al., 2024; Wu et al., 2024), where reward models in one language can provide effective guidance for tasks in another. Secondly, informative evaluation can also be obtained from well-trained LLMs, such as GPT, LLaMA and Gemini (Achiam et al., 2023; Dubey et al., 2024; Team et al., 2024). For certain tasks, such models can provide evaluation closely aligned with human preferences; see e.g., Ji et al. (2023); Lee et al. (2023). Lastly, there are scenarios where rule-based or heuristic reward functions—built upon experts knowledge and accumulated experience—are inexpensive to obtain and instructive in evaluating the LLM. Taking summarization tasks as an example, expert summaries are available on datasets such as XSum (Narayan et al., 2018) and TL-DR (Völske et al., 2017). Similarity with those expert solutions can be measured through metrics such as ROUGE (Lin, 2004) and BERTScore (Zhang et al., 2019), and be employed for scoring the LLM generations.

Motivated by these considerations, this paper studies how to leverage imperfect reward models to learn a near-optimal policy with fewer human annotations. We consider the case where several source reward models are available, yet their quality, i.e., the similarity to human rewards, is *unknown a priori*. Our contributions are summarized as follows.

- In Sec. 3, we identify a distinctive property of RLHF arising from its KL regularization: *for any prospective policy*

---

[1]ETH Zurich [2]Google Research. Correspondence to: Jiawei Huang <jiawei.huang@inf.ethz.ch>.

*Proceedings of the 42$^{nd}$ International Conference on Machine Learning*, Vancouver, Canada. PMLR 267, 2025. Copyright 2025 by the author(s).

*candidates, its coverability for the optimal policy improves as its policy value increases.* Enlightened by this, we propose two new principles for transfer learning in the context of RLHF (illustrated in Fig. 1).

Firstly, policy value serves as a crucial criterion to select the transfer policy, since exploiting policies with high values does not conflict with exploration. Secondly, combining insights from offline RL theory, we prove that the policy distilled by offline learning techniques from the online data generated by any no-regret online algorithm, converges to the optimal one at a rate of $\widetilde{O}(T^{-\frac{1}{2}})$ after a finite time. Such a bound improves existing sample complexity results and regret bounds in online RLHF by eliminating the dependencies on the size of the state and action space, or the complexity of the policy class (up to log-covering number). This suggests the offline policy computed with the data from the online learning process is a promising candidate to transfer from, which we term as *"self-transfer learning"*.

- In Sec. 4, following above principles, we design a transfer learning algorithm named **T**ransfer **P**olicy **O**ptimization (TPO; Alg. 1) with provable benefits. At the core of TPO is the self-transfer learning and an adaptive policy selection strategy that picks transfer policies based on estimated value gaps. In the early stage, the cumulative online regret of TPO can be significantly reduced, as long as one of the source rewards is of high quality. After finite time, the self-transfer mechanism ensures the regret of TPO grows only at a rate of $\widetilde{O}(\sqrt{T})$, independent of the standard structural complexity measures. Compared with transfer learning in the pure-reward maximization RL, our result is novel in that it exploits the policy coverage property induced by the regularization term in RLHF.

- To reduce computational overheads and improve scalablity to practical RLHF scenarios, in Sec. 5, we propose an empirical version of TPO that selects transfer policy based on the win rates competing with the learning policy, rather than relying on the estimated policy values. On the one hand, win rates can be estimated much more efficiently than the policy values. On the other hand, as justified by our theory, win rates help to identify *lower bounds* for the coverability of the optimal policy. Notably, our empirical TPO is general: its core transfer learning technique can be modularized and combined with various reward-model-free policy optimization methods, such as DPO (Rafailov et al., 2024), IPO (Azar et al., 2024), XPO (Xie et al., 2024), to boost their performance. The effectiveness of our approach is demonstrated in fine-tuning T5 models on summarization tasks in Sec. 6.

## 1.1. Related Work

We summarize the most related ones on sample complexity and transfer RL here and defer the others to Appx. B.2.

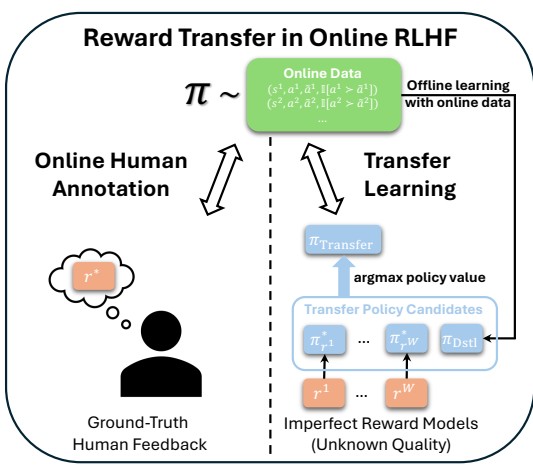

*Figure 1.* The standard online RLHF pipeline only involves learning from online human feedback (left). Our setting *additionally* leverages available imperfect reward models via transfer learning (right). Inspired by the structure induced by KL regularization, we propose novel principles for transfer learning in online RLHF: (1) selecting transfer policy $\pi_{\text{Transfer}}$ with the highest policy value; (2) self-transfer learning—involving as a candidate the policy $\pi_{\text{Dstl}}$ *distilled* from online collected data by offline learning techniques.

**Sample Complexity in RLHF**  Online RLHF emphasizes strategic exploration for sample-efficient learning in tabular and linear settings (Du et al., 2024; Novoseller et al., 2020; Pacchiano et al., 2021; Xu et al., 2020), as well as more general function approximation cases (Cen et al., 2024; Chen et al., 2022; Wang et al., 2023; Xie et al., 2024; Xiong et al., 2024; Ye et al., 2024; Zhang et al., 2024). Our work further improves sample efficiency by leveraging imperfect reward models that are readily available in a variety of practical scenarios. As an alternative, offline RLHF (Huang et al., 2024; Liu et al., 2024; Zhan et al., 2023) focuses on exploiting pre-collected datasets without exploration. What lies in between online/offline RL is hybrid RL (Chang et al., 2024; Gao et al., 2024). These methods harness online feedback, while assuming the reference policy provides good coverage and only engaging in passive exploration.

**Transfer Learning in RL and RLHF**  Transfer learning in pure-reward maximization RL has been extensively investigated in previous literature (Taylor & Stone, 2009; Zhu et al., 2023), and theoretical guarantees have been established under various conditions (Golowich & Moitra, 2022; Huang & He, 2023; Huang et al., 2022; Mann & Choe, 2013). Unlike previous works, this paper unveils new insights for transfer learning enabled by the KL regularization in RLHF. In particular, it enables us to design a policy-value-based transfer policy selection strategy, and identify a unique regime, i.e., "self-transfer learning", that can significantly improve sample efficiency. We defer more discussions to Sec. 3.2.

Most works on transfer learning in RLHF focus on empirical

approaches. For example, (Hong et al., 2024; Wu et al., 2024) investigate the cross-lingual reward transfer. To our knowledge, our empirical algorithm (Alg. 3) is novel in that it studies active transfer policy selection, which is still underexplored in existing literature. We further distinguish our work from RLAIF (Ji et al., 2023; Lee et al., 2023) or reward model selection literature (Nguyen et al., 2024). Their goal is to align LLM policies with surrogate reward models, while we study how to leverage those surrogates to accelerate the alignment with ground-truth human rewards.

## 2. Preliminary

In this section, we review the mathematical formulation of RLHF for LLMs and introduce our reward transfer setting.

### 2.1. Mathematical Formulation of RLHF

We adopt the contextual bandits framework (Lattimore & Szepesvári, 2020; Ouyang et al., 2022), where we treat the prompt space as the state space $\mathcal{S}$ and the response space as the action space $\mathcal{A}$. Without loss of generality, we assume that both $|\mathcal{S}|$ and $|\mathcal{A}|$ are finite. We denote $\rho \in \Delta(\mathcal{S})$ as the prompt distribution. An LLM can be modeled by a policy $\pi : \mathcal{S} \to \Delta(\mathcal{A})$, where, given a prompt $s \in \mathcal{S}$, the responses are generated from the conditional distribution $a \sim \pi(\cdot|s)$. Throughout this paper, we assume that all considered LLM policy $\pi$ have positive support over the entire state-action space, that is, $\min_{s,a} \pi(a|s) > 0$. This is ensured in practice by the softmax layer in LLM architectures.

**Reward Model and Preference Model** A reward model is a function $r : \mathcal{S} \times \mathcal{A} \to [0, R]$, where $R$ is a constant indicating the largest possible reward value. In the RLHF setting, the reward $r$ is unobservable and we can only access its induced preference model, denoted by $\mathbb{P}_r$. $\mathbb{P}_r(y|s, a, \widetilde{a})$ denotes the probability that $a$ is preferable to $\widetilde{a}$ given $s$ ($y = 1$) or not ($y = 0$), by reward model $r$. Following previous works, we consider the Bradley-Terry (BT) model (Bradley & Terry, 1952): $\mathbb{P}_r(y = 1|s, a, \widetilde{a}) = \sigma(r(s, a) - r(s, a'))$, where $\sigma(x) := 1/(1 + e^{-x})$ is the sigmoid function.

**RLHF Learning Setting** Given a reward model $r$, in the context of RLHF for LLM fine-tuning, we are interested in optimizing the following the KL-regularized objective:

$$\pi_r^* \leftarrow \arg\max_{\pi} J_\beta(\pi; r)$$

with $J_\beta(\pi; r) := \mathbb{E}_{s \sim \rho, a \sim \pi}[r(s, a)] - \beta \mathrm{KL}(\pi \| \pi_{\mathrm{ref}}),$ (1)

where we use $\pi_{\mathrm{ref}}$ to denote the pretrained reference policy and $\mathrm{KL}(\pi \| \pi_{\mathrm{ref}}) := \mathbb{E}_{s \sim \rho, a \sim \pi(\cdot|s)}[\log \frac{\pi(a|s)}{\pi_{\mathrm{ref}}(a|s)}]$. The above optimization problem yields a closed-form solution:

$$\pi_r^*(a|s) \propto \pi_{\mathrm{ref}}(a|s) e^{\frac{r(s,a)}{\beta}}. \quad (2)$$

Here $\beta > 0$ is a moderate constant and is critical for RLHF

in practice, for detailed reasons discussed in Appx. B.1.

We use $r^* : \mathcal{S} \times \mathcal{A} \to [0, R]$ to denote the unknown true reward function determining the human's preference $\mathbb{P}_{r^*}$. For simplicity, we omit $r^*$ and use $J_\beta(\pi)$ as a short note for $J_\beta(\pi; r^*)$. Following previous works (Xie et al., 2024; Zhang et al., 2024), we consider the function approximation setting with access to a policy candidates class $\Pi$ ($|\Pi| < +\infty$) satisfying standard assumptions as follows:

**Assumption A.** The policy class $\Pi$ satisfies: **(I) Realizability**: The optimal policy $\pi_{r^*}^* \in \Pi$. **(II) Bounded Policy Ratio**: For any $\pi \in \Pi$, $\max_{s,a} |\log \frac{\pi(a|s)}{\pi_{\mathrm{ref}}(a|s)}| \leq \frac{R}{\beta}$.

Note that $\max_{s,a} |\log \frac{\pi_r^*(a|s)}{\pi_{\mathrm{ref}}(a|s)}| \leq \frac{R}{\beta}$ holds as long as $r \in [0, R]$ (see Lem. B.1 in Appx. B.1). Thus, Assump. A-(II) is *not* actually an assumption, but can be interpreted as an additional filtering step by leveraging the boundary of $r^*$.

**Additional Notation** We use the standard big-O notations and use $\widetilde{(\cdot)}$ (e.g. $\widetilde{O}$) to suppress logarithmic terms, ***including log-covering numbers***. Given $\Pi$ satisfying Assump. A, $\mathrm{conv}(\Pi)$ denotes its convex hull, and $\mathcal{R}^\Pi$ denotes the reward class induced by $\Pi$ via Eq. (2), s.t., (1) $\forall r \in \mathcal{R}^\Pi$, $r \in [0, R]$; (2) $\exists r \in \mathcal{R}^\Pi$, $\pi_r^* = \pi_{r^*}^*$. We defer to Appx. B.1 for detailed converting process. Besides, we denote $[n] := \{1, 2, ..., n\}$ and $a \wedge b := \min\{a, b\}$. We refer the reader to Appx. A for a table of commonly used notation in this paper.

### 2.2. Reward Transfer Setting

We assume there are $W$ source reward models available, denoted by $\{r^w\}_{w=1}^W$, s.t. $\forall w, s, a, r^w(s, a) \in [0, R]$. As motivated in Sec. 1, those reward models are accessible in many scenes. Given a source reward $r^w$, let $\pi_{r^w}^*$ be the corresponding source policy and denote $\Delta(w) := J_\beta(\pi_{r^*}^*) - J_\beta(\pi_{r^w}^*)$ as its policy value gap. We define $\Delta_{\min} := \min_{w \in [W]} \Delta(w)$ to be the minimal gap for all source policies. Note that $\Delta_{\min} \geq 0$ and $\Delta_{\min} = 0$ implies $r^* \in \{r^w\}_{w \in [W]}$. We ***do not assume prior knowledge on*** $\{\Delta(w)\}_{w \in [W]}$.

**LLM Policy as Reward Model** Eq. (2) implies that there is a way to convert a given LLM policy $\pi$ to a reward model. Concretely, by choosing an arbitrary distribution $\pi_0$, we can interpret $\beta \log \frac{\pi(\cdot|\cdot)}{\pi_0(\cdot|\cdot)}$ as a reward function that $\pi$ aligns with given $\pi_0$ as the reference policy (Rosset et al., 2024). Therefore, while we consistently use the term "reward transfer" throughout the paper for clarity, our framework is general to handle transfer learning from any LLM policy through the underlying reward function it aligns with.

### 2.3. Background on Policy Coverability

We first introduce the formal definition of the policy coverage coefficient, which measures how well a given policy distribution covers the other.

**Definition 2.2.** Given any $\pi, \widetilde{\pi}$, the coverage coefficient for $\widetilde{\pi}$ by $\pi$ is defined by $\mathrm{Cov}^{\widetilde{\pi}|\pi} := \mathbb{E}_{s \sim \rho, a \sim \widetilde{\pi}(\cdot|s)}[\frac{\widetilde{\pi}(a|s)}{\pi(a|s)}]$.

The concept of policy coverage originally emerged from offline RL (Chen & Jiang, 2019; Yang et al., 2020; Zhan et al., 2022), where one aims to find a good policy given a fixed dataset. The coverage of the optimal policy by the data-generating policy naturally governs the size of the dataset required to find the optimal policy. Policy coverage was later extended to online RL, inducing novel complexity measures to characterize intrinsic statistical difficulty of the underlying MDP (Amortila et al., 2024; Xie et al., 2022).

Compared to alternative complexity measures, policy coverage is particularly suited for studying sample complexity in RLHF, since optimizing or exploring directly on the policy (LLM) space is preferred for its computational tractability. We use coverage as an analytical tool for reward transfer, and our results are based on RPO (Liu et al., 2024) and XPO (Xie et al., 2024) for offline and online RLHF, respectively:

**Lemma 2.3** (Offline RLHF; Thm. 5.3 in (Liu et al., 2024); Informal). *Given a dataset $\mathcal{D}$ generated by a policy $\pi^{\mathcal{D}}$, running RPO with any $\mathcal{R}$ including $r^*$ and $\widehat{\Pi}$ yields $\widehat{\pi}$, s.t., $\forall \pi \in \widehat{\Pi}, J_\beta(\pi) - J_\beta(\widehat{\pi}) = \widetilde{O}(e^{2R} \mathrm{Cov}^{\pi|\pi^{\mathcal{D}}} |\mathcal{D}|^{-\frac{1}{2}})$.*

**Lemma 2.4** (Online RLHF; Thm. 3.1 in (Xie et al., 2024); Informal). *Running XPO with $\Pi$ satisfying Assump. A for $T$ steps yields a policy sequence $\{\pi^t\}_{t=1}^T$, s.t. $\sum_{t=1}^T J_\beta(\pi_{r^*}^*) - J_\beta(\pi^t) = \widetilde{O}(R \cdot e^{2R} \cdot \sqrt{\mathrm{Cov}_\infty(\Pi)T})$.*

Lem. 2.3 states that it is possible to compute an offline policy competitive with any policy well-covered by the dataset. Lem. 2.4 suggests that the sample efficiency of online RLHF can be characterized by the $L_\infty$ coverability $\mathrm{Cov}_\infty(\Pi)$. Here, $\mathrm{Cov}_\infty(\Pi)$ by Xie et al. (2022) is a worst-case version of our Def. 2.2 as it takes the maximum over $s, a$ and $\widetilde{\pi}$. See Appx. D for the formal definition.

# 3. The Blessing of Regularization: A Policy Coverage Perspective

Unlike classical pure-reward maximization RL, the RLHF objective in (1) incorporates regularization with respect to $\pi_{\mathrm{ref}}$. We start by identifying distinctive properties associated with such regularization in Sec. 3.1, and discuss their implications on transfer learning in RLHF in Sec. 3.2.

## 3.1. Structural Property Induced by Regularization

**Lemma 3.1.** *Under Assump. A and assume $r^w \in [0, R]$ for all $w \in [W]$, then, for any policy $\pi \in conv(\Pi) \cup \{\pi_{r^w}^*\}_{w=1}^W$,*

$$\mathrm{Cov}^{\pi_{r^*}^*|\pi} \leq 1 + \kappa(e^{\frac{2R}{\beta}}) \cdot \frac{J_\beta(\pi_{r^*}^*) - J_\beta(\pi)}{\beta}, \quad (3)$$

*where $\kappa(x) := \frac{(x-1)^2}{x-1-\log x} = O(x)$.*

The key insight of Lem. 3.1 is that: for any prospective candidates of the optimal policy (i.e., $conv(\Pi)$[1]) or any transfer candidates (i.e., $\{\pi_{r^w}^*\}_{w=1}^W$), *its coverability of $\pi_{r^*}^*$ is controlled by its policy value gap*. Intuitively, $\mathrm{Cov}^{\widetilde{\pi}|\pi}$ becomes extremely large or even unbounded if there is a significant distribution shift between $\pi$ and $\widetilde{\pi}$. However, in the presence of regularization ($\beta > 0$), we should only consider policies with bounded policy ratio relative to $\pi_{\mathrm{ref}}$ (see Assump. A-(II)), and exclude those (near-)deterministic ones from our policy candidate class $\Pi$, because none of them can be (near-)optimal. In other words, regularization leverages prior knowledge from $\pi_{\mathrm{ref}}$ and enables a free policy filtration step before learning begins, ensuring that the remaining policies exhibit a favorable structure (Lem. 3.1).

To understand why such property is uniquely arising from regularization, consider a bandit instance with a single optimal arm and multiple suboptimal arms yields rewards $R$ and $R - 2\varepsilon$, respectively. In pure reward maximization RL ($\beta = 0$), the optimal policy $\pi_{r^*}^*$ is deterministic. A policy class $\Pi$ satisfying Assump. A may include several suboptimal deterministic policies. The coverage coefficient between any of them and $\pi_{r^*}^*$ is infinity, while their suboptimal gaps are $2\varepsilon$ and can be arbitrarily small.

## 3.2. New Insights for Transfer Learning in RLHF

In the online RLHF, the primary goal of exploration is to discover high-reward regions, i.e., the states and actions covered by the optimal policy. Therefore, in our reward transfer setup, we propose to ***transfer from the policy with the best coverage of*** $\pi_{r^*}^*$. Inspired by Lem. 3.1, we identify two novel principles for transfer learning for RLHF objective, which we will further explore in later sections.

**Principle 1: Select Transfer Policies with High Policy Value** By Lem. 3.1, exploiting a policy with high value for data collection could "help" exploration, because such a policy inherently provides good coverage for $\pi_{r^*}^*$. In other words, regularization reconciles the trade-off between exploration and exploitation. This insight allows us to use policy value as a criterion and transfer from the policy achieving the highest value among all candidates. This strategy is also practical given that policy values can be estimated well.

We emphasize that this principle is unique in the regularized setting. As exemplified by the bandit instance before, near-optimality does not imply good coverage for $\pi_{r^*}^*$ in the absence of regularization. To avoid negative transfer in pure reward maximization setting, previous algorithms typically rely on additional assumptions about task similarity and employ sophisticated strategies to balance exploiting good source tasks with exploration (Golowich & Moitra, 2022;

---

[1] Here we consider the convex hull in order to incorporate all possible uniform mixture policies induced by $\Pi$.

Huang & He, 2023), which can be challenging to generalize beyond the tabular setting. In contrast, regularization enables us to filter transfer policies directly with their policy value, facilitating the applicablity beyond the tabular setup.

**Principle 2: Transfer from the Policy Distilled from Online Data—the "Self-Transfer Learning"** We first introduce a key result by combining Lem. 3.1 and offline RLHF result in Lem. 2.3.

**Theorem 3.2.** *Under Assump. A, w.p. $1-\delta$, given an online dataset $\mathcal{D}$ generated[2] by a policy series $\{\pi^t\}_{t=1}^T \in conv(\Pi)$, running RPO with $conv(\Pi)$ and $\mathcal{R}^\Pi$ on $\mathcal{D}$ yields a distilled policy $\pi_{Dstl}$, such that,*

$$J_\beta(\pi_{r*}^*) - J_\beta(\pi_{Dstl}) \leq \qquad (4)$$

$$\widetilde{O}\Big(e^{2R}\Big(1 + \kappa(e^{\frac{2R}{\beta}})\sum_{t=1}^T \frac{J_\beta(\pi_{r*}^*) - J_\beta(\pi^t)}{\beta T}\Big)\sqrt{\frac{1}{T}}\Big).$$

To understand the significance of Thm. 3.2, consider the case when $\{\pi^t\}_{t\in[T]}$ are produced by a no-regret online learning algorithm, such as XPO in Lem. 2.4. As a result, the term $\sum_{t=1}^T \frac{J_\beta(\pi_{r*}^*)-J_\beta(\pi^t)}{\beta T}$ in Eq. (4) diminishes to 0 as $T$ increases. This implies that the policy $\pi_{Dstl}$ distilled from online data by offline learning techniques converges to $\pi_{r*}^*$ at a rate of $O(T^{-\frac{1}{2}})$, which **does not depend on** $|\mathcal{S}|, |\mathcal{A}|$ or other complexity measures such as $\text{Cov}_\infty(\Pi)$ in Lem. 2.4. This result not only strictly improves the sample complexity bounds[3] for existing online RLHF algorithms (Cen et al., 2024; Xie et al., 2024; Xiong et al., 2024; Zhang et al., 2024), but also reveals a fundamental difference from the pure reward maximization setting, where lower bounds depending on those structural complexity factors have been established (Auer et al., 2002; Dani et al., 2008). We defer detailed discussions to Appx. E.3.

More importantly, the faster rate of the convergence of $\pi_{Dstl}$ to $\pi_{r*}^*$ also indicates the potential of using $\pi_{Dstl}$ as a candidate for policy transfer. We term this regime as "*self-transfer learning*", and refer $\pi_{Dstl}$ as the "*self-transfer policy*". Notably, $\pi_{Dstl}$ continuously improves and converges to $\pi_{r*}^*$ as the dataset grows, while the source policies $\{\pi_{r^w}^*\}_{w=1}^W$ retain fixed non-zero value gaps due to the imperfections in reward models $\{r^w\}_{w=1}^W$. This reveals another benefit of self-transfer learning: it helps to avoid being restricted by suboptimal source reward models.

# 4. Provably Efficient Transfer Learning

In this section, we develop provably efficient transfer learning algorithms based on the principles in Sec. 3.2.

---

[2]See Cond. C.1 for the definition of data generation process.

[3]Beyond sample complexity, a regret bound improved to $\widetilde{O}(\sqrt{T})$ for online RLHF can be established. We defer it to Coro. 4.5 after presenting our main results.

**Outline of Main Algorithm** Our main algorithm TPO is provided in Alg. 1, which leverages Alg. 2—**T**ransfer **P**olicy **S**election (**TPS**)—as a subroutine to select source policies to transfer from. TPO can be regarded as a mixture of standard online learning and transfer learning, balanced through a hyper-parameter $\alpha \in (0,1)$. Motivated by the implication of Thm. 3.2, TPO returns the detailed policy computed with all the data collected. For convenience, we divide the total number of iterations $T$ into $K = T/N$ blocks, each containing $N$ sub-iterations. In each block, we first run $\alpha N$ iterations of an **OnL**ine learning algorithm $\text{Alg}_{\text{OL}}$, followed by $(1-\alpha)N$ iterations of transfer learning with policy selected by Alg. 2. Here, $\text{Alg}_{\text{OL}}$ can be any online algorithm with per-step no-regret guarantees, for example, XPO in Lem. 2.4. To save space, we defer to Appx. D the formal behavior assumption on $\text{Alg}_{\text{OL}}$ (Def. D.2) and concrete examples with verifications.

**A Preview of Main Theorem** Before diving into the details, we first highlight the benefits of transfer learning by presenting an informal corollary regarding the regret bound of TPO, under concrete choices of $\text{Alg}_{\text{OL}}$ and $\alpha$.

**Corollary 4.1.** *Choosing XPO (Xie et al., 2024) as $\text{Alg}_{\text{OL}}$ and $\alpha = e^{-\frac{R}{\beta}}$, TPO achieves $\widetilde{O}(W\sqrt{T})$ regret when $T$ is small and $\widetilde{O}(\sqrt{T})$ regret after $T$ is large enough.*

Coro. 4.1 is implied by our main result in Thm. 4.4. We refer to Remark F.3 for a detailed quantification of "small" and "large enough". Compared with previous online RLHF results without transfer learning, in the early stage, TPO improves the structural complexity measure coefficients (e.g. $\text{Cov}_\infty$ in XPO) to the number of source tasks $W$, which is usually much smaller. Besides, it even gets rid of such coefficient term and achieves $\widetilde{O}(\sqrt{T})$ regret over time.

Next, we take a closer look at the transfer policy selection steps in Alg. 2 in Sec. 4.1, and provide detailed analyses and discussion of TPO in Sec. 4.2.

## 4.1. Details for Alg. 2: The Transfer Policy Selection

The design of Alg. 2 follows the two principles in Sec. 3.2, which are: (1) transfer the policy with the highest (estimated) policy value, because higher policy value implies better coverage for $\pi_{r*}^*$; (2) include the self-transfer policy as a candidate, because it progressively converges to $\pi_{r*}^*$ at a faster rate than the best-known ones for online policies.

We first clarify some notation. Given a dataset $\mathcal{D} := \{(s^i, a^i, \widetilde{a}^i, y^i, \pi^i)\}_{i=1}^{|\mathcal{D}|}$, $L_\mathcal{D}(r)$ denotes the average negative log-likelihood (NLL) loss regarding the reward $r$:

$$L_\mathcal{D}(r) := \frac{1}{|\mathcal{D}|}\sum_{i \leq |\mathcal{D}|} -y^i \log\sigma\Big(r(s^i,a^i) - r(s^i,\widetilde{a}^i)\Big)$$

$$- (1-y^i)\log\sigma\Big(r(s^i,\widetilde{a}^i) - r(s^i,a^i)\Big). \quad (5)$$

---

**Algorithm 1: T**ransfer **P**olicy **O**ptimization (TPO)

---

1 **Input**: Block size $N$; Number of blocks $K = T/N$; $\{r^w\}_{w=1}^W$; $\Pi$; $\alpha \in (0,1)$; $\delta \in (0,1)$

2 For all $(k,n)$, $\mathcal{D}^{k,n}$ denotes all the data collected up to $(k,n)$, and $\mathcal{D}_{\text{OL}}^{k,n}$ only includes those collected by $\texttt{Alg}_{\text{OL}}$. See detailed definitions in Appx. F.1.

3 **for** $k = 1, 2..., K$ **do**

4     **for** $n = 1, ..., N$ **do**

5         **if** $n \leq \alpha N$ **then** $\pi^{k,n} \leftarrow \texttt{Alg}_{\text{OL}}(\alpha T, \Pi, \delta; \mathcal{D}_{\text{OL}}^{k,n})$ **else** $\pi^{k,n} \leftarrow \text{TPS}(T, \Pi, \delta, \{r^w\}_{w=1}^W; \mathcal{D}^{k,n})$

6         Collect data $(s^{k,n}, a^{k,n}, \widetilde{a}^{k,n}, y^{k,n}) \sim \rho \times \pi^{k,n} \times \pi_{\text{ref}} \times \mathbb{P}_{r^*}(\cdot|\cdot, \cdot, \cdot)$.

7     **end**

8 **end**

9 **return** $\widehat{\pi}_{r^*}^*$ computed by RPO with $\mathcal{D}^{K,N+1}$.

---

We will use $\mathbb{E}_{\rho,\pi}[r] := \mathbb{E}_{s\sim\rho, a\sim\pi}[r(s,a)]$ as a short note. In Line 4 of Alg. 2, $N(w; \mathcal{D}) := \sum_{i\leq|\mathcal{D}|} \mathbb{I}[\pi^i = \pi_{r^w}^*]$ denotes the number samples collected with $\pi_{r^w}^*$ in the dataset, following the convention that $1/N(\cdot, \cdot) = +\infty$ if $N(\cdot, \cdot) = 0$. In Line 6, we leverage RPO (Liu et al., 2024) to compute the self-transfer policy $\pi_{\text{Dstl}}$ and a reward function $\widehat{r}_{\text{Dstl}}$. To save space, we defer the details of RPO to Appx. C.

Next, we explain our value estimation strategy. Note that in RLHF setting, we cannot access $r^*$ directly but only the preference comparison samples following the BT model. Thus, we instead estimate the value gain relative to $J_\beta(\pi_{\text{ref}})$.

**Optimistic Estimation for** $J_\beta(\pi_{r^w}^*) - J_\beta(\pi_{\text{ref}})$ For policies induced by imperfect source reward models, we adopt UCB-style optimistic policy evaluation to efficiently balance exploration and exploitation. Intuitively, by utilizing the MLE reward estimator $\widehat{r}_{\text{MLE}}$, the estimation error $\widehat{r}_{\text{MLE}} - r^*$ under the distribution of $\pi_{r^w}^*$ is related to the number of samples from $\pi_{r^w}^*$ occurring in the dataset. Therefore, we can quantify the value estimation error as follows.

**Lemma 4.2** (Value Est Error for $\{\pi_{r^w}^*\}_{w\in[W]}$)**.** *Under Assump. A and Def. D.2, w.p. $1 - \delta$, in each call of Alg. 2:*

$$\forall w \in [W], \quad J_\beta(\pi_{r^w}^*) - J_\beta(\pi_{\text{ref}}) \leq \widehat{V}(\pi_{r^w}^*; \mathcal{D})$$
$$\leq J_\beta(\pi_{r^w}^*) - J_\beta(\pi_{\text{ref}}) + \widetilde{O}\Big(\frac{e^{2R}}{\sqrt{N(w; \mathcal{D})}}\Big).$$

**Pessimistic Estimation for** $J_\beta(\pi_{\text{Dstl}}) - J_\beta(\pi_{\text{ref}})$ The main challenge in estimating the value of $\pi_{\text{Dstl}}$ is that, $\pi_{\text{Dstl}}$ is not fixed but changing and improving. The previous optimistic strategy is not applicable here, since the coverage of $\pi_{\text{Dstl}}$ by the dataset is unclear, making it difficult to quantify the uncertainty in estimation via count-based bonus term. Fortunately, given that $\pi_{\text{Dstl}}$ is improving over time, it is more important when it surpasses all the other source policies. Therefore, it suffices to construct a tight lower bound for $J_\beta(\pi_{\text{Dstl}}) - J_\beta(\pi_{\text{ref}})$; see line 7. By leveraging $\widehat{r}_{\text{MLE}}$ and the optimality of $(\pi_{\text{Dstl}}, \widehat{r}_{\text{Dstl}})$ for the RPO loss, we can show:

**Lemma 4.3** (Value Est Error for $\pi_{\text{Dstl}}$)**.** *Under Assump. A*

*and Def. D.2, w.p. $1 - \delta$, in each call of Alg. 2:*

$$J_\beta(\pi_{r^*}^*) - J_\beta(\pi_{ref}) - \widetilde{O}\Big(\frac{Re^{2R}}{\sqrt{|\mathcal{D}|}} \cdot \big(\text{Cov}^{\pi_{r^*}^*|\pi_{\text{mix}}^{\mathcal{D}}} \wedge \frac{\sqrt{\mathcal{C}(\Pi)}}{\alpha}\big)\Big)$$

$$\leq \widehat{V}(\pi_{Dstl}; \mathcal{D}) \leq J_\beta(\pi_{Dstl}) - J_\beta(\pi_{ref}). \tag{6}$$

Here $\pi_{\text{mix}}^{\mathcal{D}} := \frac{1}{|\mathcal{D}|} \sum_{i\leq|\mathcal{D}|} \pi^i$ denotes the mixture policy. In the LHS, the coefficient takes minimum over two factors $\text{Cov}^{\pi_{r^*}^*|\pi_{\text{mix}}^{\mathcal{D}}}$ and $\sqrt{\mathcal{C}(\Pi)}/\alpha$, resulting from two different ways to estimate the value gap of $\pi_{\text{Dstl}}$. According to offline RLHF theory (see Lem. C.2), $\pi_{\text{Dstl}}$ is competitive with any $\pi \in \text{conv}(\Pi)$ well-covered by the dataset distribution, or equivalently, $J_\beta(\pi_{r^*}^*) - J_\beta(\pi_{\text{Dstl}}) = J_\beta(\pi_{r^*}^*) - J_\beta(\pi) + \widetilde{O}(|\mathcal{D}|^{-\frac{1}{2}} \text{Cov}^{\pi|\pi_{\text{mix}}^{\mathcal{D}}})$. By choosing $\pi = \pi_{r^*}^*$, we obtain the first bound with the factor $\text{Cov}^{\pi_{r^*}^*|\pi_{\text{mix}}^{D}}$. Next, considering $\pi = \frac{1}{\alpha k N} \sum_{i\leq k, j\leq \alpha N} \pi^{i,j}$, the uniform mixture of policies generated by $\texttt{Alg}_{\text{OL}}$ so far, leads to the second bound involving $\sqrt{\mathcal{C}(\Pi)}/\alpha$. This also explains why we still involve normal online learning in TPO—to provide another safeguard for the quality of the transfer policy.

### 4.2. Main Theorem and Interpretation

We establish the per-step regret bound for TPO below.

**Theorem 4.4** (Total Regret)**.** *Suppose $\texttt{Alg}_{\text{OL}}$ is a no-regret instance satisfying Def. D.2, whose regret grows as $\widetilde{O}(Re^{2R}\sqrt{\mathcal{C}(\Pi)\widetilde{T}})$ for any intermediate step $\widetilde{T}$ and some policy class complexity measure $\mathcal{C}(\Pi)$. Then, w.p. $1 - 2\delta$, for any $T/K \leq t \leq T$, running TPO yields a regret bound:*

$$\sum_{\tau \leq t} J_\beta(\pi_{r^*}^*) - J_\beta(\pi^{k(\tau), n(\tau)}) = Reg_{OL}^{(t)} + Reg_{Trf}^{(t)}$$

$$Reg_{OL}^{(t)} := \widetilde{O}(Re^{2R}\sqrt{\alpha\mathcal{C}(\Pi)t}), \tag{7}$$

$$Reg_{Trf}^{(t)} := \widetilde{O}\Big( \sum_{\tau \leq t: \alpha N < n(\tau) \leq N} \Delta_{\min} \wedge \iota^{k(\tau), n(\tau)} \tag{8}$$

$$+ e^{2R}\sqrt{(1-\alpha)Wt} \wedge \sum_{w: \Delta(w) > 0} \frac{e^{4R}}{\Delta(w)}\Big).$$

---

**Algorithm 2: T**ransfer **P**olicy **S**election (TPS)

1 **Input**: Source tasks $\{r^w\}_{w=1}^W$; Policy class $\Pi$; $T, \delta$; Dataset $\mathcal{D} := \{(s^i, a^i, \widetilde{a}^i, y^i, \pi^i)\}_{i \leq |\mathcal{D}|}$.

2 // Optimistic estimation for $J_\beta(\pi_{r^w}^*) - J_\beta(\pi_{\text{ref}})$.

3 $\widehat{r}_{\text{MLE}} \leftarrow \arg\min_{r \in \mathcal{R}^\Pi} L_{\mathcal{D}}(r)$.

4 $\forall w \in [W]$, $\widehat{V}(\pi_{r^w}^*; \mathcal{D}) \leftarrow \mathbb{E}_{\rho, \pi_{r^w}^*}[\widehat{r}_{\text{MLE}}] - \mathbb{E}_{\rho, \pi_{\text{ref}}}[\widehat{r}_{\text{MLE}}] - \beta \text{KL}(\pi_{r^w}^* \| \pi_{\text{ref}}) + 16e^{2R}\sqrt{\frac{1}{N(w; \mathcal{D})} \log \frac{|\Pi|WT}{\delta}}$.

5 // Pessimistic estimation for $J_\beta(\pi_{\text{Dstl}}) - J_\beta(\pi_{\text{ref}})$.

6 $\pi_{\text{Dstl}}, \widehat{r}_{\text{Dstl}} \leftarrow \text{RPO}(\text{conv}(\Pi), \mathcal{R}^\Pi, \mathcal{D}, \eta)$ with $\eta = c \cdot (1 + e^R)^{-2}\sqrt{\frac{1}{|\mathcal{D}|} \log \frac{|\Pi|T}{\delta}}$.

7 $\widehat{V}(\pi_{\text{Dstl}}; \mathcal{D}) \leftarrow \mathbb{E}_{\rho, \pi_{\text{Dstl}}}[\widehat{r}_{\text{Dstl}}] - \mathbb{E}_{\rho, \pi_{\text{ref}}}[\widehat{r}_{\text{Dstl}}] - \beta \text{KL}(\pi_{\text{Dstl}} \| \pi_{\text{ref}}) + \frac{1}{\eta}L_{\mathcal{D}}(\widehat{r}_{\text{Dstl}}) - \frac{1}{\eta}L_{\mathcal{D}}(\widehat{r}_{\text{MLE}}) - 2ce^{2R}\sqrt{\frac{1}{|\mathcal{D}|} \log \frac{|\Pi|T}{\delta}}$.

8 **return** $\arg\max_{\pi \in \{\pi_{r^w}^*\}_{w \in [W]} \cup \{\pi_{\text{Dstl}}\}} \widehat{V}(\pi; \mathcal{D})$. // Selecting Transfer Policy by Estimated Value

---

*Here we denote $k(\tau) := \lceil \frac{\tau}{N} \rceil$ and $n(\tau) := \tau \mod N$ to be the block index and inner iteration index for step $\tau$; $\iota^{k(\tau), n(\tau)} := \widetilde{O}(Re^{2R}(\text{Cov}^{\pi_{r^*}^* | \pi_{\text{mix}}^\tau} \wedge \frac{\sqrt{\mathcal{C}(\Pi)}}{\alpha})\sqrt{\frac{1}{\tau}})$, where $\pi_{\text{mix}}^\tau := \frac{1}{\tau}\sum_{i \leq \tau}\pi^{k(i), n(i)}$ is the mixture policy up to $\tau$; $\Delta(w)$ and $\Delta_{\min}$ denote value gaps as defined in Sec. 2.2.*

We decompose the total regret into two parts depending on their origins. $\text{Reg}_{\text{OL}}^{(t)}$ comes from the regret by running the online algorithm $\text{Alg}_{\text{OL}}$. It is weighted by $\alpha$ since we only allocate $\alpha$-proportion of the samples for $\text{Alg}_{\text{OL}}$. $\text{Reg}_{\text{Trf}}^{(t)}$ represents the regret from transfer policies. The first term in Eq. (8) reflects the benefits of utilizing transfer policy over online learning. Here $\Delta_{\min}$ is contributed by source reward models $\{r^w\}_{w \in [W]}$, and the term $\iota^{k(\tau), n(\tau)}$ is due to the "self-transfer policy" $\pi_{\text{Dstl}}$, as we derived in Lem. 4.3. The second term in Eq. (8) results from the imperfection of source reward models: without prior knowledge on their quality, additional cost has to be paid during exploration.

Next, we elaborate the benefits of transfer learning by taking a closer look at $\text{Reg}_{\text{Trf}}^{(t)}$ in Eq. (8). Note that the lower $\text{Reg}_{\text{Trf}}^{(t)}$ is, the faster $\widehat{\pi}_{r^*}^*$ in TPO converges to $\pi_{r^*}^*$. When $\Delta_{\min} = 0$, i.e. $r^*$ is realizable in $\{r^w\}_{w \in [W]}$, we have $\text{Reg}_{\text{Trf}}^{(t)} = \widetilde{O}(\sqrt{Wt} \wedge \sum_{w: \Delta(w) > 0} \frac{1}{\Delta(w)})$ and the benefit of transfer learning is clear. Thus, in the following, we only focus on the case $\Delta_{\min} > 0$. We separately consider two scenarios, according to the relationship between $t$ and $\Delta_{\min}$. For clarity, we will omit the constant terms $R$ and $e^R$.

**Stage 1:** $t < \frac{W^2}{\Delta_{\min}^2}$ This corresponds to the early learning stage, when $t$ is relatively small. In this case, Thm. 4.4 implies the following regret bound:

$$\text{Reg}_{\text{Trf}}^{(t)} = \widetilde{O}(\sqrt{1 - \alpha}(\sqrt{Wt} + \Delta_{\min}t)) = \widetilde{O}(W\sqrt{t}), \quad (9)$$

which can be further improved to $\widetilde{O}(\sqrt{Wt})$ if $t < \frac{W}{\Delta_{\min}^2}$. This suggests at the earlier stage, the benefits of transfer is contributed mostly by the source reward models $\{r^w\}_{w \in [W]}$. In general, we can expect the number of source tasks $W$

much lower than the policy class complexity measure $\mathcal{C}(\Pi)$. Therefore, Eq. (9) implies a significant improvement over the typical online learning regret bound without transfer.

**Stage 2:** $t \geq \frac{W^2}{\Delta_{\min}^2}$ In this case, the second term in Eq. (8) is controlled by $O(\sum_{w \in [W]} \frac{1}{\Delta(w)}) = O(\frac{W}{\Delta_{\min}}) = O(\sqrt{t})$, and we have the following regret bound:

$$\text{Reg}_{\text{Trf}}^{(t)} = \widetilde{O}\left(\sqrt{\frac{\mathcal{C}(\Pi)t}{\alpha^2}} \wedge \sum_{\tau \leq t}(\text{Cov}^{\pi_{r^*}^* | \pi_{\text{mix}}^\tau})^2\right). \quad (10)$$

At the first glance, the RHS is controlled by $\widetilde{O}(\sqrt{\mathcal{C}(\Pi)t}/\alpha)$, which implies transfer learning at most suffer a factor of $1/\alpha$ larger regrets than no transfer. However, in fact, the term $\sqrt{\sum_{\tau \leq t}(\text{Cov}^{\pi_{r^*}^* | \pi_{\text{mix}}^\tau})^2}$ yields a much tighter bound, which only grows as $\widetilde{O}(\sqrt{t})$ after finite time, and is independent of $\mathcal{C}(\Pi)$. To see this, by Lem. 3.1 and the concavity of $J_\beta(\cdot)$, we have $\text{Cov}^{\pi_{r^*}^* | \pi_{\text{mix}}^t} = 1 + \widetilde{O}(\kappa(e^{\frac{2R}{\beta}}) \cdot \frac{\text{Reg}_{\text{OL}}^{(t)} + \text{Reg}_{\text{Trf}}^{(t)}}{\beta t})$. Note that Eq. (9) and (10) already indicate a regret upper bound $\text{Reg}_{\text{Trf}}^{(t)} = \widetilde{O}(\Gamma\sqrt{t})$, where $\Gamma$ is a short note of a coefficient depending on $\alpha$, $W$, $\{\Delta(w)\}_{w \in [W]}$ and $\mathcal{C}(\Pi)$, but not $t$. This implies $\text{Cov}^{\pi_{r^*}^* | \pi_{\text{mix}}^t}$ converges to 1 at the rate of $O(1/\sqrt{t})$, and $\text{Reg}_{\text{Trf}}^{(t)} = \widetilde{O}(\sqrt{t})$ after finite time.

Although the above provable benefits in Stage 2 result primarily from "self-transfer learning", high-quality source reward models also play an important role here. According to Eq. (9), small $\Delta_{\min}$ can lead to small $\Gamma$ and therefore, accelerate the convergence of $\text{Cov}^{\pi_{r^*}^* | \pi_{\text{mix}}^t}$ towards 1.

**Remarks on choice of $\alpha$** We treat $\alpha$ as a hyperparameter. Without prior knowledge, a simple choice is $\alpha = e^{-\frac{R}{\beta}}$ with guarantee in Coro. 4.1. Under prior beliefs that high-quality source reward models are available, we may prefer smaller $\alpha$. Besides, due to the self-transfer learning, it is wise to gradually decay $\alpha$ to 0 as the iteration number grows.

**Improved regret bound for standard online RLHF as a implication** Although our focus is transfer learning, the standard online RLHF setting can be recovered when no

---

**Algorithm 3:** Empirical TPO

---

**1** **Input**: $K$, $N$ and $\{r^w\}_{w \in [W]}$; **Initialize**: $\pi^1_{\text{OL}} \leftarrow \pi_{\text{ref}}$;

**2** For all $(k, n) \in [K] \times [N]$, and all $w \in [W]$, $\mathcal{D}^{k,n} := \cup_{j=1}^{n-1} \{(s^{k,j}, a^{k,j}, \widetilde{a}^{k,j}, y^{k,j})\}$,

   $N^{k,n}(\cdot) := \sum_{j<n} \mathbb{I}[\cdot = \pi^{k,j}]$, and $\widehat{\mathbb{P}}^{k,n}_{r^*}(\cdot \succ \pi^k_{\text{OL}}) := \frac{1}{N^{k,n}(\cdot)} \sum_{j<n} \mathbb{I}[\cdot = \pi^{k,j}] y^{k,j}$.

**3** **for** $k = 1, 2..., K$ **do**

**4**    **for** $n = 1, ... N$ **do**

**5**      $\forall w \in [W]$, $\widehat{\text{WR}}^{\pi^*_{r^w}} \leftarrow \widehat{\mathbb{P}}^{k,n}_{r^*}(\pi^*_{r^w} \succ \pi^k_{\text{OL}}) + c\sqrt{\log \frac{1}{\delta} / N^{k,n}(\pi^*_{r^w})}$;

**6**      $\widehat{\text{WR}}^{\pi^k_{\text{OL}}} \leftarrow \mathbb{P}_{r^*}(\pi^k_{\text{OL}} \succ \pi^k_{\text{OL}}) = 0.5$. // $\widehat{\text{WR}}^{\pi^k_{\text{OL}}}$ can be treated as a hyperparameter taking value other than 0.5.

**7**      $\pi^{k,n} \leftarrow \arg\max_{\pi \in \{\pi^*_{r^w}\}_{w=1}^{W} \cup \{\pi^k_{\text{OL}}\}} \widehat{\text{WR}}^{\pi}$.

**8**      Collect online data $(s^{k,n}, a^{k,n}, \widetilde{a}^{k,n}, y^{k,n}) \sim \rho \times \pi^{k,n} \times \pi^k_{\text{OL}} \times \mathbb{P}_{r^*}(\cdot|\cdot, \cdot, \cdot)$.

**9**    **end**

**10**    $\pi^{k+1}_{\text{OL}} \leftarrow \text{Alg}_{\text{PO}}(\pi^k_{\text{OL}}, \mathcal{D}^{k,N+1})$;

**11** **end**

**12** **return** $\pi^{K+1}_{\text{OL}}$.

---

source tasks are present, i.e., $W = 0$. As stated below, TPO achieves $\widetilde{O}(\sqrt{T})$ regret over time by purely utilizing "self-transfer learning", thereby strictly improving existing results (Cen et al., 2024; Xie et al., 2024; Xiong et al., 2024; Zhang et al., 2024)[4].

**Corollary 4.5.** *When $W = 0$, under the same condition of Thm. 4.4, TPO reduces to a standard online RLHF method with $\widetilde{O}(\sqrt{T})$ regret after finite time.*

## 5. From Theory to an Empirical Algorithm

In terms of computational overheads, TPO requires solving multiple minimax optimization problems, which restricts its applicability to fine-tune LLMs in practice. To address this, adhering to the design principles of TPO, we introduce a more computationally efficient alternative in Alg. 3.

**Key Insight: Estimating Win Rates instead of Policy Values** As discussed in Sec. 4, several optimization steps are designed to estimate policy values used for transfer policy selection, because they help to identify the policies' coverability for optimal policy (i.e. $\text{Cov}^{\pi^*_{r^*}|\cdot}$). The key insight in our empirical algorithm design is to *find a more accessible indicator to infer* $\text{Cov}^{\pi^*_{r^*}|\cdot}$. This leads us to the policy win rates, i.e., the probability that human prefer the generation by one policy over another. Formally, given two policies $\pi, \widetilde{\pi}$, the win rate of $\widetilde{\pi}$ over $\pi$ is defined by: $\mathbb{P}_{r^*}(\widetilde{\pi} \succ \pi) := \mathbb{E}_{s \sim \rho, a \sim \widetilde{\pi}, a' \sim \pi}[\mathbb{P}_{r^*}(y = 1|s, a, a')]$.

Win rates between two policies can be unbiasedly estimated by querying human preferences with their generated responses. Moreover, win rates can be used to construct a lower bound for $\text{Cov}^{\pi^*_{r^*}|\cdot}$, as stated in Lem. 5.1 below.

**Lemma 5.1.** *Under BT-model[5], for any $\pi$: $\text{Cov}^{\pi^*_{r^*}|\pi} \geq$*

$$\max_{\gamma>0, \bar{\pi}} \left(\sqrt{(\gamma + 2\mathbb{P}_{r^*}(\pi \succ \bar{\pi})) \log \frac{1+\gamma}{\gamma}} + \sqrt{\frac{J_\beta(\pi^*_{r^*}) - J_\beta(\bar{\pi})}{2\beta}}\right)^{-1}.$$

Note that we may not identify the policy with the best coverage for $\pi^*_{r^*}$ through the lower bound above. However, it still provides useful guidance for practice: we can filter out policies yielding high lower bound. In Lem. 5.1, for any fixed $\gamma$ and comparator $\bar{\pi}$, the lower bound for $\text{Cov}^{\pi^*_{r^*}|\pi}$ increases as $\mathbb{P}_{r^*}(\pi \succ \bar{\pi})$ decay to 0, suggesting prioritizing transferring from policies with high win rates.

The key question now is how to choose the comparator $\bar{\pi}$. According to Lem. 5.1, ideally, the comparator should be close to $\pi^*_{r^*}$, so that $J_\beta(\pi^*_{r^*}) - J_\beta(\bar{\pi})$ becomes negligible, allowing the win rate term to dominate the lower bound. Since we do not know $\pi^*_{r^*}$ in advance, empirically, we can choose the learning policy as the comparator, which is optimized and progressively converges to $\pi^*_{r^*}$.

**From Insights to Practice** Next, we walk through empirical TPO in Alg. 3 and explain how we integrate these insights into the algorithm design. Alg. 3 utilizes an iterative online learning framework, which repeatedly collects online data and optimizes the policy. We start by initializing the online learning policy $\pi^1_{\text{OL}}$ with the reference policy $\pi_{\text{ref}}$. For computational efficiency, in each iteration $k$, we avoid separately computing online exploration policies and self-transfer learning policies as done in TPO. Instead, we only compute one policy $\pi^k_{\text{OL}}$ (updated from $\pi^{k-1}_{\text{OL}}$) by $\text{Alg}_{\text{PO}}$. Here $\text{Alg}_{\text{PO}}$ is a placeholder for an arbitrary **P**olicy **O**ptimization algorithm, and we do not restrict the concrete choice. Such a design increases the modularity of our em-

---

[4]While omitted in our result, the dependence on the log-covering number (e.g., $\log |\Pi|$) matches with those previous works.

[5]Lem. 5.1 can be generalized beyond BT-model. Besides, it is possible to construct a lower bound involving $\mathbb{P}_{r^*}(\bar{\pi} \succ \pi)$ instead. See Lem. G.3 and Remark G.4 in Appx. G for more details.

pirical TPO, making it possible to combine with various policy optimization methods and enhance their performance. For example, $\text{Alg}_{\text{PO}}$ may be instantiated by DPO (Rafailov et al., 2024), resulting in a transfer learning framework built on iterative-DPO (Xiong et al., 2024; Yuan et al., 2024). Besides, one may consider other advanced (online) methods, such as XPO (Xie et al., 2024), IPO (Azar et al., 2024), etc.

As the core ingredients of our empirical TPO, during data collection, the algorithm selects the policy $\pi^{k,n} \in \{\pi_{r^*}^*\}_{w=1}^W \cup \{\pi_{\text{OL}}^k\}$ with the highest win rate when competing against $\pi_{\text{OL}}^k$. Intuitively, we encourage transfer learning if $\{\pi_{r^*}^*\}_{w=1}^W$ includes high-quality candidates; otherwise, the algorithm conducts standard iterative policy optimization with $\text{Alg}_{\text{PO}}$ by default. This strategy also aligns with the heuristic principle: ***learn from an expert until surpassing it***. Lastly, since the win rates are unknown in advance, the selection process is formulated as a multi-armed bandit problem. We employ a UCB subroutine (line 7) to balance the exploration and exploitation during the win rates estimation.

# 6. Experiments

In this section, we evaluate Alg. 3 on the summarization task using the XSum dataset (Narayan et al., 2018). We consider T5 series models (Raffel et al., 2020) and choose T5-small (80M) as the base model for fine-tuning. Human reward $r^*$ is simulated by the reward model (Dong et al., 2024) distilled from Llama3-8B (Dubey et al., 2024). For the source rewards in transfer learning, we consider a collection of imperfect reward models and LLM policies, including, (a) ROUGE-Lsum score (Lin, 2004), (b) BERTScore (Zhang et al., 2019), (c) T5-base (250M), (d) T5-large (770M). As illustrated in Sec. 2.2, for LLM policies (c) and (d), we treat their log-probability predictions on the given prompt $s$ and response $a$ as the reward scores.

For $\text{Alg}_{\text{PO}}$, we consider three instantiations: DPO (Rafailov et al., 2024), IPO (Azar et al., 2024) and XPO (Xie et al., 2024). To save space, we present and interpret the results with DPO as the choice below and defer other experiment results and also the concrete setups to Appx. I.

**Experiment Results and Discussion** We run Alg. 3 for $K = 3$ iterations and compare its performance with three baselines: (I) vanilla iterative-DPO without transfer learning (i.e., setting $\text{Alg}_{\text{PO}} = $ DPO and $W = 0$); (II) purely exploiting the worst source reward—ROUGE score; (III) purely exploiting the best source reward—T5-large. Concretely, baseline (I) removes the transfer learning component in Alg. 3 by assigning $\pi^{k,n} = \pi_{\text{OL}}^k$ for all $n \in [N]$. For baselines (II) and (III), the worst (ROUGE-LSum) and best (T5-Large) reward models from the candidates (a)-(d) are selected, and pure transfer learning is then performed using the responses recommended by the chosen reward models,

|  | Without Transfer | Purely Exploit ROUGE-Lsum | Purely Exploit T5-Large |
|---|---|---|---|
| Iter 1 | $52.1 \pm 1.2$ | $53.1 \pm 1.1$ | $49.5 \pm 0.9$ |
| Iter 2 | $53.3 \pm 1.6$ | $54.5 \pm 1.3$ | $49.1 \pm 0.4$ |
| Iter 3 | $54.0 \pm 1.2$ | $53.3 \pm 1.5$ | $50.6 \pm 0.3$ |

*Table 1.* Win rates (%) of the policies trained by empirical TPO (Alg. 3) are competed with 3 baselines, presented across 3 columns. Baseline (I): without transfer, i.e., iterative-DPO. Baseline (II): purely utilizing ROUGE-LSum (the lowest-quality source task) in transfer learning. Baseline (III): purely utilizing T5-Large (the highest-quality source task) in transfer learning. Results are averaged with 3 random seeds and 95% confidence levels are reported.

i.e., $\pi^{k,n} = \pi_{r^w}^*$ in Alg. 3 for the selected $r^w$. Here the worst and best reward models are selected based on the final policy value when aligning with the given reward model.

Table. 1 reports the win rates of the policies learned by Alg. 3 competing with the three baselines. As shown in Column 1, comparing with normal online learning, our transfer strategy demonstrates clear advantages. Furthermore, Column 2 and 3 suggest that, without prior knowledge of source tasks quality, our method avoids being misled by low-quality tasks and achieves competitive performance compared to exploiting the best reward candidate.

Notably, as suggested by the additional results in Appx. I, $\pi_{\text{OL}}^k$ improves over time and $\mathbb{P}_{r^*}(\pi_{r^w}^* \succ \pi_{\text{OL}}^k)$ for any $w \in [W]$ continuously decreases. In iteration 3, our empirical TPO automatically switches back to online learning and avoids being restricted by the sub-optimality of source reward models. In the end, it results in higher win rates than purely exploiting the best source reward model T5-Large over 3 iterations.

# 7. Conclusion

This paper studies reward transfer in the context of online RLHF. We contribute TPO, a provable and efficient transfer learning algorithm that leverages the structure induced by the KL regularizer. Based on that, we further develop a UCB-based empirical alternative and evaluate its effectiveness through LLM experiments. Several promising directions remain for future exploration. Firstly, an interesting avenue is to develop transfer learning strategies beyond RLHF setting, for example, the Nash Learning from Human Feedback setting. Secondly, while we focus on policy-level transfer, a finer-grained prompt-wise knowledge transfer may be possible, which allows transfer from different policies in different states. Thirdly, due to resource limitations, we leave the examination of our methods in fine-tuning much larger-scale language models to the future work.

## Acknowledgement

The work is supported by ETH research grant and Swiss National Science Foundation (SNSF) Project Funding No. 200021-207343 and SNSF Starting Grant.

## Impact Statement

This paper presents work whose goal is to advance the field of Machine Learning. There are many potential societal consequences of our work, none which we feel must be specifically highlighted here.

## Reproducibility Statement

The code of all the experiments and the running instructions can be found in https://github.com/jiaweihhuang/RLHF_RewardTransfer.

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

## Outline of the Appendix

# A. Frequently Used Notation

| Notation | Description |
|---|---|
| $\mathcal{S}, \mathcal{A}$ | State space and action space |
| $\rho$ | Prompt distribution (initial state distribution) |
| $r$ | Reward model |
| $r^*$ | Ground-truth reward model (reflecting human preferences) |
| $\mathbb{P}_r(y\|s, a, a')$ | Preference under $r$ |
| $\mathbb{P}_r(\pi \succ \widetilde{\pi})$ | Win rate of $\pi$ over $\widetilde{\pi}$ under $r$ |
| $\{r^w\}_{w \in [W]}$ | Imperfect source reward model |
| $\pi$ | LLM policy |
| $\pi_{\mathrm{mix}}^t$ | Uniform mixture policy $\frac{1}{t} \sum_{i \le t} \pi^i$ of a policy sequence $\pi^1, ..., \pi^t$. Sometimes, given a dataset $\mathcal{D} = \{(x^i, \pi^i)\}_{i \le \|\mathcal{D}\|}$, with a bit abuse of notation, we use $\pi_{\mathrm{mix}}^{\mathcal{D}}$ to refer the mixture policy $\frac{1}{\|\mathcal{D}\|} \sum_{i \le \|\mathcal{D}\|} \pi^i$. |
| $\mathrm{Cov}^{\widetilde{\pi}\|\pi}$ | Coverage coefficient |
| $\Pi$ | The policy class |
| $\mathcal{R}^{\Pi}$ | The reward function class converted from $\Pi$, see Appx. B.1 |
| $\mathrm{conv}(\Pi)$ | Convex hull of $\Pi$ |
| $\beta$ | Regularization coefficient in RLHF objective |
| $J_{\beta}(\cdot)$ | Regularized policy value (Eq. (1)) |
| $\Delta(w)$ | Value gap for $\pi_{r^w}^*$, i.e. $J_{\beta}(\pi_{r^*}^*) - J_{\beta}(\pi_{r^w}^*)$ |
| $\Delta_{\min}$ | Minimal value gap $\min_{w \in [W]} \Delta(w)$ |
| $a \wedge b$ | $\min\{a, b\}$ |
| $[n]$ | $\{1, 2, ..., n\}$ |
| $O(\cdot), \Omega(\cdot), \Theta(\cdot), \widetilde{O}(\cdot), \widetilde{\Omega}(\cdot), \widetilde{\Theta}(\cdot)$ | Standard Big-O notations, $\widetilde{(\cdot)}$ omits the log terms. |

For completeness, we provide the definition of convex hull here.

**Definition A.1** (Convex Hull). Given a policy class $\Pi$ with finite cardinality (i.e. $\|\Pi\| < +\infty$), we denote $\mathrm{conv}(\Pi)$ as its convex hull, such that, $\forall n \in [\mathbb{N}^*]$, $\forall \lambda^1, ..., \lambda^n \ge 0$ with $\sum_{i=1}^n \lambda^i = 1$, and any $\pi^1, ..., \pi^n \in \Pi$, we have:

$$\sum_{i=1}^n \lambda^i \pi^i \in \mathrm{conv}(\Pi).$$

*Remark* A.2. Note that in the contextual bandit setting, the state action density induced by a policy and the policy distribution collapse with each other. Therefore, given a policy sequence $\pi^1, ..., \pi^t$, the uniform mixture policy $\pi_{\mathrm{mix}}^t(\cdot\|\cdot) = \frac{1}{t} \sum_{i \le t} \pi^t(\cdot\|\cdot)$ is directly a valid policy as a mapping from $\mathcal{S}$ to $\Delta(\mathcal{A})$, which induces the state-action density $\pi_{\mathrm{mix}}^t(\cdot\|\cdot)$.

Besides, we will use $\mathrm{OL}$ and $\mathrm{OFF}$ as abbreviations of "online learning" and "offline learning", respectively.

# B. Missing Details in the Main Text

## B.1. Extended Preliminary

**More Elaborations on the Necessity of Regularization**    The RLHF objective Eq. (1) typically involves a regularization term $\beta \ne 0$. This regularization is critical in practice for several reasons. Firstly, it prevents overfitting to human preferences, which can possibly be noisy and biased (Gao et al., 2023; Ouyang et al., 2022). Moreover, pure reward maximization prefers near-deterministic policies, potentially causing mode collapse. In contrast, regularization encourages the fine-tuned model to retain diversity from the reference policy (Jaques et al., 2017; 2019). Thirdly, reference policies are pretrained on a significantly larger corpus than the post-training data, enabling them to encode more general-purpose knowledge. Regularization helps mitigate catastrophic forgetting, ensuring the model retains this broad knowledge base.

**Formal Definition for $\mathcal{R}^\Pi$** Given a policy class $\Pi$ satisfying Assump. A, we use $\mathcal{R}^\Pi$ to denote the reward function class converted from $\Pi$, such that (1) $\forall r \in \mathcal{R}^\Pi$, $r(\cdot, \cdot) \in [0, R]$; (2) $\exists r \in \mathcal{R}^\Pi$, $\pi_r^* = \pi_{r^*}^*$. A possible construction satisfying this is given by

$$\mathcal{R}^\Pi := \{r_{|\pi} | r_{|\pi}(s, a) := \text{Clip}_{[0,R]}[\beta \log \frac{\pi(a|s)}{\pi_{\text{ref}}(a|s)} - \min_{a' \in \mathcal{A}} \beta \log \frac{\pi(a'|s)}{\pi_{\text{ref}}(a'|s)}], \ \pi \in \Pi\}. \tag{11}$$

The rationale behind such a construction lies in that $r_{|\pi_{r^*}^*}$ provably differs from $r^*$ by at most of an action-independent constant under Assump. A. We prove this in the following.

For any $s \in \mathcal{S}$, we denote $a_s := \arg\min_{a' \in \mathcal{A}} \log \frac{\pi_{r^*}^*(a'|s)}{\pi_{\text{ref}}(a'|s)}$. According to Eq. (2) and the fact that $r^* \in [0, R]$, for any $s \in \mathcal{S}$ and $a, a' \in \mathcal{A}$, we have:

$$0 \le \beta \log \frac{\pi_{r^*}^*(a|s)}{\pi_{\text{ref}}(a|s)} - \min_{a' \in \mathcal{A}} \beta \log \frac{\pi_{r^*}^*(a'|s)}{\pi_{\text{ref}}(a'|s)} = r^*(s, a) - r^*(s, a_s) \le R,$$

where the first inequality is because $a_s$ takes the minimal over $\mathcal{A}$. Therefore, $r_{|\pi_{r^*}^*}(s, a) \in [0, R]$ and $r_{|\pi_{r^*}^*}(s, a) - r^*(s, a) = r^*(s, a_s)$, which is action-independent. In another word, $r_{|\pi_{r^*}^*}$ induces the same optimal policy $\pi_{r^*}^*$.

Under the objective in Eq. (1), the realizability assumption in (Liu et al., 2024) can be relaxed and the reward model class $\mathcal{R}^\Pi$ can be used in their RPO objective. Because any per-state action-independent shift on the reward space does not change the induced policy in Eq. (2). As a result, throughout this paper, we will not distinguish between $r^*$ and $r_{|\pi_{r^*}^*}$.

**Remarks on Assumption A-(II)**

**Lemma B.1.** *If $r^*(s, a) \in [0, R]$ for all $(s, a) \in \mathcal{S} \times \mathcal{A}$, we have:*

$$\max_{s \in \mathcal{S}, a \in \mathcal{A}} |\log \frac{\pi_{r^*}^*(a|s)}{\pi_{ref}(a|s)}| \le \frac{R_{\max}}{\beta}.$$

*Proof.* By definition, for any $s$,

$$\forall a \in \mathcal{A}, \quad \pi_{r^*}^*(a|s) = \pi_{\text{ref}}(a|s) e^{\frac{r^*(s,a)}{\beta}} / Z(s),$$

where $Z(s) := \sum_{a \in \mathcal{A}} \pi_{\text{ref}}(a|s) e^{\frac{r^*(s,a)}{\beta}}$. Because $r^*(s, a) \in [0, R]$, obviously, $1 \le Z(s) \le e^{\frac{R}{\beta}}$. Therefore,

$$\forall a \in \mathcal{A}, \quad |\log \frac{\pi_{r^*}^*(a|s)}{\pi_{\text{ref}}(a|s)}| = |\frac{r^*(s,a)}{\beta} - \log Z(s)| \le \max\{\frac{R}{\beta} - \log Z(s), \log Z(s)\} \le \frac{R}{\beta}.$$

$\square$

### B.2. Other Related Works

**Other Related RLHF Literature** Various approaches have been developed for reward-model-free online exploration. For example, DPO (Rafailov et al., 2024) implicitly optimizes the same objective as RLHF without explicit reward modeling. DPO is further extended to different settings; see e.g., online DPO (Guo et al., 2024), iterative DPO (Dong et al., 2024; Pang et al., 2024; Xu et al., 2023), etc.

Another direction is to go beyond Bradley-Terry reward model assumption. A particularly promising set of techniques formulates RLHF as a two-player zero-sum game (Yue et al., 2012), aiming to select policies preferred by the rater to others (Munos et al., 2023; Rosset et al., 2024; Swamy et al., 2024; Ye et al., 2024). Investigating knowledge transfer within this framework is an exciting direction for future work.

**RL Theory in Pure-Reward Maximization Setting** In the classical pure-reward maximization RL setting, sample efficiency is a central topic, with extensive research dedicated to strategic exploration and fundamental complexity measures for online learning (Du et al., 2021; Foster et al., 2021; Jiang et al., 2017; Jin et al., 2021a; Russo & Van Roy, 2013).

Besides the literature already mentioned in Sec. 2.3, there is a rich literature (Jiang & Huang, 2020; Jin et al., 2021b; Uehara et al., 2020; Xie et al., 2021) investigating the role of policy coverage (or density ratio) in offline learning.

**Regularized RL** Sample complexity in regularized RL has also been studied in previous works (Geist et al., 2019; Tiapkin et al., 2023; Ziebart, 2010; Ziebart et al., 2008). Nonetheless, most of them focus on tabular settings and do not consider the transfer learning.

## C. Offline Learning Results in Previous Literature

In this section, we recall and adapt some results from (Liu et al., 2024), which are useful for proofs in other places.

**RPO Optimization Objective** For completeness, we provide the optimization objective of RPO. Given a policy class $\widetilde{\Pi}$ and a reward function class $\mathcal{R}$, the RPO objective solves a mini-max optimization problem defined as follows:

$$\text{RPO}(\widetilde{\Pi}, \mathcal{R}, \mathcal{D}, \eta) = \arg\max_{\pi \in \widetilde{\Pi}} \min_{r \in \mathcal{R}} L_{\mathcal{D}}(r) + \eta \mathbb{E}_{s \sim \rho, a \sim \pi, \widetilde{a} \sim \pi_{\texttt{ref}}}[r(s, a) - r(s, \widetilde{a})] - \beta \text{KL}(\pi \| \pi_{\texttt{ref}}), \tag{12}$$

where we choose $\pi_{\texttt{ref}}$ as the base policy in (Liu et al., 2024). In Alg. 2, we set $\widetilde{\Pi} = \text{conv}(\Pi)$ and $\mathcal{R} = \mathcal{R}^{\Pi}$.

**Condition C.1** (Sequential Data Generation). We say a dataset $\mathcal{D} := \{(s^i, a^i, \widetilde{a}^i, y^i, \pi^i)\}_{i \leq |\mathcal{D}|}$ is generated sequentially, if it is generated following:

$$\forall i \leq |\mathcal{D}|, \quad \pi^i \sim \text{Alg}(\cdot | \{(s^j, a^j, \widetilde{a}^j, y^j, \pi^j)\}_{j < i}),$$
$$s^i \sim \rho, \ a^i \sim \pi^i(\cdot | s^i), \ \widetilde{a}^i \sim \pi_{\texttt{ref}}(\cdot | s^i), \ y^i \sim \mathbb{P}_{r^*}(\cdot | s^i, a^i, \widetilde{a}^i),$$

where Alg denotes an algorithm computing the next policy only with the interaction history.

**Lemma C.2.** *[Adapted from Thm. 5.3 in (Liu et al., 2024)] Under Assump. A, given any $\delta \in (0, 1)$, by running RPO (Eq. (12)) with $conv(\Pi), \mathcal{R}^{\Pi}, \delta$ and a dataset $\mathcal{D} := \{(s^i, a^i, \widetilde{a}^i, y^i, \pi^i)\}_{i \leq |\mathcal{D}|}$ satisfying Cond. C.1, by choosing $\eta = (1 + e^R)^2 \sqrt{24 |\mathcal{D}| \log \frac{|\Pi|}{\delta}}$, we have:*

$$\forall \pi \in conv(\Pi), \quad J_{\beta}(\pi) - J_{\beta}(\pi_{Dstl}) \leq C_{OFF} e^{2R} \cdot COV^{\pi | \pi_{mix}^{\mathcal{D}}} \sqrt{\frac{1}{|\mathcal{D}|} \log \frac{|\Pi|}{\delta}},$$

*where we use $\pi_{mix}^{\mathcal{D}} := \frac{1}{|\mathcal{D}|} \sum_{i \leq |\mathcal{D}|} \pi^i$ as a short note of the uniform mixture policy.*

*Proof.* The main difference comparing with (Liu et al., 2024) is that we consider sequentially generated dataset while they study dataset generated by a fixed dataset distribution. In the following, we show how to extend their results to our setting.

Firstly, we check the assumptions. Note that we consider feed RPO (Liu et al., 2024) by the reward function class $\mathcal{R}^{\Pi}$ converted from a policy class $\Pi$ satisfying Assump. A, through Eq. (11). Therefore, the optimal reward is also realizabile in $\mathcal{R}^{\Pi}$, and the basic assumptions required by RPO (Liu et al., 2024) are satisfied.

Next, we adapt the proofs in (Liu et al., 2024). Note that we can directly start with their Eq. (D.4), because their bounds in Eq. (D.2) and Eq. (D.3) only involve optimality of the choice of $\pi_{\text{Dstl}}$ and realizability. We move the KL-regularization terms to the LHS and merge to $J_{\beta}(\pi)$ and $J_{\beta}(\pi_{\text{Dstl}})$, and we choose $\pi_{\texttt{ref}}$ as the base policy in RPO. The adapted results to our notations would be:

$$\forall \pi \in \text{conv}(\Pi), \ J_{\beta}(\pi) - J_{\beta}(\pi_{\text{Dstl}})$$
$$\leq \max_{r \in \mathcal{R}^{\Pi}} \mathbb{E}_{s \sim \rho, a \sim \pi(\cdot|s), \widetilde{a} \sim \pi_{\texttt{ref}}(\cdot|s)}[(r^*(s, a) - r^*(s, \widetilde{a})) - (r(s, a) - r(s, \widetilde{a}))]$$
$$+ \eta^{-1}(\mathcal{L}_{\mathcal{D}}(r^*) - \mathcal{L}_{\mathcal{D}}(r)).$$

Recall $\mathcal{L}_{\mathcal{D}}$ is the (unnormalized) negative log-likelihood (NLL) loss, defined in Eq. (5). Since the dataset $\mathcal{D}$ is generated sequentially (Cond. C.1), we can apply the concentration results in Lem. H.2, which is a variant of Lemma D.1 in (Liu et al., 2024) for sequentially generated data:

$$\text{w.p. } 1 - \delta, \quad \forall \pi \in \text{conv}(\Pi), \ J_{\beta}(\pi) - J_{\beta}(\pi_{\text{Dstl}})$$
$$\leq \mathbb{E}_{s \sim \rho, a \sim \pi(\cdot|s), \widetilde{a} \sim \pi_{\texttt{ref}}(\cdot|s)}[(r^*(s, a) - r^*(s, \widetilde{a})) - (r_{\leftarrow \pi}(s, a) - r_{\leftarrow \pi}(s, \widetilde{a}))]$$

$$+ \eta^{-1}(\mathcal{L}_{\mathcal{D}}(r^*) - \mathcal{L}_{\mathcal{D}}(r_{\leftarrow\pi}))$$

$$\leq \mathbb{E}_{s\sim\rho, a\sim\pi(\cdot|s), \widetilde{a}\sim\pi_{\mathrm{ref}}(\cdot|s)}[(r^*(s,a) - r^*(s,\widetilde{a})) - (r_{\leftarrow\pi}(s,a) - r_{\leftarrow\pi}(s,\widetilde{a}))]$$

$$- \frac{1}{\eta|\mathcal{D}|} \sum_{i\leq|\mathcal{D}|} \mathbb{E}_{s\sim\rho, a\sim\pi^i(\cdot|s), \widetilde{a}\sim\pi_{\mathrm{ref}}(\cdot|s)}[\mathbb{H}^2(\mathbb{P}_{r_{\leftarrow\pi}}(\cdot|s,a,\widetilde{a})\|\mathbb{P}_{r^*}(\cdot|s,a,\widetilde{a}))] + \frac{2}{\eta|\mathcal{D}|} \log\frac{|\Pi|}{\delta}$$

$$= \mathbb{E}_{s\sim\rho, a\sim\pi(\cdot|s), \widetilde{a}\sim\pi_{\mathrm{ref}}(\cdot|s)}[(r^*(s,a) - r^*(s,\widetilde{a})) - (r_{\leftarrow\pi}(s,a) - r_{\leftarrow\pi}(s,\widetilde{a}))]$$

$$- \frac{1}{\eta} \mathbb{E}_{s\sim\rho, a\sim\pi_{\mathrm{mix}}^{\mathcal{D}}(\cdot|s), \widetilde{a}\sim\pi_{\mathrm{ref}}(\cdot|s)}[\mathbb{H}^2(\mathbb{P}_{r_{\leftarrow\pi}}(\cdot|s,a,\widetilde{a})\|\mathbb{P}_{r^*}(\cdot|s,a,\widetilde{a}))] + \frac{2}{\eta|\mathcal{D}|} \log\frac{|\Pi|}{\delta},$$

where we denote $r_{\leftarrow\pi} := \arg\max_{r\in\mathcal{R}^\Pi} \mathbb{E}_{s\sim\rho, a\sim\pi(\cdot|s), \widetilde{a}\sim\pi_{\mathrm{ref}}(\cdot|s)}[(r^*(s,a) - r^*(s,\widetilde{a})) - (r(s,a) - r(s,\widetilde{a}))]$.

The rest of the proofs in (Liu et al., 2024) can be adapted here, and by choosing $\eta = (1+e^R)^{-2}\sqrt{\frac{24}{|\mathcal{D}|}\log\frac{|\Pi|}{\delta}}$, we can inherit the following guarantee:

$$\text{w.p. } 1-\delta, \quad \forall\pi\in\text{conv}(\Pi), \ J_\beta(\pi) - J_\beta(\pi_{\mathrm{Dstl}}) \leq \frac{\sqrt{6}}{4}\cdot(1+e^R)^2(C_{\pi_{\mathrm{mix}}^{\mathcal{D}}}(\mathcal{R}^\Pi; \pi; \pi_{\mathrm{ref}})^2 + 1)\sqrt{\frac{1}{|\mathcal{D}|}\log\frac{|\Pi|}{\delta}}.$$

Here $C_{\pi_{\mathrm{mix}}^{\mathcal{D}}}(\mathcal{R}^\Pi; \pi; \pi_{\mathrm{ref}})$ is the coverage coefficient (adapted from Assump. 5.2 (Liu et al., 2024)) with $\pi_{\mathrm{mix}}^{\mathcal{D}}$, which can be upper bounded by:

$$C_{\pi_{\mathrm{mix}}^{\mathcal{D}}}(\mathcal{R}^\Pi; \pi; \pi_{\mathrm{ref}}) \leq \max_{r\in\mathcal{R}^\Pi} \frac{\mathbb{E}_{s\sim\rho, a\sim\pi(\cdot|s), \widetilde{a}\sim\pi_{\mathrm{ref}}(\cdot|s)}[|(r^*(s,a) - r^*(s,\widetilde{a})) - (r(s,a) - r(s,\widetilde{a}))|]}{\sqrt{\mathbb{E}_{s\sim\rho, a\sim\pi_{\mathrm{mix}}^{\mathcal{D}}(\cdot|s), \widetilde{a}\sim\pi_{\mathrm{ref}}(\cdot|s)}[|(r^*(s,a) - r^*(s,\widetilde{a})) - (r(s,a) - r(s,\widetilde{a}))|^2]}}$$

$$\leq \sqrt{\mathbb{E}_{s\sim\rho, a\sim\pi(\cdot|s)}\left[\frac{\pi(a|s)}{\pi_{\mathrm{mix}}^{\mathcal{D}}(a|s)}\right]}$$

(AM-GM inequality; Holds for any $r$ and therefore including the one achieves the maximum)

$$= \sqrt{\mathrm{Cov}^{\pi|\pi_{\mathrm{mix}}^{\mathcal{D}}}}.$$

Therefore, we finish the proof. We simplify the upper bound by using $\mathrm{Cov}^{\pi|\pi_{\mathrm{mix}}^{\mathcal{D}}} \geq 1$ and $e^R \geq 1$. $\square$

## D. Details for Online Learning Oracle Example in Sec. 4

**Definition D.1** ($L_\infty$ Coverability; (Xie et al., 2022; 2024))**.** The $L_\infty$ coverability is defined by:

$$\mathrm{Cov}_\infty(\Pi) := \inf_{\mu\in\Delta(\mathcal{S})\times\Delta(\mathcal{A})} \sup_{\pi\in\Pi} \sup_{s\in\mathcal{S}, a\in\mathcal{A}} \frac{\pi(a|s)}{\mu(a|s)}$$

**Definition D.2** (No-Regret Online Algorithm)**.** Given any $\delta\in(0,1)$, iteration number $\widetilde{T}$, and a policy class $\Pi$ satisfying Assump. A, the online learning algorithm $\texttt{Alg}_{\mathrm{OL}}$ iteratively computes policy to collect samples and conducts no-regret learning. W.p. $1-\delta$, it produces a sequence of online policies $\pi^1, ..., \pi^{\widetilde{T}}$, such that, $\forall t\in[\widetilde{T}]$,

$$\sum_{i\leq t} J_\beta(\pi_{r^*}^*) - J_\beta(\pi^i) \leq C_{\mathrm{OL}}Re^{2R}\sqrt{\mathcal{C}(\Pi)t\log^{c_0}\frac{|\Pi|\widetilde{T}}{\delta}},$$

where $C_{\mathrm{OL}} > 0$ and $c_0 \geq 1$ are absolute constants, and $\mathcal{C}(\Pi)$ denotes some complexity measure for $\Pi$.

**Proposition D.3.** *[Example for Online Oracle in Def. D.2] The XPO algorithm in (Xie et al., 2024) can fulfill the requirements in Def. D.2.*

*Proof.* We start by generalizing Eq.(35) in the proof of Theorem 3.1 in (Xie et al., 2024) to all $t\in[\widetilde{T}]$. Note that we consider bandit setting so $R$ in (Xie et al., 2024) collapse with $R$. Suppose at the end of iteration $t$, XPO generated a sequence of policies $\pi^1, ..., \pi^t$, we have:

$$\frac{1}{t}\sum_{i=1}^t J_\beta(\pi_{r^*}^*) - J_\beta(\pi^i)$$

$$\leq \frac{6R}{t} + \frac{\text{SEC}_{\text{RLHF}}(\Pi, t, \beta, \pi_{\text{ref}})}{2\eta t} + \frac{\eta}{2}R^2 + \frac{1}{t}\mathbb{E}_{i=2}^t \mathbb{E}_{s \sim \rho, a \sim \pi_{\text{ref}}(\cdot|s)}[\beta \log \pi^t(a|s) - \beta \log \pi^*(a|s)]$$
$$+ \frac{\eta}{2t}\sum_{i=2}^t (i-1)\mathbb{E}_{s \sim \rho, a \sim \pi_{\text{mix}}^{i-1}(\cdot|s), \tilde{a} \sim \text{ref}(\cdot|s)}\left[\left(\beta \log \frac{\pi^i(a|s)}{\pi_{\text{ref}}(a'|s)} - r^*(s, a) - \beta \log \frac{\pi^i(\tilde{a}|s)}{\pi_{\text{ref}}(\tilde{a}|s)} + r^*(s, \tilde{a})\right)^2\right],$$

where we choose $\tilde{\pi}^{(t)}$ in (Xie et al., 2024) to be $\pi_{\text{ref}}$ and denote $\pi_{\text{mix}}^{i-1} = \frac{1}{i-1}\sum_{j=1}^{i-1}\pi^j$.

Note that Lemma C.5 in (Xie et al., 2024) holds w.p. $1 - \delta$ for all $t \in [\tilde{T}]$. Following their proofs till Eq.(43) in (Xie et al., 2024), we can show that for any $t \in [\tilde{T}]$

$$\frac{1}{t}\sum_{i=1}^t J_\beta(\pi_{r^*}^*) - J_\beta(\pi^i) \leq O(Re^{2R}\sqrt{\frac{\text{SEC}_{\text{RLHF}}(\Pi, t, \beta, \pi_{\text{ref}})}{t}\log\frac{|\Pi|T}{\delta}}),$$

Based on the arguments in (Xie et al., 2024), $\text{SEC}_{\text{RLHF}}(\Pi, t, \beta, \pi_{\text{ref}})$ can be controlled by $c_0 \cdot \text{Cov}_\infty^\Pi \log^{c_1}(|\Pi|t)$ for some absolute constant $c_0, c_1 > 0$. Here $\text{Cov}_\infty^\Pi$ is the $L_\infty$ coverability coefficient (Def. D.1) and plays the role. Therefore, we finish the verification. $\square$

# E. Proofs for Results in Section 3

### E.1. Proof for Lemma 3.1

We first introduce some useful results from (Sason & Verdú, 2016). Given two probability distribution $P, Q \in \Delta(\mathcal{A})$, we use $D_{+\infty}(P\|Q)$ to denote the Renyi divergence of order $\alpha = +\infty$. We follow the definition of $\chi^2$-divergence in (Sason & Verdú, 2016) as follows:

$$\chi^2(P\|Q) = \mathbb{E}_{s \sim P}\left[\frac{P(x)}{Q(x)}\right] - 1.$$

**Lemma E.1** (Theorem 7 in (Sason & Verdú, 2016))**.** *Given $P, Q \in \Delta(\mathcal{A})$, such that $P \neq Q$ and $P(a), Q(a) > 0$ for all $a \in \mathcal{A}$, we have:*

$$KL(P\|Q) \leq \kappa_1(e^{D_{+\infty}(P\|Q)}) \cdot KL(Q\|P),$$

*where $\kappa_1 : (0, 1) \cup (1, +\infty) \to (0, +\infty)$, $\kappa_1(t) = \frac{t \log t + (1-t)}{(t-1) - \log t}$.*

**Lemma E.2** (Eq. 182; Theorem 9 in (Sason & Verdú, 2016) for $\alpha = 2$)**.** *Under the same condition as Lem. E.1,*

$$\chi^2(P\|Q) \leq \frac{KL(P\|Q)}{\kappa_2(e^{D_{+\infty}(P\|Q)})},$$

*where $\kappa_2(t) := \frac{t \log t + (1-t)}{(t-1)^2}$.*

**Lemma E.3.** *For any policy $\pi$,*

$$J_\beta(\pi_{r^*}^*) - J_\beta(\pi) = \beta\mathbb{E}_{s \sim \rho}[KL(\pi(\cdot|s)\|\pi_{r^*}^*(\cdot|s))].$$

*Proof.* A shorter proof can be done by directly assigning $\nu = \pi$ in Lemma 3.1 of (Xie et al., 2024), and here we provide another one without detouring through it.

$$J_\beta(\pi_{r^*}^*) - J_\beta(\pi)$$
$$=\mathbb{E}_{s \sim \rho, a \sim \pi_{r^*}^*}[r^*(s, a)] - \mathbb{E}_{s \sim \rho, a \sim \pi}[r^*(s, a)] - \beta\mathbb{E}_{s \sim \rho, a \sim \pi_{r^*}^*}[\log\frac{\pi_{r^*}^*(a|s)}{\pi_{\text{ref}}(a|s)}] + \beta\mathbb{E}_{s \sim \rho, a \sim \pi}[\log\frac{\pi(a|s)}{\pi_{\text{ref}}(a|s)}]$$
$$=\beta\mathbb{E}_{s \sim \rho, a \sim \pi_{r^*}^*}[\log\frac{\pi_{r^*}^*(a|s)}{\pi_{\text{ref}}(a|s)}] - \beta\mathbb{E}_{s \sim \rho, a \sim \pi}[\log\frac{\pi_{r^*}^*(a|s)}{\pi_{\text{ref}}(a|s)}] - \beta\mathbb{E}_{s \sim \rho, a \sim \pi_{r^*}^*}[\log\frac{\pi_{r^*}^*(a|s)}{\pi_{\text{ref}}(a|s)}] + \beta\mathbb{E}_{s \sim \rho, a \sim \pi}[\log\frac{\pi(a|s)}{\pi_{\text{ref}}(a|s)}]$$

$$
\begin{aligned}
&= \beta \mathbb{E}_{s\sim\rho, a\sim\pi} [\log \frac{\pi(a|s)}{\pi_{r^*}^*(a|s)}] \\
&= \beta \mathbb{E}_{s\sim\rho} [\mathrm{KL}(\pi(\cdot|s)\|\pi_{r^*}^*(\cdot|s))].
\end{aligned}
$$

where the second equality holds because for any $s, a$

$$
r^*(s, a) = \beta \log \frac{\pi_{r^*}^*(a|s)}{\pi_{\mathrm{ref}}(a|s)} + Z(s)
$$

for some $Z(s)$ independent w.r.t. $a$. □

**Lemma 3.1.** *Under Assump. A and assume $r^w \in [0, R]$ for all $w \in [W]$, then, for any policy $\pi \in conv(\Pi) \cup \{\pi_{r^w}^*\}_{w=1}^W$,*

$$
Cov^{\pi_{r^*}^*|\pi} \le 1 + \kappa(e^{\frac{2R}{\beta}}) \cdot \frac{J_\beta(\pi_{r^*}^*) - J_\beta(\pi)}{\beta}, \tag{3}
$$

*where $\kappa(x) := \frac{(x-1)^2}{x-1-\log x} = O(x)$.*

We prove a stronger result in Lem. E.4 below, where we consider the policy class including all the policy having bounded ratio with $\pi_{\mathrm{ref}}$.

$$
\Pi_{\le \frac{R}{\beta}} := \{\pi : \mathcal{S} \to \Delta(\mathcal{A}) | \max_{s,a} |\log \frac{\pi(a|s)}{\pi_{\mathrm{ref}}(a|s)}| \le \frac{R}{\beta}\}.
$$

Lem. 3.1 then holds directly as a corollary by combining with Lem. H.1, Lem. B.1 and the fact that $r^w \in [0, R]$ for all $w \in [W]$.

**Lemma E.4.** *For any policy $\pi \in \Pi_{\le \frac{R}{\beta}}$,*

$$
Cov^{\pi_{r^*}^*|\pi} \le 1 + \kappa(e^{\frac{2R}{\beta}}) \cdot \frac{J_\beta(\pi_{r^*}^*) - J_\beta(\pi)}{\beta}, \tag{13}
$$

*where $\kappa(x) := \frac{(x-1)^2}{x-1-\log x} = O(x)$.*

*Proof.* Given any $\pi \in \Pi_{\le \frac{R}{\beta}}$, we consider a fixed $s > 0$, and apply Lem. E.1 and Lem. E.2 with $P = \pi_{r^*}^*(\cdot|s)$ and $Q = \pi(\cdot|s)$. Since those two lemmas holds when $P \ne Q$, we first check the case when $\pi_{r^*}^*(\cdot|s) \ne \pi(\cdot|s)$:

$$
\begin{aligned}
\mathbb{E}_{a\sim\pi_{r^*}^*(a|s)} [\frac{\pi_{r^*}^*(a|s)}{\pi(a|s)}] - 1 &= \chi^2(\pi_{r^*}^*(\cdot|s)\|\pi(\cdot|s)) \le \frac{1}{\kappa_2(\zeta)} \mathrm{KL}(\pi_{r^*}^*(\cdot|s)\|\pi(\cdot|s)) \\
&\le \frac{1}{\kappa_2(\zeta)} \cdot \kappa_1(\zeta) \cdot \mathrm{KL}(\pi(\cdot|s)\|\pi_{r^*}^*(\cdot|s)) \\
&= \frac{(\zeta - 1)^2}{\zeta - 1 - \log \zeta} \cdot \mathrm{KL}(\pi(\cdot|s)\|\pi_{r^*}^*(\cdot|s)).
\end{aligned}
$$

where we use $\zeta := e^{D_{+\infty}(\pi_{r^*}^*(\cdot|s)\|\pi(\cdot|s))} > 1$ as a short note.

We define $\kappa(x) = \frac{(x-1)^2}{x-1-\log x}$. Note that,

$$
\begin{aligned}
\kappa'(x) &= \frac{2(x-1)}{x-1-\log x} - \frac{(x-1)^2(1-x^{-1})}{(x-1-\log x)^2} \\
&= \frac{x-1}{x-1-\log x} \frac{2(x-1) - 2\log x - (x-1)(1-x^{-1})}{x-1-\log x} \\
&= \frac{x-1}{x-1-\log x} \frac{x - x^{-1} - 2\log x}{x-1-\log x}.
\end{aligned}
$$

Now, we consider $g(x) := x - x^{-1} - 2\log x$ for $x \in (1, +\infty)$. Note that, $g(1) = 0$ and

$$g'(x) = 1 + \frac{1}{x^2} - \frac{2}{x} \geq 0.$$

Therefore, $\kappa'(x) \geq 0$, which implies $\kappa(x)$ is increasing for all $x > 1$.

Under Assump. A,

$$D_{+\infty}(\pi_{r^*}^*(\cdot|s)\|\pi(\cdot|s)) = \log\exp(\max_a \frac{\pi_{r^*}^*(a|s)}{\pi(a|s)}) \leq \frac{2R}{\beta},$$

which implies $\zeta \leq e^{\frac{2R}{\beta}}$. Therefore,

$$\mathbb{E}_{a\sim\pi_{r^*}^*(a|s)}[\frac{\pi_{r^*}^*(a|s)}{\pi(a|s)}] - 1 \leq \kappa(e^{\frac{2R}{\beta}}) \cdot \mathrm{KL}(\pi(\cdot|s)\|\pi_{r^*}^*(\cdot|s)).$$

Note that the above inequality also holds when $\pi(\cdot|s) = \pi_{r^*}^*(\cdot|s)$. Therefore, combining with Lem. E.3, we have:

$$\mathrm{Cov}^{\pi_{r^*}^*|\pi} = \mathbb{E}_{s\sim\rho, a\sim\pi_{r^*}^*(a|s)}[\frac{\pi_{r^*}^*(a|s)}{\pi(a|s)}] \leq 1 + \kappa(e^{\frac{2R}{\beta}}) \cdot \mathbb{E}_{s\sim\rho}[\mathrm{KL}(\pi(\cdot|s)\|\pi_{r^*}^*(\cdot|s))]$$

$$= 1 + \kappa(e^{\frac{2R}{\beta}}) \cdot \frac{J_\beta(\pi_{r^*}^*) - J_\beta(\pi)}{\beta}.$$

$\square$

### E.2. Another Bound for Policy Coverage Coefficient

In the following, we provide another bound for the coverage coefficient between the optimal policies induced by different reward models. Although we do not use this lemma in the proofs for other results in this paper, it indicates a different upper bound, and possibly, it is tighter than the one in Lem. 3.1 in some cases.

**Lemma E.5.** *Under Assump. A, given any bounded reward model $r$, and the associated optimal policy $\pi_r^*$ (defined by Eq. (1)), the coverage coefficient between $\pi_r^*$ and $\pi_{r^*}^*$ can be controlled by:*

$$\mathrm{Cov}^{\pi_{r^*}^*|\pi_r^*} \leq \min_{b\in\mathbb{R}} \mathbb{E}_{s\sim\rho}[\mathbb{E}_{a\sim\pi_{r^*}^*}^2[\exp(\frac{|r^*(s,a) - r(s,a) - b|}{\beta})]].$$

*Proof.* By definition, the state-wise coverage coefficient

$$\mathrm{Cov}^{\pi_{r^*}^*|\pi_r^*}(s) := \mathbb{E}_{a\sim\pi_{r^*}^*(\cdot|s)}[\frac{\pi_{r^*}^*(a|s)}{\pi_r^*(a|s)}]$$

$$= \mathbb{E}_{a\sim\pi_{r^*}^*(\cdot|s)}[\exp(\frac{r^*(s,a) - r(s,a)}{\beta})] \cdot \frac{Z_r(s)}{Z_{r^*}(s)}$$

Here we denote $Z_r(s) = \sum_a \pi_{\mathrm{ref}}(a|s)\exp(\frac{1}{\beta}r(s,a))$ and similar for $Z_{r^*}(s)$. Therefore,

$$\frac{Z_r(s)}{Z_{r^*}(s)} = \sum_a \frac{\pi_{\mathrm{ref}}(a|s)\exp(\frac{1}{\beta}r(s,a))}{Z_{r^*}(s)} = \sum_a \pi_{r^*}^*(s) \cdot \exp(\frac{1}{\beta}(r(s,a) - r^*(s,a))) = \mathbb{E}_{a\sim\pi_{r^*}^*}[\exp(\frac{r(s,a) - r^*(s,a)}{\beta})].$$

We remark that one important fact we leverage in the second equality is that $\pi_{r^*}^*(a|s) > 0$ for all $a \in \mathcal{A}$. Considering introducing an arbitrary $b \in \mathcal{R}$, we should have:

$$\mathrm{Cov}^{\pi_{r^*}^*|\pi_r^*} = \mathbb{E}_{a\sim\pi_{r^*}^*(\cdot|s)}[\exp(\frac{r^*(s,a) - r(s,a) + b}{\beta})] \cdot \mathbb{E}_{a\sim\pi_{r^*}^*(\cdot|s)}[\exp(\frac{r(s,a) - \widetilde{r}(s,a) - b}{\beta})]$$

$$\leq \mathbb{E}_{a\sim\pi_{r^*}^*(\cdot|s)}^2[\exp(\frac{|r^*(s,a) - r(s,a) + b|}{\beta})]$$

Given that $b$ is arbirtary, we can pick the best one:

$$\mathrm{Cov}^{\pi_{r^*}^*|\pi_r^*} \leq \min_{b\in\mathbb{R}} \mathbb{E}_{a\sim\pi_{r^*}^*}^2[\exp(\frac{|r^*(s,a) - r(s,a) - b|}{\beta})].$$

$\square$

## E.3. Proof for Theorem 3.2

**Theorem 3.2.** *Under Assump. A, w.p. $1 - \delta$, given an online dataset $\mathcal{D}$ generated[6] by a policy series $\{\pi^t\}_{t=1}^T \in conv(\Pi)$, running RPO with $conv(\Pi)$ and $\mathcal{R}^\Pi$ on $\mathcal{D}$ yields a distilled policy $\pi_{Dstl}$, such that,*

$$J_\beta(\pi_{r^*}^*) - J_\beta(\pi_{Dstl}) \leq \tag{4}$$
$$\widetilde{O}\Big(e^{2R}\Big(1 + \kappa(e^{\frac{2R}{\beta}}) \sum_{t=1}^T \frac{J_\beta(\pi_{r^*}^*) - J_\beta(\pi^t)}{\beta T}\Big)\sqrt{\frac{1}{T}}\Big).$$

We refer to Lem. C.2 for the detailed hyperparameter setups.

*Proof.* By Lem. C.2, w.p. $1 - \delta$,

$$J_\beta(\pi_{r^*}^*) - J_\beta(\pi_{\text{Dstl}}) \leq C_{\text{OFF}} e^{2R} \cdot \text{Cov}^{\pi_{r^*}^* | \pi_{\text{mix}}^T} \sqrt{\frac{1}{T} \log \frac{|\Pi|}{\delta}},$$

where $\pi_{\text{mix}}^T := \frac{1}{T} \sum_{t \in [T]} \pi^t$ is the uniform mixture policy, and the coverage coefficient can be upper bounded by:

$$\text{Cov}^{\pi_{r^*}^* | \pi_{\text{mix}}^T} \leq 1 + \kappa(e^{\frac{2R}{\beta}}) \cdot \frac{J_\beta(\pi_{r^*}^*) - J_\beta(\pi_{\text{mix}}^T)}{\beta} \tag{Lem. 3.1}$$

$$\leq 1 + \kappa(e^{\frac{2R}{\beta}}) \cdot \sum_{t=1}^T \frac{J_\beta(\pi_{r^*}^*) - J_\beta(\pi_{\text{mix}}^t)}{\beta T}$$

Here in the last step, we use the fact that KL divergence is convex, and therefore, $\text{KL}(\pi_{\text{mix}}^T \| \pi_{\text{ref}}) \leq \frac{1}{T} \sum_{t=1}^T \text{KL}(\pi^t \| \pi_{\text{ref}})$, which implies $J_\beta(\pi_{\text{mix}}^T) \geq \frac{1}{T} \sum_{t=1}^T J_\beta(\pi^t)$.

$\square$

**Implication for Online RLHF**  If we consider the policy sequence generated by a no-regret online learning algorithm, we have the following corollary.

**Corollary E.6.** *Under Assump. A, suppose $\pi^1, ..., \pi^T$ is generated by a no-regret online learning algorithm with $\sum_{t=1}^T J_\beta(\pi_{r^*}^*) - J_\beta(\pi^t) = \widetilde{O}(\mathcal{C}(\Pi)\sqrt{T})$ for some structural complexity measure $\mathcal{C}(\Pi)$, as long as $T = \widetilde{\Omega}(\beta^{-2}\mathcal{C}(\Pi)^2\kappa^2(e^{\frac{2R}{\beta}}))$, running RPO yields an offline policy s.t. $J_\beta(\pi_{r^*}^*) - J_\beta(\pi_{Dstl}) = \widetilde{O}(e^{2R}T^{-\frac{1}{2}})$.*

The proof is straightforward by noting that $\sum_{t=1}^T \frac{J_\beta(\pi_{r^*}^*) - J_\beta(\pi^t)}{\beta T} = O(\mathcal{C}(\Pi)\sqrt{T}/T)$, which decays to 0 as $T$ increases. Coro. E.6 is remarkable as it implies an $\widetilde{O}(\varepsilon^{-2})$ sample complexity bound to learn an $\varepsilon$-optimal policy for online RLHF (for $\varepsilon$ smaller than a threshold), which **does not depend on** the number of states and actions or other complexity measures. In contrast, in previous online RLHF literature (Cen et al., 2024; Xie et al., 2024; Xiong et al., 2024; Zhang et al., 2024), for the uniform mixture policy $\pi_{\text{mix}}^T := \frac{1}{T}\sum_{t=1}^T \pi^t$, the regret-to-PAC conversion implies a value gap $J_\beta(\pi_{r^*}^*) - J_\beta(\pi_{\text{mix}}^T) = \widetilde{O}(\sqrt{\frac{\mathcal{C}(\Pi)}{T}})$, which has an additional factor $\mathcal{C}(\Pi)$ regarding the complexity of the function class. This suggests a strict improvement.

Moreover, this marks a fundamental difference from the pure reward maximization setting, where lower bounds depending on those factors has been established (Auer et al., 2002; Dani et al., 2008).

**Other Previous Works Reporting Faster Convergence Rate**  Several recent works also report faster convergence rate than the information-theoretic lower bounds for online pure reward maximization RL, by exploiting the structure induced by KL regularization. (Shi et al., 2024) investigates the tabular softmax parametrization setting and establishes quadratic convergence results. In contrast, our result is more general, applying to arbitrary policy class.

The work of (Zhao et al., 2024) is more related to ours. They consider general reward function classes and derive an $O(\varepsilon^{-1}\text{Poly}(D))$ sample complexity bound, where $D$ is a coefficient related to the coverage of the distribution $\rho \times \pi_{\text{ref}}$.

---

[6]See Cond. C.1 for the definition of data generation process.

While their dependence on $\varepsilon$ is better than ours, their definition of $D$ is not always satisfactory. For example, in the worst case one would have $D = \Omega(\frac{1}{\min_{s \in \mathcal{S}} \rho(s)})$. This indicates that their bound scales with the number of states, once noticing that $\frac{1}{\min_{s \in \mathcal{S}} \rho(s)}$ is no smaller than $|\mathcal{S}|$. In contrast, the largest coverage-related coefficient in our result is $O(\kappa^2(e^{\frac{2R}{\beta}})) = O(e^{\frac{4R}{\beta}})$, which remains small and is free of $|\mathcal{S}|$. Therefore, our Coro. E.6 can outperform the bound in (Zhao et al., 2024) in many scenarios.

More importantly, the primary focus of our work is on reward transfer, which is orthogonal to these studies.

# F. Proofs for the Main Algorithm and Results in Sec. 4

## F.1. Additional Algorithm Details

**Missing Details for TPO (Alg. 1)** For any given $(k, n) \in [K] \times [N]$, we use $\mathcal{D}^{k,n} := \cup_{i<k \text{ or } i=k,j<n}\{s^{i,j}, a^{i,j}, \tilde{a}^{i,j}, y^{i,j}, \pi^{i,j}\}$ to denote all the collected data up to step $(k, n)$; $\mathcal{D}_{\text{OL}}^{k,n} := \cup_{i<k, j \leq \alpha N \text{ or } i=k, j \leq n \wedge \alpha N}\{s^{i,j}, a^{i,j}, \tilde{a}^{i,j}, y^{i,j}, \pi^{i,j}\}$ denotes the data collected by $\text{Alg}_{\text{OL}}$ up to step $(k, n)$.

## F.2. Some Useful Lemmas

**Lemma F.1** (MLE Reward Estimation Error). *In each call of Alg. 2 with a policy class $\Pi$ satisfying Assump. A and a dataset $\mathcal{D}$ generated by a sequence of policies $\pi^1, ..., \pi^{|\mathcal{D}|}$, then, for any policy $\pi$, given any $\delta \in (0, 1)$, with probability at least $1 - \delta$, for all $w \in [W]$, we have:*

$$\left| \left( \mathbb{E}_{\rho,\pi}[r^*] - \mathbb{E}_{\rho,\pi_{ref}}[r^*] \right) - \left( \mathbb{E}_{\rho,\pi}[\hat{r}_{MLE}] - \mathbb{E}_{\rho,\pi_{ref}}[\hat{r}_{MLE}] \right) \right| \leq 16 e^{2R} \sqrt{\frac{\text{Cov}^{\pi|\pi_{mix}^{\mathcal{D}}}}{|\mathcal{D}|} \cdot \log \frac{|\Pi|}{\delta}},$$

*where we use $\pi_{mix}^{\mathcal{D}} := \frac{1}{|\mathcal{D}|} \sum_{i \leq |\mathcal{D}|} \pi^i$ as a short note.*

*Proof.* For any policy $\pi \in \Pi$, by applying Lem. H.3 with $\pi_{mix}^{\mathcal{D}}$ and $r \leftarrow \hat{r}_{MLE}$, we have:

$$\left| \left( \mathbb{E}_{\rho,\pi}[r^*] - \mathbb{E}_{\rho,\pi_{ref}}[r^*] \right) - \left( \mathbb{E}_{\rho,\pi}[\hat{r}_{MLE}] - \mathbb{E}_{\rho,\pi_{ref}}[\hat{r}_{MLE}] \right) \right|$$

$$\leq \mathbb{E}_{s \sim \rho, a \sim \pi(\cdot|s), \tilde{a} \sim \pi_{ref}(\cdot|s)}[|\left( r^*(s,a) - r^*(s,\tilde{a}) \right) - \left( \hat{r}_{MLE}(s,a) - \hat{r}_{MLE}(s,\tilde{a}) \right)|]$$

$$\leq 8\sqrt{2}e^{2R} \sqrt{\text{Cov}^{\pi|\pi_{mix}^{\mathcal{D}}} \cdot \frac{1}{|\mathcal{D}|} \cdot \sum_{i \leq |\mathcal{D}|} \mathbb{E}_{s \sim \rho, a \sim \pi^i(\cdot|s), \tilde{a} \sim \pi_{ref}(\cdot|s)}[\mathbb{H}^2(\mathbb{P}_{\hat{r}_{MLE}}(\cdot|s,a,\tilde{a}) \| \mathbb{P}_{r^*}(\cdot|s,a,\tilde{a}))]}.$$

By applying Lem. H.2, and the fact that $\hat{r}_{MLE}, r^* \in \mathcal{R}^{\Pi}$, for any $\delta \in (0,1)$, w.p. $1 - \delta$, we have:

$$\frac{1}{|\mathcal{D}|} \sum_{i \leq |\mathcal{D}|} \mathbb{E}_{s \sim \rho, a \sim \pi^i(\cdot|s), \tilde{a} \sim \pi_{ref}(\cdot|s)}[\mathbb{H}^2(\mathbb{P}_{\hat{r}_{MLE}}(\cdot|s,a,\tilde{a}) \| \mathbb{P}_{r^*}(\cdot|s,a,\tilde{a}))]$$

$$\leq L_{\mathcal{D}}(\hat{r}_{MLE}) - L_{\mathcal{D}}(r^*) + \frac{2}{|\mathcal{D}|} \log \frac{|\Pi|}{\delta}$$

$$\leq \frac{2}{|\mathcal{D}|} \log \frac{|\Pi|}{\delta}. \qquad \text{(Assump. A and } \hat{r}_{MLE} \text{ minimizes the negative log-likelihood)}$$

Therefore, we finish the proof. $\qquad \square$

**Lemma 4.2** (Value Est Error for $\{\pi_{r^w}^*\}_{w \in [W]}$). *Under Assump. A and Def. D.2, w.p. $1 - \delta$, in each call of Alg. 2:*

$$\forall w \in [W], \quad J_{\beta}(\pi_{r^w}^*) - J_{\beta}(\pi_{ref}) \leq \hat{V}(\pi_{r^w}^*; \mathcal{D})$$

$$\leq J_{\beta}(\pi_{r^w}^*) - J_{\beta}(\pi_{ref}) + \tilde{O}\left(\frac{e^{2R}}{\sqrt{N(w; \mathcal{D})}}\right).$$

*Proof.* Note that $\frac{\text{Cov}^{\pi^*_{rw}|\pi^{\mathcal{D}}_{\text{mix}}}}{|\mathcal{D}|} \leq \frac{1}{N(w;\mathcal{D})}$, where we recall that $N(w;\mathcal{D}) := \sum_{i \leq |\mathcal{D}|} \mathbb{I}[\pi^i = \pi^*_{rw}]$ denotes the number of occurrences of $\pi^*_{rw}$ in the dataset. By Lem. F.1, w.p. $1 - \delta'$, for all $w \in [W]$, and any $(k, n) \in [K] \times [N]$ occurs in the call of Alg. 1 such that $n > \alpha N$:

$$\left| \left( \mathbb{E}_{\rho,\pi^*_{rw}}[r^*] - \mathbb{E}_{\rho,\pi_{\text{ref}}}[r^*] \right) - \left( \mathbb{E}_{\rho,\pi^*_{rw}}[\widehat{r}_{\text{MLE}}] - \mathbb{E}_{\rho,\pi_{\text{ref}}}[\widehat{r}_{\text{MLE}}] \right) \right| \leq 16e^{2R} \sqrt{\frac{1}{N(w;\mathcal{D}^{k,n})} \log \frac{|\Pi|W}{\delta'}}.$$

Recall

$$\widehat{V}(\pi^*_{rw};\mathcal{D}) := \mathbb{E}_{\rho,\pi^*_{rw}}[\widehat{r}_{\text{MLE}}] - \mathbb{E}_{\rho,\pi_{\text{ref}}}[\widehat{r}_{\text{MLE}}] - \beta \text{KL}(\pi^*_{rw}\|\pi_{\text{ref}}) + 16e^{2R} \sqrt{\frac{1}{N(w;\mathcal{D}^{k,n})} \log \frac{|\Pi|WT}{\delta}}.$$

By taking the union bound for all $T$ iterations (choosing $\delta' = \delta/T$), we finish the proof. $\qquad\square$

**Lemma F.2** (Estimation Error for Self-Transfer Policy). *For any $k > 1$ and $\alpha N < n \leq N$, in each call of Alg. 2 in the iteration $(k, n)$ of Alg. 1 with a dataset $\mathcal{D} := \{(s^i, a^i, \widetilde{a}^i, y^i, \pi^i)\}_{i \leq |\mathcal{D}|}$ satisfying Cond. C.1, then, given any $\delta \in (0, 1)$, w.p. $1 - \delta$:*

$$\widehat{V}(\pi_{Dstl};\mathcal{D}) \leq J_\beta(\pi_{Dstl}) - J_\beta(\pi_{ref})$$

$$\widehat{V}(\pi_{Dstl};\mathcal{D}) \geq J_\beta(\pi^*_{r*}) - J_\beta(\pi_{ref}) - c' \cdot Re^{2R} \cdot \left( Cov^{\pi^*_{r*}|\pi^{\mathcal{D}}_{mix}} \wedge \frac{\sqrt{\mathcal{C}(\Pi)}}{\alpha} \right) \cdot \sqrt{\frac{1}{|\mathcal{D}|} \log^{c_0} \frac{|\Pi|T}{\delta}},$$

*where we use $\pi^{\mathcal{D}}_{mix} := \frac{1}{|\mathcal{D}|} \sum_{i \leq |\mathcal{D}|} \pi^{|\mathcal{D}|}$ as a short note, and $c'$ is some absolute constant.*

*Proof.* Recall that

$$\widehat{V}(\pi_{\text{Dstl}};\mathcal{D}) := \mathbb{E}_{\rho,\pi_{\text{Dstl}}}[\widehat{r}_{\text{Dstl}}] - \mathbb{E}_{\rho,\pi_{\text{ref}}}[\widehat{r}_{\text{Dstl}}] - \beta \text{KL}(\pi_{\text{Dstl}}\|\pi_{\text{ref}})$$
$$+ \frac{1}{\eta} L_{\mathcal{D}}(\widehat{r}_{\text{Dstl}}) - \frac{1}{\eta} L_{\mathcal{D}}(\widehat{r}_{\text{MLE}}) - \texttt{bonus},$$

Here we use $\texttt{bonus} := 2c \cdot e^{2R} \sqrt{\frac{1}{|\mathcal{D}|} \log \frac{|\Pi|T}{\delta}}$ as a short note of the bonus term. By definition,

$$\widehat{V}(\pi_{\text{Dstl}};\mathcal{D})$$

$$\leq \mathbb{E}_{\rho,\pi_{\text{Dstl}}}[r^*] - \mathbb{E}_{\rho,\pi_{\text{ref}}}[r^*] - \beta \text{KL}(\pi_{\text{Dstl}}\|\pi_{\text{ref}}) + \frac{1}{\eta} L_{\mathcal{D}}(r^*) - \frac{1}{\eta} L_{\mathcal{D}}(\widehat{r}_{\text{MLE}}) - \texttt{bonus}$$
$$\text{(Pessimistic estimation of } \widehat{r}_{\text{Dstl}} \text{ in Eq. (12))}$$

$$\leq J_\beta(\pi_{\text{Dstl}}) - J_\beta(\pi_{\text{ref}}) + \frac{2}{\eta|\mathcal{D}|} \log \frac{|\Pi|}{\delta} - \texttt{bonus} \qquad\qquad\qquad\qquad \text{(Lem. H.2)}$$

$$\leq J_\beta(\pi_{\text{Dstl}}) - J_\beta(\pi_{\text{ref}}) + 2c \cdot e^{2R} \sqrt{\frac{1}{|\mathcal{D}|} \log \frac{|\Pi|T}{\delta}} - \texttt{bonus}. \qquad\qquad\qquad (14)$$

The last step is because of our choice of $\eta = (1 + e^R)^{-2} \sqrt{\frac{24}{|\mathcal{D}|} \log \frac{|\Pi|T}{\delta}}$.

For the lower bound, note that for any policy $\pi \in \text{conv}(\Pi)$, we have:

$$J_\beta(\pi) - J_\beta(\pi_{\text{ref}}) - \widehat{V}(\pi_{\text{Dstl}};\mathcal{D})$$

$$= \left( \mathbb{E}_{\rho,\pi}[r^*] - \mathbb{E}_{\rho,\pi_{\text{ref}}}[r^*] - \beta \text{KL}(\pi\|\pi_{\text{ref}}) \right)$$

$$- \left( \mathbb{E}_{\rho,\pi_{\text{Dstl}}}[\widehat{r}_{\text{Dstl}}] - \mathbb{E}_{\rho,\pi_{\text{ref}}}[\widehat{r}_{\text{Dstl}}] - \beta \text{KL}(\pi_{\text{Dstl}}\|\pi_{\text{ref}}) + \frac{1}{\eta} L_{\mathcal{D}}(\widehat{r}_{\text{Dstl}}) \right) + \frac{1}{\eta} L_{\mathcal{D}}(\widehat{r}_{\text{MLE}}) + \texttt{bonus}$$

$$\leq \left( \mathbb{E}_{\rho,\pi}[r^*] - \mathbb{E}_{\rho,\pi_{\text{ref}}}[r^*] - \beta \text{KL}(\pi\|\pi_{\text{ref}}) \right) - \min_{r \in \mathcal{R}^\Pi} \left( \mathbb{E}_{\rho,\pi}[r] - \mathbb{E}_{\rho,\pi_{\text{ref}}}[r] - \beta \text{KL}(\pi\|\pi_{\text{ref}}) + \frac{1}{\eta} L_{\mathcal{D}}(r) \right)$$
$$\text{(Optimality of } \pi_{\text{Dstl}} \text{ in RPO;)}$$

$$+ \frac{1}{\eta} L_{\mathcal{D}}(r^*) + \texttt{bonus} \qquad\qquad (\widehat{r}_{\text{MLE}} \text{ minimizes } L_{\mathcal{D}})$$

$$\leq \mathbb{E}_{s \sim \rho, a \sim \pi, \widetilde{a} \sim \pi_{\text{ref}}}[|r^*(s,a) - r^*(s,\widetilde{a}) - r_{\pi;\mathcal{D}}(s,a) + r_{\pi;\mathcal{D}}(s,\widetilde{a})|] + \frac{1}{\eta} L_{\mathcal{D}}(r^*) - \frac{1}{\eta} L_{\mathcal{D}}(r_{\pi;\mathcal{D}}) + \texttt{bonus}$$
$$\text{(We use } r_{\pi;\mathcal{D}} \text{ to denote the reward achieves the above minimum)}$$

$$\leq \frac{2}{\eta|\mathcal{D}|} \log \frac{|\Pi|}{\delta} + 8\sqrt{2} e^{2R} \sqrt{\frac{\text{Cov}^{\pi|\pi_{\text{mix}}^{\mathcal{D}}}}{|\mathcal{D}|} \cdot \sum_{i \leq |\mathcal{D}|} \mathbb{E}_{s \sim \rho, a \sim \pi^i(\cdot|s), \widetilde{a} \sim \pi_{\text{ref}}(\cdot|s)}[\mathbb{H}^2(\mathbb{P}_{r_{\pi;\mathcal{D}}}(\cdot|s,a,\widetilde{a}) \| \mathbb{P}_{r^*}(\cdot|s,a,\widetilde{a}))]}$$

$$- \frac{1}{\eta} \sum_{i \leq |\mathcal{D}|} \mathbb{E}_{s \sim \rho, a \sim \pi^i, \widetilde{a} \sim \pi_{\text{ref}}}[\mathbb{H}^2(\mathbb{P}_{r_{\pi;\mathcal{D}}}(\cdot|s,a,\widetilde{a}) \| \mathbb{P}_{r^*}(\cdot|s,a,\widetilde{a}))] + \texttt{bonus} \qquad \text{(Lem. H.2 and Lem. H.3)}$$

$$\leq \frac{2}{\eta|\mathcal{D}|} \log \frac{|\Pi|}{\delta} + 64\eta e^{4R} \frac{\text{Cov}^{\pi|\pi_{\text{mix}}^{\mathcal{D}}}}{|\mathcal{D}|} + \texttt{bonus} \qquad\qquad (ax - bx^2 \leq \frac{a^2}{4b})$$

$$\leq 4c_2 \cdot e^{2R} \cdot \text{Cov}^{\pi|\pi_{\text{mix}}^{\mathcal{D}}} \cdot \sqrt{\frac{1}{|\mathcal{D}|} \log \frac{|\Pi|T}{\delta}} + \texttt{bonus}. \qquad\qquad (15)$$

where the last step is because of our choice of $\eta = c \cdot (1 + e^R)^{-2} \sqrt{\frac{24}{|\mathcal{D}|} \log \frac{|\Pi|T}{\delta}}$.

Next, we evaluate some choice of $\pi$. We first consider $\pi = \pi_{r^*}^* \in \text{conv}(\Pi)$, the above result implies,

$$J_\beta(\pi_{r^*}^*) - J_\beta(\pi_{\text{ref}}) - \widehat{V}(\pi_{\text{Dstl}}; \mathcal{D}) \leq 4c_2 \cdot e^{2R} \cdot \text{Cov}^{\pi_{r^*}^*|\pi_{\text{mix}}^{\mathcal{D}}} \cdot \sqrt{\frac{1}{|\mathcal{D}|} \log \frac{|\Pi|T}{\delta}} + \texttt{bonus}. \qquad (16)$$

Secondly, we consider the mixture policy $\pi = \pi_{\text{mix}}^{k-1} := \frac{1}{\alpha(k-1)N} \sum_{i=1}^{k-1} \pi^{i,j} \in \text{conv}(\Pi)$. Because of the convexity of KL divergence, $J(\pi)$ is concave in $\pi$, by Jensen's inequality, we have:

$$J_\beta(\pi_{r^*}^*) - J_\beta(\pi_{\text{mix}}^{k-1}) = J_\beta(\pi_{r^*}^*) - \mathbb{E}_{s \sim \rho, a \sim \pi_{\text{mix}}^{k-1}(\cdot|s)}[r^*(s,a)] + \beta \text{KL}(\pi_{\text{mix}}^{k-1} \| \pi_{\text{ref}})$$

$$\leq J_\beta(\pi_{r^*}^*) - \frac{1}{\alpha(k-1)N} \sum_{i=1}^{k-1} \sum_{1 \leq j \leq \alpha N} \left( \mathbb{E}_{s \sim \rho, a \sim \pi_{\text{OL}}^i(\cdot|s)}[r^*(s,a)] - \beta \text{KL}(\pi^{i,j} \| \pi_{\text{ref}}) \right)$$

$$\leq C_{\text{OL}} R e^{2R} \sqrt{\frac{\mathcal{C}(\Pi)}{\alpha(k-1)N} \log^{c_0} \frac{|\Pi|T}{\delta}} \leq C_{\text{OL}} R e^{2R} \sqrt{\frac{2\mathcal{C}(\Pi)}{\alpha k N} \log^{c_0} \frac{|\Pi|T}{\delta}}. \qquad \text{(Cond. D.2)}$$

Note that $|\mathcal{D}| \pi_{\text{mix}}^{\mathcal{D}} \geq \alpha(k-1)N \pi_{\text{mix}}^{k-1}$, which implies $\text{Cov}^{\pi_{\text{mix}}^{k-1}|\pi_{\text{mix}}^{\mathcal{D}}} \leq \frac{|\mathcal{D}|}{\alpha(k-1)N} \leq \frac{kN}{\alpha(k-1)N} \leq \frac{2}{\alpha}$. Therefore, by Eq. (15),

$$J_\beta(\pi_{\text{mix}}^{k-1}) - J_\beta(\pi_{\text{ref}}) - \widehat{V}(\pi_{\text{Dstl}}; \mathcal{D}) \leq 4c_2 \cdot e^{2R} \cdot \frac{2}{\alpha} \cdot \sqrt{\frac{1}{|\mathcal{D}|} \log \frac{|\Pi|T}{\delta}}.$$

Combining the above two inequalities together, we have:

$$J_\beta(\pi_{r^*}^*) - J_\beta(\pi_{\text{ref}}) - \widehat{V}(\pi_{\text{Dstl}}; \mathcal{D})$$

$$\leq J_\beta(\pi_{r^*}^*) - J_\beta(\pi_{\text{mix}}^{k-1}) + J_\beta(\pi_{\text{mix}}^{k-1}) - J_\beta(\pi_{\text{ref}}) - \widehat{V}(\pi_{\text{Dstl}}; \mathcal{D})$$

$$\leq C_{\text{OL}} R e^{2R} \sqrt{\frac{2\mathcal{C}(\Pi)}{\alpha k N} \log^{c_0} \frac{|\Pi|T}{\delta}} + 4c_2 \cdot e^{2R} \cdot \frac{2}{\alpha} \cdot \sqrt{\frac{1}{|\mathcal{D}|} \log \frac{|\Pi|T}{\delta}} + \texttt{bonus}$$

$$\leq c_3 R \cdot e^{2R} \sqrt{\frac{\mathcal{C}(\Pi)}{\alpha^2 |\mathcal{D}|} \log^{c_0} \frac{|\Pi|T}{\delta}} + \texttt{bonus}. \qquad\qquad (17)$$

Therefore, under our choice of $\texttt{bonus} = 2c \cdot \sqrt{\frac{1}{|\mathcal{D}|} \log \frac{|\Pi|T}{\delta}}$, Eq. (14), Eq. (16) and Eq. (17) imply,

$$\widehat{V}(\pi_{\text{Dstl}}; \mathcal{D}) \leq J_\beta(\pi_{\text{Dstl}}) - J_\beta(\pi_{\text{ref}})$$

$$\widehat{V}(\pi_{\mathrm{Dstl}}; \mathcal{D}) \geq J_\beta(\pi_{r^*}^*) - J_\beta(\pi_{\mathrm{ref}}) - c' R e^{2R} \cdot \Big( \mathrm{Cov}^{\pi_{r^*}^* | \pi_{\mathrm{mix}}^{\mathcal{D}}} \wedge \frac{\sqrt{\mathcal{C}(\Pi)}}{\alpha} \Big) \cdot \sqrt{\frac{1}{|\mathcal{D}|} \log^{c_0} \frac{|\Pi| T}{\delta}}.$$

$\square$

**Lemma 4.3** (Value Est Error for $\pi_{\mathrm{Dstl}}$). *Under Assump. A and Def. D.2, w.p. $1 - \delta$, in each call of Alg. 2:*

$$J_\beta(\pi_{r^*}^*) - J_\beta(\pi_{\mathrm{ref}}) - \widetilde{O}\Big( \frac{R e^{2R}}{\sqrt{|\mathcal{D}|}} \cdot (\mathrm{Cov}^{\pi_{r^*}^* | \pi_{\mathrm{mix}}^{\mathcal{D}}} \wedge \frac{\sqrt{\mathcal{C}(\Pi)}}{\alpha}) \Big)$$

$$\leq \widehat{V}(\pi_{\mathrm{Dstl}}; \mathcal{D}) \leq J_\beta(\pi_{\mathrm{Dstl}}) - J_\beta(\pi_{\mathrm{ref}}). \tag{6}$$

*Proof.* By applying Lem. F.2 with appropriate constants, and taking the union bound over all iterations, we can finish the proof. $\square$

### F.3. Proof for Thm. 4.4

**Theorem 4.4** (Total Regret). *Suppose $\mathrm{Alg}_{\mathrm{OL}}$ is a no-regret instance satisfying Def. D.2, whose regret grows as $\widetilde{O}(R e^{2R} \sqrt{\mathcal{C}(\Pi)\widetilde{T}})$ for any intermediate step $\widetilde{T}$ and some policy class complexity measure $\mathcal{C}(\Pi)$. Then, w.p. $1 - 2\delta$, for any $T/K \leq t \leq T$, running TPO yields a regret bound:*

$$\sum_{\tau \leq t} J_\beta(\pi_{r^*}^*) - J_\beta(\pi^{k(\tau), n(\tau)}) = Reg_{OL}^{(t)} + Reg_{Trf}^{(t)}$$

$$Reg_{OL}^{(t)} := \widetilde{O}(R e^{2R} \sqrt{\alpha \mathcal{C}(\Pi) t}), \tag{7}$$

$$Reg_{Trf}^{(t)} := \widetilde{O}\Big( \sum_{\tau \leq t : \alpha N < n(\tau) \leq N} \Delta_{\min} \wedge \iota^{k(\tau), n(\tau)} \tag{8}$$

$$+ e^{2R} \sqrt{(1-\alpha) W t} \wedge \sum_{w : \Delta(w) > 0} \frac{e^{4R}}{\Delta(w)} \Big).$$

*Here we denote $k(\tau) := \lceil \frac{\tau}{N} \rceil$ and $n(\tau) := \tau \bmod N$ to be the block index and inner iteration index for step $\tau$; $\iota^{k(\tau), n(\tau)} := \widetilde{O}(R e^{2R} \big( \mathrm{Cov}^{\pi_{r^*}^* | \pi_{\mathrm{mix}}^\tau} \wedge \frac{\sqrt{\mathcal{C}(\Pi)}}{\alpha} \big) \sqrt{\frac{1}{\tau}})$, where $\pi_{\mathrm{mix}}^\tau := \frac{1}{\tau} \sum_{i \leq \tau} \pi^{k(i), n(i)}$ is the mixture policy up to $\tau$; $\Delta(w)$ and $\Delta_{\min}$ denote value gaps as defined in Sec. 2.2.*

Throught the proof, we follow the convention that $1/0 = +\infty$.

*Proof.* Since we divide the total budget $T$ to $K$ batches with batch size $N$, we will use two indices $\widetilde{K} \in [K]$ and $\widetilde{N} \in [N]$ to represent the current iteration number, i.e. the $\widetilde{N}$-th iteration in the $\widetilde{K}$-th batch. We will divide the indices of previous iterations to two parts, depending on whether we conduct normal online learning (the first $\alpha N$ samples in each batch) or do transfer learning (the rest $(1 - \alpha)N$ samples in each batch):

$$\mathcal{I}_{\widetilde{K}, \widetilde{N}}^{\mathrm{OL}} := \{(k, n) | k < \widetilde{K}, n \leq \alpha N, \text{ or } k = \widetilde{K}, n \leq \widetilde{N} \wedge \alpha N\},$$

$$\mathcal{I}_{\widetilde{K}, \widetilde{N}}^{\mathrm{Trf}} := \{(k, n) | k < \widetilde{K}, \alpha N < n \leq N, \text{ or } k = \widetilde{K}, \alpha N < n \leq \widetilde{N}\},$$

$$\mathcal{I}_{\widetilde{K}, \widetilde{N}} := \mathcal{I}_{\widetilde{K}, \widetilde{N}}^{\mathrm{OL}} \cup \mathcal{I}_{\widetilde{K}, \widetilde{N}}^{\mathrm{Trf}} = \{(k, n) | k < \widetilde{K}, n \leq N, \text{ or } k = \widetilde{K}, n \leq \widetilde{N}\}.$$

For the policies generated by online algorithm, under the condition in Def. D.2, w.p. $1 - \delta$, for any $\widetilde{K} \in [K], \widetilde{N} \in [N]$ we have:

$$\sum_{(k, n) \in \mathcal{I}_{\widetilde{K}, \widetilde{N}}^{\mathrm{OL}}} J_\beta(\pi_{r^*}^*) - J_\beta(\pi^{k, n}) \leq C_{\mathrm{OL}} R e^{2R} \sqrt{\mathcal{C}(\Pi) |\mathcal{I}_{\widetilde{K}, \widetilde{N}}^{\mathrm{OL}}| \log^{c_0} \frac{|\Pi| T}{\delta}}. \tag{18}$$

Next, we focus on the performance of transfer policies. We first introduce a few notation for convenience.

**Additional Notations** We use $\pi_{\text{Dstl}}^{k,n}$ to denote the offline policy computed by Alg. 2 called by Alg. 1 at iteration $(k,n)$ for some $\alpha N < n \leq N$. We denote $\mathcal{E}_{\text{Dstl}}^{k,n} := \{\pi_{\text{Dstl}}^{k,n} = \pi^{k,n}\}$ to be the event that Alg. 2 returns $\pi_{\text{Dstl}}^{k,n}$ as the policy, and use $\mathcal{E}_w^{k,n} := \{\pi_{r^w}^* = \pi^{k,n}\}$ to denote the event that Alg. 2 pick and return $\pi_{r^w}^*$. Besides, we use $\neg\mathcal{E}_{\text{Dstl}}^{k,n} := \bigcup_{w \in [W]} \mathcal{E}_w^{k,n}$ as a short note for the event that Alg. 2 does not return the offline policy $\pi_{\text{Dstl}}^{k,n}$. Recall the definition $\Delta(w) := J_\beta(\pi_{r^*}^*) - J_\beta(\pi_{r^w}^*)$, and $\Delta_{\min} = \min_{w \in [W]} \Delta(w)$. We will use $w^*$ to denote the index of the task achieves $\Delta_{\min}$ (or any of the tasks if multiple maximizers exist). Given the dataset $\mathcal{D}^{k,n}$ we use $\pi_{\text{mix}}^{k,n} := \frac{1}{|\mathcal{D}^{k,n}|} \sum_{i,j \in \mathcal{D}^{k,n}} \pi^{i,j}$ to be the uniform mixture policy from $\mathcal{D}^{k,n}$.

Then, we decompose the accumulative value gap depending on whether $\mathcal{E}_{\text{Dstl}}^{k,n}$ is true or not. We use $\mathbb{I}[\mathcal{E}]$ as the indicator function, which takes value 1 if $\mathcal{E}$ happens and otherwise 0. For any $\widetilde{K} \in [K], \widetilde{N} \in [N]$, we have:

$$\sum_{(k,n) \in \mathcal{I}_{\widetilde{K}, \widetilde{N}}^{\text{Trf}}} J_\beta(\pi_{r^*}^*) - J_\beta(\pi^{k,n})$$

$$= \sum_{(k,n) \in \mathcal{I}_{\widetilde{K}, \widetilde{N}}^{\text{Trf}}} \mathbb{I}[\mathcal{E}_{\text{Dstl}}^{k,n}](J_\beta(\pi_{r^*}^*) - J_\beta(\pi^{k,n})) + \sum_{(k,n) \in \mathcal{I}_{\widetilde{K}, \widetilde{N}}^{\text{Trf}}} \mathbb{I}[\neg\mathcal{E}_{\text{Dstl}}^{k,n}](J_\beta(\pi_{r^*}^*) - J_\beta(\pi^{k,n})). \tag{19}$$

**Part-(1) Upper Bound the First Part in Eq. (19)** We first bound the accumulative error when $\mathbb{I}[\mathcal{E}_{\text{Dstl}}^{k,n}] = 1$. On the good events in Lem. 4.2 and Lem. 4.3 (which holds w.p. $1 - \delta$), $\mathbb{I}[\mathcal{E}_{\text{Dstl}}^{k,n}] = 1$ implies

$$\widehat{V}(\pi_{\text{Dstl}}; \mathcal{D}^{k,n}) \geq \max_{w \in [W]} \widehat{V}(\pi_{r^w}^*; \mathcal{D}^{k,n}) \geq \max_{w \in [W]} J_\beta(\pi_{r^w}^*) - J_\beta(\pi_{\text{ref}}) = J_\beta(\pi_{r^*}^*) - J_\beta(\pi_{\text{ref}}) - \Delta_{\min},$$

and as implied by Lem. 4.3

$$J_\beta(\pi_{r^*}^*) - J_\beta(\pi_{\text{Dstl}}) \leq \Delta_{\min},$$

$$J_\beta(\pi_{r^*}^*) - J_\beta(\pi_{\text{Dstl}}) \leq c_2 Re^{2R} \cdot \left( \text{Cov}^{\pi_{r^*}^* | \pi_{\text{mix}}^{k,n}} \wedge \frac{\sqrt{\mathcal{C}(\Pi)}}{\alpha} \right) \cdot \sqrt{\frac{1}{|\mathcal{D}^{k,n}|} \log^{c_0} \frac{|\Pi|T}{\delta}}.$$

Combining all the results above, we conclude that

$$J_\beta(\pi_{r^*}^*) - J_\beta(\pi_{\text{Dstl}}) \leq \Delta_{\min} \wedge c_2 Re^{2R} \cdot \left( \text{Cov}^{\pi_{r^*}^* | \pi_{\text{mix}}^{k,n}} \wedge \frac{\sqrt{\mathcal{C}(\Pi)}}{\alpha} \right) \cdot \sqrt{\frac{1}{|\mathcal{D}^{k,n}|} \log^{c_0} \frac{|\Pi|T}{\delta}}$$

$$= \Delta_{\min} \wedge \iota^{k,n}.$$

Here for simplicity, we use

$$\iota^{k,n} := c_2 Re^{2R} \cdot \left( \text{Cov}^{\pi_{r^*}^* | \pi_{\text{mix}}^{k,n}} \wedge \frac{\sqrt{\mathcal{C}(\Pi)}}{\alpha} \right) \cdot \sqrt{\frac{1}{|\mathcal{D}^{k,n}|} \log^{c_0} \frac{|\Pi|T}{\delta}}$$

as a short note, indexed by $k, n$. Therefore,

$$\sum_{(k,n) \in \mathcal{I}_{\widetilde{K}, \widetilde{N}}^{\text{Trf}}} \mathbb{I}[\mathcal{E}_{\text{Dstl}}^{k,n}](J_\beta(\pi_{r^*}^*) - J_\beta(\pi^{k,n})) = \sum_{(k,n) \in \mathcal{I}_{\widetilde{K}, \widetilde{N}}^{\text{Trf}}} \mathbb{I}[\mathcal{E}_{\text{Dstl}}^{k,n}](\Delta_{\min} \wedge \iota^{k,n}). \tag{20}$$

**Part-(2) Upper Bound the Second Part in Eq. (19)** Next, we bound the accumulative error when $\mathbb{I}[\neg\mathcal{E}_{\text{Dstl}}^{k,n}] = 1$. Note that,

$$\sum_{(k,n) \in \mathcal{I}_{\widetilde{K}, \widetilde{N}}^{\text{Trf}}} \mathbb{I}[\neg\mathcal{E}_{\text{Dstl}}^{k,n}](J_\beta(\pi_{r^*}^*) - J_\beta(\pi^{k,n})) = \sum_{(k,n) \in \mathcal{I}_{\widetilde{K}, \widetilde{N}}^{\text{Trf}}} \sum_{\substack{w \in [W] \\ \Delta(w) > 0}} \mathbb{I}[\mathcal{E}_w^{n,k}]\Delta(w)$$

Here we only focus on those source tasks with $\Delta(w) > 0$, since transferring from $\pi^*_{rw}$ with $\Delta(w) = 0$ does not incur regret. We separate source tasks into two sets $\mathcal{W}_{\leq 2\Delta_{\min}} := \{w \in [W] | \Delta(w) \leq 2\Delta_{\min}\}$ and $\mathcal{W}_{>2\Delta_{\min}} := \{w \in [W] | \Delta(w) > 2\Delta_{\min}\}$. For $w \in \mathcal{W}_{>2\Delta_{\min}}$, on the same good events in Lem. 4.2 and Lem. 4.3, $\mathbb{I}[\mathcal{E}^{n,k}_w] = 1$ implies

$$
\begin{aligned}
\Delta_{\min} =& J_\beta(\pi^*_{r^*}) - J_\beta(\pi_{\text{ref}}) - J_\beta(\pi^*_{r^{w*}}) + J_\beta(\pi_{\text{ref}}) \\
\geq & J_\beta(\pi^*_{r^*}) - J_\beta(\pi_{\text{ref}}) - \widehat{V}^{k,n}(\pi^*_{r^{w*}}; \mathcal{D}^{k,n-1}) \\
\geq & J_\beta(\pi^*_{r^*}) - J_\beta(\pi_{\text{ref}}) - \widehat{V}^{k,n}(\pi^*_{r^w}; \mathcal{D}^{k,n-1}) \\
\geq & J_\beta(\pi^*_{r^*}) - J_\beta(\pi_{\text{ref}}) - J_\beta(\pi^*_{r^w}) + J_\beta(\pi_{\text{ref}}) - 32 \cdot e^{2R} \sqrt{\frac{1}{N(w; \mathcal{D}^{k,n})} \log \frac{|\Pi| WT}{\delta}} \\
= & \Delta(w) - 32 \cdot e^{2R} \sqrt{\frac{1}{N(w; \mathcal{D}^{k,n})} \log \frac{|\Pi| WT}{\delta}}.
\end{aligned}
$$

In the following, we use $c_1 = 32$ as a short note, then the above implies

$$
N(w; \mathcal{D}^{k,n}) \leq \frac{c_1^2 e^{4R}}{(\Delta(w) - \Delta_{\min})^2} \log \frac{|\Pi| WT}{\delta} \leq \frac{4c_1^2 e^{4R}}{\Delta(w)^2} \log \frac{|\Pi| WT}{\delta}
$$

and therefore,

$$
\forall w \in \mathcal{W}_{>2\Delta_{\min}}, \quad \sum_{(k,n) \in \mathcal{I}^{\text{Trf}}_{\widetilde{K}, \widetilde{N}}} \mathbb{I}[\mathcal{E}^{n,k}_w] \Delta(w) \leq \frac{16 c_1^2 e^{4R}}{\Delta(w)} \log \frac{|\Pi| WT}{\delta},
$$

For $w \in \mathcal{W}_{\leq 2\Delta_{\min}}$, we introduce a new event $\mathcal{E}^{n,k}_{2\iota < \Delta_{\min}} := \{2\iota^{k,n} \leq \Delta_{\min}\}$. Note that, when $\mathbb{I}[\mathcal{E}^{n,k}_{2\iota < \Delta_{\min}}] = 0$, i.e. $2\iota \geq \Delta_{\min}$, we automatically have:

$$
\forall w \in \mathcal{W}_{\leq 2\Delta_{\min}}, \quad \mathbb{I}[\mathcal{E}^{n,k}_w] \Delta(w) \leq 2\mathbb{I}[\mathcal{E}^{n,k}_w] \Delta_{\min} \leq 4\mathbb{I}[\mathcal{E}^{n,k}_w] \cdot (\Delta_{\min} \wedge \iota^{n,k}). \tag{21}
$$

On the other hand, on the good events of Lem. 4.2 and Lem. 4.3, when $\mathbb{I}[\mathcal{E}^{n,k}_w \cap \mathcal{E}^{n,k}_{2\iota < \Delta_{\min}}] = 1$, we must have:

$$
\begin{aligned}
J_\beta(\pi^*_{r^*}) - J_\beta(\pi_{\text{ref}}) - \iota^{k,n} \leq & \widehat{V}^{k,n}(\pi^{k,n}_{\text{Dstl}}; \mathcal{D}^{k,n-1}) && \text{(Lem. 4.2 and Lem. 4.3)} \\
\leq & \widehat{V}^{k,n}(\pi^*_{r^w}; \mathcal{D}^{k,n-1}) && (w \text{ is chosen}) \\
\leq & J_\beta(\pi^*_{r^w}) - J_\beta(\pi_{\text{ref}}) + 32 \cdot e^{2R} \sqrt{\frac{1}{N(w; \mathcal{D}^{k,n})} \log \frac{|\Pi| WT}{\delta}},
\end{aligned}
$$

which implies,

$$
N(w; \mathcal{D}^{k,n}) \leq \frac{c_1^2 e^{4R}}{(\Delta_{\min} - \iota^{k,n})^2} \log \frac{|\Pi| WT}{\delta} \leq \frac{4c_1^2 e^{4R}}{\Delta_{\min}^2} \log \frac{|\Pi| WT}{\delta}.
$$

Therefore,

$$
\forall w \in \mathcal{W}_{\leq 2\Delta_{\min}}, \quad \sum_{(k,n) \in \mathcal{I}^{\text{Trf}}_{\widetilde{K}, \widetilde{N}}} \mathbb{I}[\mathcal{E}^{n,k}_w \cap \mathcal{E}^{n,k}_{2\iota < \Delta_{\min}}] \Delta(w) \leq \frac{8 c_1^2 e^{4R}}{\Delta(w)} \log \frac{|\Pi| WT}{\delta}.
$$

Combining with Eq. (21), we have:

$$
\forall w \in \mathcal{W}_{\leq 2\Delta_{\min}}, \quad \sum_{(k,n) \in \mathcal{I}^{\text{Trf}}_{\widetilde{K}, \widetilde{N}}} \mathbb{I}[\mathcal{E}^{n,k}_w] \Delta(w) \leq 4 \sum_{(k,n) \in \mathcal{I}^{\text{Trf}}_{\widetilde{K}, \widetilde{N}}} \mathbb{I}[\mathcal{E}^{n,k}_w] \cdot (\Delta_{\min} \wedge \iota^{n,k}) + \frac{8 c_1^2 e^{4R}}{\Delta(w)} \log \frac{|\Pi| WT}{\delta}.
$$

By merging the analysis for $w \in \mathcal{W}_{\leq 2\Delta_{\min}}$ and $w \in \mathcal{W}_{>2\Delta_{\min}}$, we have:

$$
\forall w \in [W], \quad \sum_{(k,n) \in \mathcal{I}^{\text{Trf}}_{\widetilde{K}, \widetilde{N}}} \mathbb{I}[\mathcal{E}^{n,k}_w] \Delta(w) \leq \frac{16 c_1^2 e^{4R}}{\Delta(w)} \log \frac{|\Pi| WT}{\delta} + 4 \sum_{(k,n) \in \mathcal{I}^{\text{Trf}}_{\widetilde{K}, \widetilde{N}}} \mathbb{I}[\mathcal{E}^{n,k}_w] \Delta_{\min} \wedge \iota^{n,k}, \tag{22}
$$

Note that for those $\Delta(w) \leq 4c_1 e^{2\max} \cdot \sqrt{\frac{1}{\sum_{(k,n)\in\mathcal{I}^{\mathrm{Trf}}_{\widetilde{K},\widetilde{N}}} \mathbb{I}[\mathcal{E}^{n,k}_w]} \log \frac{|\Pi|WT}{\delta}}$, we automatically have

$$
\sum_{(k,n)\in\mathcal{I}^{\mathrm{Trf}}_{\widetilde{K},\widetilde{N}}} \mathbb{I}[\mathcal{E}^{n,k}_w]\Delta(w) \leq 4c_1 e^{2\max} \cdot \sqrt{\frac{1}{\sum_{(k,n)\in\mathcal{I}^{\mathrm{Trf}}_{\widetilde{K},\widetilde{N}}} \mathbb{I}[\mathcal{E}^{n,k}_w]} \log \frac{|\Pi|WT}{\delta} \sum_{(k,n)\in\mathcal{I}^{\mathrm{Trf}}_{\widetilde{K},\widetilde{N}}} \mathbb{I}[\mathcal{E}^{n,k}_w]}
$$

$$
= 4c_1 e^{2\max} \cdot \sqrt{\sum_{(k,n)\in\mathcal{I}^{\mathrm{Trf}}_{\widetilde{K},\widetilde{N}}} \mathbb{I}[\mathcal{E}^{n,k}_w] \log \frac{|\Pi|WT}{\delta}}.
$$

On the other hand, when $\Delta(w) > 4c_1 e^{2\max} \cdot \sqrt{\frac{1}{\sum_{(k,n)\in\mathcal{I}^{\mathrm{Trf}}_{\widetilde{K},\widetilde{N}}} \mathbb{I}[\mathcal{E}^{n,k}_w]} \log \frac{|\Pi|WT}{\delta}}$, the bound in Eq. (22) is tighter, since

$$
\frac{16c_1^2 e^{4R}}{\Delta(w)} \log \frac{|\Pi|WT}{\delta} \leq 4c_1 e^{2\max} \sqrt{\sum_{(k,n)\in\mathcal{I}^{\mathrm{Trf}}_{\widetilde{K},\widetilde{N}}} \mathbb{I}[\mathcal{E}^{n,k}_w] \log \frac{|\Pi|WT}{\delta}}.
$$

Combining the above discussions,

$$
\sum_{(k,n)\in\mathcal{I}^{\mathrm{Trf}}_{\widetilde{K},\widetilde{N}}} \mathbb{I}[\neg\mathcal{E}^{k,n}_{\mathrm{Dstl}}](J_\beta(\pi^*_{r^*}) - J_\beta(\pi^{k,n})) = \sum_{(k,n)\in\mathcal{I}^{\mathrm{Trf}}_{\widetilde{K},\widetilde{N}}} \sum_{\substack{w\in[W]\\\Delta(w)>0}} \mathbb{I}[\mathcal{E}^{n,k}_w]\Delta(w)
$$

$$
\leq \sum_{\substack{w\in[W]\\\Delta(w)>0}} \min\{\frac{16c_1^2 e^{4R}}{\Delta(w)} \log \frac{|\Pi|WT}{\delta}, 4c_1 e^{2\max} \sqrt{\sum_{(k,n)\in\mathcal{I}^{\mathrm{Trf}}_{\widetilde{K},\widetilde{N}}} \mathbb{I}[\mathcal{E}^{n,k}_w] \log \frac{|\Pi|WT}{\delta}}\}
$$

$$
+ 4 \sum_{(k,n)\in\mathcal{I}^{\mathrm{Trf}}_{\widetilde{K},\widetilde{N}}} \mathbb{I}[\mathcal{E}^{n,k}_w]\Delta_{\min} \wedge \iota^{n,k}
$$

$$
\leq \min\{\sum_{\substack{w\in[W]\\\Delta(w)>0}} \frac{16c_1^2 e^{4R}}{\Delta(w)} \log \frac{|\Pi|WT}{\delta}, \sum_{\substack{w\in[W]\\\Delta(w)>0}} 4c_1 e^{2\max} \sqrt{\sum_{(k,n)\in\mathcal{I}^{\mathrm{Trf}}_{\widetilde{K},\widetilde{N}}} \mathbb{I}[\mathcal{E}^{n,k}_w] \log \frac{|\Pi|WT}{\delta}}\}
$$

$$
(\min\{a,b\} + \min\{x,y\} \leq \min\{a+x, b+y\})
$$

$$
+ 4 \sum_{(k,n)\in\mathcal{I}^{\mathrm{Trf}}_{\widetilde{K},\widetilde{N}}} \mathbb{I}[\mathcal{E}^{n,k}_w]\Delta_{\min} \wedge \iota^{n,k}
$$

$$
\leq \min\{\sum_{\substack{w\in[W]\\\Delta(w)>0}} \frac{16c_1^2 e^{4R}}{\Delta(w)} \log \frac{|\Pi|WT}{\delta}, 4c_1 e^{2\max} \sqrt{W|\mathcal{I}^{\mathrm{Trf}}_{\widetilde{K},\widetilde{N}}| \log \frac{|\Pi|WT}{\delta}}\}
$$

$$
(\text{Cauchy-Schwarz inequality and } \sum_{\substack{w\in[W]\\\Delta(w)>0}} \mathbb{I}[\mathcal{E}^{n,k}_w] \leq |\mathcal{I}^{\mathrm{Trf}}_{\widetilde{K},\widetilde{N}}|)
$$

$$
+ 4 \sum_{(k,n)\in\mathcal{I}^{\mathrm{Trf}}_{\widetilde{K},\widetilde{N}}} \mathbb{I}[\mathcal{E}^{n,k}_w]\Delta_{\min} \wedge \iota^{n,k}. \tag{23}
$$

**Merge Everything Together**  Combining Eq. (20) and Eq. (23), we have:

$$
\sum_{(k,n)\in\mathcal{I}^{\mathrm{Trf}}_{\widetilde{K},\widetilde{N}}} J_\beta(\pi^*_{r^*}) - J_\beta(\pi^{k,n})
$$

$$
= \sum_{(k,n)\in\mathcal{I}^{\mathrm{Trf}}_{\widetilde{K},\widetilde{N}}} \mathbb{I}[\mathcal{E}^{k,n}_{\mathrm{Dstl}}](J_\beta(\pi^*_{r^*}) - J_\beta(\pi^{k,n})) + \sum_{(k,n)\in\mathcal{I}^{\mathrm{Trf}}_{\widetilde{K},\widetilde{N}}} \mathbb{I}[\neg\mathcal{E}^{k,n}_{\mathrm{Dstl}}](J_\beta(\pi^*_{r^*}) - J_\beta(\pi^{k,n}))
$$

$$
\leq \min\{\sum_{\substack{w\in[W]\\\Delta(w)>0}} \frac{16c_1^2 e^{4R}}{\Delta(w)} \log \frac{|\Pi|WT}{\delta}, 4c_1 e^{2\max} \sqrt{W|\mathcal{I}^{\mathrm{Trf}}_{\widetilde{K},\widetilde{N}}| \log \frac{|\Pi|WT}{\delta}}\}
$$

$$+ 4 \sum_{(k,n) \in \mathcal{I}_{\widetilde{K}, \widetilde{N}}^{\mathrm{Trf}}} \Delta_{\min} \wedge \iota^{n,k}.$$

Combining the value gap for the online parts, we have:

$$\sum_{(k,n) \in \mathcal{I}_{\widetilde{K}, \widetilde{N}}} J_\beta(\pi_{r^*}^*) - J_\beta(\pi^{k,n})$$

$$\leq C_{\mathrm{OL}} R e^{2R} \sqrt{\mathcal{C}(\Pi) |\mathcal{I}_{\widetilde{K}, \widetilde{N}}^{\mathrm{OL}}| \log^{c_0} \frac{|\Pi| \alpha T}{\delta}} + 4 \sum_{(k,n) \in \mathcal{I}_{\widetilde{K}, \widetilde{N}}^{\mathrm{Trf}}} \Delta_{\min} \wedge \iota^{n,k}$$

$$+ \min \{ \sum_{\substack{w \in [W] \\ \Delta(w) > 0}} \frac{16 c_1^2 e^{4R}}{\Delta(w)} \log \frac{|\Pi| W T}{\delta}, 4 c_1 e^{2\max} \sqrt{W |\mathcal{I}_{\widetilde{K}, \widetilde{N}}^{\mathrm{Trf}}| \log \frac{|\Pi| W T}{\delta}} \}.$$

Note that for $\widetilde{K} > 1$, denote $t_{\widetilde{K}, \widetilde{N}} := (\widetilde{K} - 1) N + \widetilde{N}$, we have:

$$|\mathcal{I}_{\widetilde{K}, \widetilde{N}}^{\mathrm{OL}}| \leq \alpha \widetilde{K} N \leq 2 \alpha t_{\widetilde{K}, \widetilde{N}}, \quad |\mathcal{I}_{\widetilde{K}, \widetilde{N}}^{\mathrm{Trf}}| \leq (1 - \alpha) \widetilde{K} N \leq 2(1 - \alpha) t_{\widetilde{K}, \widetilde{N}}.$$

Therefore,

$$\sum_{(k,n) \in \mathcal{I}_{\widetilde{K}, \widetilde{N}}} J_\beta(\pi_{r^*}^*) - J_\beta(\pi^{k,n})$$

$$= \widetilde{O} \Big( \sum_{(k,n) \in \mathcal{I}_{\widetilde{K}, \widetilde{N}}^{\mathrm{Trf}}} \Delta_{\min} \wedge \iota^{n,k} + R e^{2R} \sqrt{\alpha \mathcal{C}(\Pi) t_{\widetilde{K}, \widetilde{N}}} + e^{2\max} \sqrt{(1 - \alpha) W t_{\widetilde{K}, \widetilde{N}}} \wedge \sum_{\substack{w \in [W] \\ \Delta(w) > 0}} \frac{e^{4R}}{\Delta(w)} \Big),$$

where in the last step, we omit the constant and logarithmic terms. By replacing $t_{\widetilde{K}, \widetilde{N}} \leftarrow t$, $\sum_{(k,n) \in \mathcal{I}_{\widetilde{K}, \widetilde{N}}} \leftarrow \sum_{\tau \leq t}$, $k \leftarrow k(\tau) := \lceil \frac{\tau}{N} \rceil$ and $n \leftarrow n(\tau) := \tau \% N$, we finish the proof.

$\square$

*Remark* F.3. Under the choices of $\mathtt{Alg}_{\mathrm{OL}} \leftarrow$ XPO (Xie et al., 2024) and $\alpha = e^{-\frac{R}{\beta}}$, according to Eq. (9), TPO achieves $\widetilde{O}(W\sqrt{T})$ regret if $T < \frac{W^2}{\Delta_{\min}^2}$. On the other hand, by Lem. 3.1,

$$\mathtt{Cov}^{\pi_{r^*}^* | \pi_{\mathrm{mix}}^t} = 1 + \widetilde{O}(\kappa(e^{\frac{2R}{\beta}}) \cdot \frac{\mathrm{Reg}_{\mathrm{OL}}^{(t)} + \mathrm{Reg}_{\mathrm{Trf}}^{(t)}}{\beta t}) = 1 + \widetilde{O}(\kappa(e^{\frac{2R}{\beta}}) \cdot \frac{1}{\alpha \beta} \sqrt{\frac{\mathcal{C}(\Pi)}{t}}).$$

Therefore, consider $T_0 = \widetilde{\Theta}(\frac{\kappa^2 (e^{\frac{2R}{\beta}} \mathcal{C}(\Pi)}{\alpha^2 \beta^2})$, where $\widetilde{\Theta}(\cdot)$ hides at most poly-log of $T$. At least when $T > T_0$, Eq. (10) implies $\mathrm{Reg}_{\mathrm{Trf}}^{(t)} = 1 + O(1) < C$ for some constant $C$. Since $\alpha = e^{-\frac{R}{\beta}}$, we also have $\mathrm{Reg}_{\mathrm{OL}}^{(T)} = \widetilde{O}(\sqrt{T})$ for any $T > 0$. As a result, the total regret of TPO grows as $\widetilde{O}(\sqrt{T})$ as long as $T > T_0$.

## G. Connection between Win Rate and Policy Coverage Coefficient

**Lemma G.1.** *Given two probability vector $u, v \in \Delta(\mathcal{A})$ and a reward function $r : \mathcal{A} \to \mathbb{R}$, consider a preference model based on $r$, satisfying,*

$$\mathbb{P}_r(y = 1 | a, a') \geq \frac{1}{2},$$

*for any $a, a' \in \mathcal{A}$ satisfying $r(a) \geq r(a')$. Then,*

$$\sum_a \sqrt{u(a) v(a)} \leq \min_{\gamma > 0} \sqrt{(\gamma + 2\mathbb{P}_r(v \succ u)) \log \frac{1 + \gamma}{\gamma}},$$

*where $\mathbb{P}_r(u \succ v) := \mathbb{E}_{a \sim u, a' \sim v}[\mathbb{P}_r(y = 1 | a, a')]$.*

*Proof.* We first of all sort the action space $\mathcal{A}$ to $\mathcal{A}_{\text{sorted}} := \{a_1, a_2, ..., a_{|\mathcal{A}|}\}$ according to reward function $r$, such that, for any $1 \leq i < j \leq |\mathcal{A}|$, $r(a_i) \leq r(a_j)$. Besides, we use $F^u(\cdot)$ to denote the cumulative distribution function regarding $u$:

$$\forall 1 \leq i \leq |\mathcal{A}|, \quad F^u(a_i) := \sum_{j=1}^{i} u(a_i),$$

and $F^v$ is defined similarly. Then we have:

$$\sum_{a \in \mathcal{A}} \sqrt{u(a)v(a)} = \sum_{i=1}^{|\mathcal{A}|} \sqrt{u(a_i)v(a_i)} = \sum_{i=1}^{|\mathcal{A}|} \sqrt{\frac{u(a_i)}{\gamma + F^u(a_i)}} \cdot \sqrt{(\gamma + F^u(a_i))v(a)} \quad \text{(Introducing a parameter } \gamma > 0\text{)}$$

$$\leq \sqrt{\sum_{i=1}^{|\mathcal{A}|} \frac{u(a_i)}{\gamma + F^u(a_i)}} \cdot \sqrt{\sum_{i=1}^{|\mathcal{A}|} (\gamma + F^u(a_i))v(a_i)} \quad \text{(Cauchy–Schwarz inequality)}$$

$$\leq \sqrt{\sum_{i=1}^{|\mathcal{A}|} \frac{u(a_i)}{\gamma + F^u(a_i)}} \cdot \sqrt{\gamma + 2\mathbb{P}_r(v \succ u)},$$

where in the last step, we use the fact that $\gamma \sum_{i=1}^{|\mathcal{A}|} v(a_i) = \gamma$ and

$$\mathbb{P}_r(v \succ u) = \mathbb{E}_{a \sim v, a' \sim u}[\mathbb{P}_r(y = 1|a, a')] \geq \sum_{i=1}^{|\mathcal{A}|} v(a_i) \sum_{j=1}^{i} u(a_j)\mathbb{P}_r(y = 1|a_i, a_j) \geq \frac{1}{2} \sum_{i=1}^{|\mathcal{A}|} v(a_i)F^u(a_i).$$

For the first part, we can upper bound by the following:

$$\sum_{i=1}^{|\mathcal{A}|} \frac{u(a_i)}{\gamma + F^u(a_i)} = \sum_{i=1}^{|\mathcal{A}|} \frac{u(a_i)}{\gamma + \sum_{j=1}^{i} u(a_j)} = \sum_{i=1}^{A} 1 - \frac{\gamma + \sum_{j=1}^{i-1} u(a_j)}{\gamma + \sum_{j=1}^{i} u(a_j)}$$

$$\leq \sum_{i=1}^{A} \log \frac{\gamma + \sum_{j=1}^{i} u(a_j)}{\gamma + \sum_{j=1}^{i-1} u(a_j)} \quad (1 - x \leq \log \frac{1}{x})$$

$$\leq \log \frac{1 + \gamma}{\gamma}$$

Therefore, $\sum_{a \in \mathcal{A}} \sqrt{u(a)v(a)} \leq \sqrt{(\gamma + 2\mathbb{P}_r(v \succ u)) \log \frac{1+\gamma}{\gamma}}$. Since $\gamma$ is arbitrary, we can take the minimum over $\gamma > 0$, and finish the proof. $\qquad \square$

**Lemma G.2.** *For any policy $\pi, \widetilde{\pi}$,*

$$1 - \mathbb{TV}(\pi(\cdot|s)\|\widetilde{\pi}(\cdot|s)) \leq \min_{\gamma > 0} \sqrt{(\gamma + 2\mathbb{P}_{r^*}(\pi(\cdot|s) \succ \widetilde{\pi}(\cdot|s))) \log \frac{1+\gamma}{\gamma}},$$

$$1 - \mathbb{E}_{s \sim \rho}[\mathbb{TV}(\pi(\cdot|s)\|\widetilde{\pi}(\cdot|s))] \leq \min_{\gamma > 0} \sqrt{(\gamma + 2\mathbb{P}_{r^*}(\pi \succ \widetilde{\pi})) \log \frac{1+\gamma}{\gamma}}.$$

*Proof.*

$$1 - \mathbb{TV}(\pi(\cdot|s)\|\widetilde{\pi}(\cdot|s)) \leq 1 - \mathbb{H}^2(\pi(\cdot|s)\|\widetilde{\pi}(\cdot|s)) = \sum_{a \in \mathcal{A}} \sqrt{\pi(a|s)\widetilde{\pi}(a|s)}$$

$$\leq \min_{\gamma > 0} \sqrt{\left(\gamma + 2\mathbb{P}_{r^*}(\pi(\cdot|s) \succ \widetilde{\pi}(\cdot|s))\right) \log \frac{1+\gamma}{\gamma}}.$$

where in the last step we apply Lem. G.1 with $\pi(\cdot|s)$ as $v(\cdot)$, $\widetilde{\pi}(\cdot|s)$ as $u(\cdot)$, $r^*(s, \cdot)$ as the reward function. Then, we finish the proof for the first inequality.

For the second inequality, by taking the expectation over $s \sim \rho$ and the concavity of $\sqrt{\cdot}$ function, we have:

$$1 - \mathbb{E}_{s\sim\rho}[\mathbb{TV}(\pi(\cdot|s)\|\widetilde{\pi}(\cdot|s))] \leq \min_{\gamma > 0} \sqrt{\left(\gamma + 2\mathbb{P}_{r^*}(\pi \succ \widetilde{\pi})\right) \log \frac{1+\gamma}{\gamma}}.$$

By choosing $\gamma = \mathbb{P}_{r^*}(\pi \succ \widetilde{\pi})$ (note that this choice ensures $\gamma > 0$ since $\widetilde{\pi}(\cdot|\cdot) > 0$), we finish the proof. $\qquad\square$

**Lemma G.3.** *[The Complete Version of Lem. 5.1] Given any policy $\pi$, under the assumption that $\mathbb{P}_{r^*}(y = 1|s, a, a') \geq \frac{1}{2}$ for any $a, a' \in \mathcal{A}$ satisfying $r^*(s, a) \geq r^*(s, a')$, we have:*

$$Cov^{\pi_{r^*}^*|\pi} \geq \max_{\gamma > 0, \bar{\pi}} \left( \sqrt{\gamma + 2\mathbb{P}_{r^*}(\bar{\pi} \succ \pi) \log \frac{1+\gamma}{\gamma}} + \sqrt{\frac{J_\beta(\pi_{r^*}^*) - J_\beta(\bar{\pi})}{2\beta}} \right)^{-1} \tag{24}$$

$$Cov^{\pi_{r^*}^*|\pi} \geq \max_{\gamma > 0, \bar{\pi}} \left( \sqrt{\gamma + 2\mathbb{P}_{r^*}(\pi \succ \bar{\pi}) \log \frac{1+\gamma}{\gamma}} + \sqrt{\frac{J_\beta(\pi_{r^*}^*) - J_\beta(\bar{\pi})}{2\beta}} \right)^{-1}. \tag{25}$$

*where $\bar{\pi}$ is an arbitrary intermediate policy, $\mathbb{P}_{r^*}(\pi \succ \widetilde{\pi}) := \mathbb{E}_{s\sim\rho, a\sim\pi, a'\sim\widetilde{\pi}}[\mathbb{P}_{r^*}(y = 1|s, a, a')]$ and $\mathbb{P}_{r^*}(\widetilde{\pi} \succ \pi) = 1 - \mathbb{P}_{r^*}(\pi \succ \widetilde{\pi}) = \mathbb{E}_{s\sim\rho, a\sim\widetilde{\pi}, a'\sim\pi}[\mathbb{P}_{r^*}(y = 1|s, a, a')].$*

*Proof.* We have:

$$\mathbb{E}_{a\sim\pi_{r^*}^*(\cdot|s)}\left[\frac{\pi_{r^*}^*(a|s)}{\pi(a|s)}\right] - 1 = \chi^2(\pi_{r^*}^*(\cdot|s)\|\pi(\cdot|s))$$

$$\geq \exp(\mathrm{KL}(\pi_{r^*}^*(\cdot|s)\|\pi(\cdot|s))) - 1 \qquad \text{(Theorem 5 in (Gibbs \& Su, 2002))}$$

$$\geq \frac{1}{2} \cdot \frac{1}{1 - \mathbb{TV}(\pi_{r^*}^*(\cdot|s)\|\pi(\cdot|s))} - 1. \qquad \text{(Bretagnolle–Huber inequality)}$$

Now, we introduce an arbitrary intermediate policy $\widetilde{\pi}$,

$$\mathbb{TV}(\pi_{r^*}^*(\cdot|s)\|\pi(\cdot|s)) \geq \mathbb{TV}(\bar{\pi}(\cdot|s)\|\pi(\cdot|s)) - \mathbb{TV}(\bar{\pi}(\cdot|s)\|\pi_{r^*}^*(\cdot|s)) \qquad \text{(Reverse triangle inequality)}$$

$$\geq \mathbb{TV}(\bar{\pi}(\cdot|s)\|\pi(\cdot|s)) - \sqrt{\frac{1}{2}\mathrm{KL}(\bar{\pi}(\cdot|s)\|\pi_{r^*}^*(\cdot|s))}. \qquad \text{(Pinsker's inequality)}$$

Applying Lem. G.2 with $(\pi, \widetilde{\pi}) \leftarrow (\pi, \pi)$, we have:

$$\mathbb{E}_{a\sim\pi_{r^*}^*(\cdot|s)}\left[\frac{\pi_{r^*}^*(a|s)}{\pi(a|s)}\right] \geq \frac{1}{1 - \mathbb{TV}(\bar{\pi}(\cdot|s)\|\pi(\cdot|s)) + \sqrt{\frac{1}{2}\mathrm{KL}(\bar{\pi}(\cdot|s)\|\pi_{r^*}^*(\cdot|s))}}$$

$$\geq \frac{1}{\sqrt{(\gamma + 2\mathbb{P}_{r^*}(\bar{\pi}(\cdot|s) \succ \pi(\cdot|s))) \log \frac{1+\gamma}{\gamma}} + \sqrt{\frac{1}{2}\mathrm{KL}(\bar{\pi}(\cdot|s)\|\pi_{r^*}^*(\cdot|s))}}.$$

By taking the expectation over $s \sim \rho$, and leveraging the convexity of $1/x$ and the concavity of $\sqrt{\cdot}$ functions, we have:

$$\mathbb{E}_{s\sim\rho, a\sim\pi_{r^*}^*(\cdot|s)}\left[\frac{\pi_{r^*}^*(a|s)}{\pi(a|s)}\right] \geq \frac{1}{\mathbb{E}_{s\sim\rho}\left[\sqrt{(\gamma + 2\mathbb{P}_{r^*}(\bar{\pi}(\cdot|s) \succ \pi(\cdot|s))) \log \frac{1+\gamma}{\gamma}}\right] + \mathbb{E}_{s\sim\rho}\left[\sqrt{\frac{1}{2}\mathrm{KL}(\bar{\pi}(\cdot|s)\|\pi_{r^*}^*(\cdot|s))}\right]}$$

$$\geq \frac{1}{\sqrt{(\gamma + 2\mathbb{P}_{r^*}(\bar{\pi} \succ \pi)) \log \frac{1+\gamma}{\gamma}} + \sqrt{\frac{J_\beta(\pi_{r^*}^*) - J_\beta(\pi)}{2\beta}}}.$$

Note that the above results hold for any $\gamma > 0$ and any $\bar{\pi}$, we finish the proof by taking the maximum over them.

The second inequality in Lem. G.3 can be proved similarly by applying Lem. G.2 with $(\pi, \widetilde{\pi}) \leftarrow (\pi, \bar{\pi})$. All the discussion are the same. $\qquad\square$

*Remark* G.4. We provide some remarks about Lem. G.3.

- Lem. 5.1 is a direct corollary of Eq. (25).

- Notably, Eq. (24) has a different implication compared with Eq. (25) that we follow in the algorithm design in Sec. 5. More concretely, Eq. (24) suggests we should also disregard those source policies that strongly dominate $\bar{\pi}$ when $\bar{\pi}$ is close to $\pi_{r^*}^*$. This makes sense because those source policies may achieve high rewards or win rates by incurring a high KL divergence with $\pi_{\text{ref}}$, and therefore, they may not provide good coverage for $\pi_{r^*}^*$.

- However, in Alg. 3, we intentionally do not filter out but instead prioritize source policies with exceptionally high win rates. Because they likely provide good coverage for high-reward regions and can be advantageous for practical LLM training. Nonetheless, for completeness, we bring this theory-practice gap into attention.

# H. Useful Lemmas

**Lemma H.1** (Convex Hull Fulfills Assump. A). *Given $\Pi$ satisfying Assump. A, conv$(\Pi)$ also satisfies Assump. A.*

*Proof.* The realizability condition is obviously. We verify Assump. A. Note that for any $\pi \in \text{conv}(\Pi)$, there exists $\lambda^1, ..., \lambda^n \geq 0$ and $\pi^1, ..., \pi^n \in \Pi$, s.t. $\sum_{i=1}^n \lambda^i = 1$ and $\pi = \sum_{i=1}^n \lambda^i \pi^i$, which implies,

$$\forall s, a \quad \frac{\pi(a|s)}{\pi_{\text{ref}}(a|s)} = \sum_{i=1}^n \lambda^i \frac{\pi^i(a|s)}{\pi_{\text{ref}}(a|s)} \geq \exp(-\frac{R}{\beta}), \qquad \frac{\pi(a|s)}{\pi_{\text{ref}}(a|s)} = \sum_{i=1}^n \lambda^i \frac{\pi^i(a|s)}{\pi_{\text{ref}}(a|s)} \leq \exp(\frac{R}{\beta}).$$

Therefore, conv$(\Pi)$ fulfills Assump. A-(II), which finishes the proof. $\square$

**Lemma H.2** (MLE Guarantees; Adapted from Lemma C.6 in (Xie et al., 2024)). *Consider a policy class $\Pi$ satisfying Assump. A, and recall the reward class $\mathcal{R}^\Pi$ converted by Eq. (11). Given a dataset $\mathcal{D} := \{(s^i, a^i, \widetilde{a}^i, y^i, \pi^i)\}_{i \leq |\mathcal{D}|}$ satisfying Cond. C.1 and any $\delta \in (0, 1)$, w.p. $1 - \delta$,*

$$\forall r \in \mathcal{R}^\Pi, \quad \frac{1}{|\mathcal{D}|} \sum_{i \leq |\mathcal{D}|} \mathbb{E}_{s \sim \rho, a \sim \pi^i(\cdot|s), \widetilde{a} \sim \pi_{ref}(\cdot|s)}[\mathbb{H}^2(\mathbb{P}_r(\cdot|s, a, \widetilde{a}) \| \mathbb{P}_{r^*}(\cdot|s, a, \widetilde{a}))] \leq L_{\mathcal{D}}(r) - L_{\mathcal{D}}(r^*) + \frac{2}{|\mathcal{D}|} \log \frac{|\Pi|}{\delta}.$$

*Proof.* The proof is almost identical to Lemma C.6 in (Xie et al., 2024), except we replace $\mathbb{P}_\pi$ in their paper by $\mathbb{P}_r$. So we omit it here. Besides, note that our NLL loss is normalized, while (Xie et al., 2024) consider unnormalized version. This results in the additional $\frac{1}{|\mathcal{D}|}$ factor here. $\square$

**Lemma H.3** (From reward error to Hellinger Distance). *Given any policy $\pi$, any reward function $r$ with bounded value range $[-R, R]$, and another arbitrary $\widetilde{\pi}$ with positive support on $\mathcal{S} \times \mathcal{A}$, we have:*

$$\mathbb{E}_{s \sim \rho, a \sim \pi(\cdot|s), \widetilde{a} \sim \pi_{ref}(\cdot|s)}[|(r^*(s, a) - r^*(s, \widetilde{a})) - (r(s, a) - r(s, \widetilde{a}))|]$$

$$\leq 8\sqrt{2}e^{2R} \sqrt{\mathcal{C}ov^{\pi|\widetilde{\pi}} \cdot \mathbb{E}_{s \sim \rho, a \sim \widetilde{\pi}(\cdot|s), \widetilde{a} \sim \pi_{ref}(\cdot|s)}[\mathbb{H}^2(\mathbb{P}_r(\cdot|s, a, \widetilde{a}) \| \mathbb{P}_{r^*}(\cdot|s, a, \widetilde{a}))]}.$$

*Proof.* For any reward function $r$, we have:

$$\left| \left( \mathbb{E}_{\rho, \pi}[r^*] - \mathbb{E}_{\rho, \pi_{\text{ref}}}[r^*] \right) - \left( \mathbb{E}_{\rho, \pi}[r] - \mathbb{E}_{\rho, \pi_{\text{ref}}}[r] \right) \right|$$

$$\leq \mathbb{E}_{s \sim \rho, a \sim \pi(\cdot|s), \widetilde{a} \sim \pi_{ref}(\cdot|s)}[|(r^*(s, a) - r^*(s, \widetilde{a})) - (r(s, a) - r(s, \widetilde{a}))|]$$

$$\leq 4e^{2R} \mathbb{E}_{s \sim \rho, a \sim \pi(\cdot|s), \widetilde{a} \sim \pi_{ref}(\cdot|s)}[|\sigma(r^*(s, a) - r^*(s, \widetilde{a})) - \sigma(r(s, a) - r(s, \widetilde{a}))|] \qquad \text{(Lem. H.4 with } C = 2R)$$

$$= 4e^{2R} \sum_{s, a, \widetilde{a}} \sqrt{\rho(s)\pi_{\text{ref}}(\widetilde{a}|s)} \frac{\pi(a|s)}{\sqrt{\widetilde{\pi}(a|s)}}$$

$$\cdot \sqrt{\rho(s)\pi_{\text{ref}}(\widetilde{a}|s)} \sqrt{\widetilde{\pi}(a|s)} \left| \sigma(r^*(s, a) - r^*(s, \widetilde{a})) - \sigma(r(s, a) - r(s, \widetilde{a})) \right|$$

$$\leq 4e^{2R} \sqrt{\sum_{s, a, \widetilde{a}} \rho(s)\pi_{\text{ref}}(\widetilde{a}|s) \frac{\pi^2(a|s)}{\widetilde{\pi}(a|s)}}$$

$$\sqrt{\mathbb{E}_{s\sim\rho,a\sim\widetilde{\pi}(\cdot|s),\widetilde{a}\sim\pi_{\text{ref}}(\cdot|s)}\Big[\Big|\sigma\Big(r^*(s,a)-r^*(s,\widetilde{a})\Big)-\sigma\Big(r(s,a)-r(s,\widetilde{a})\Big)\Big|^2\Big]}$$

(Cauchy–Schwarz inequality)

$$=4e^{2R}\sqrt{\text{Cov}^{\pi|\widetilde{\pi}}}\cdot\sqrt{\mathbb{E}_{s\sim\rho,a\sim\widetilde{\pi}(\cdot|s),\widetilde{a}\sim\pi_{\text{ref}}(\cdot|s)}\Big[\Big|\sigma\Big(r^*(s,a)-r^*(s,\widetilde{a})\Big)-\sigma\Big(r(s,a)-r(s,\widetilde{a})\Big)\Big|^2\Big]}. \tag{26}$$

Note that,

$$\mathbb{E}_{s\sim\rho,a\sim\widetilde{\pi}(\cdot|s),\widetilde{a}\sim\pi_{\text{ref}}(\cdot|s)}\Big[\Big|\sigma\Big(r^*(s,a)-r^*(s,\widetilde{a})\Big)-\sigma\Big(r(s,a)-r(s,\widetilde{a})\Big)\Big|^2\Big]$$

$$\leq 8\mathbb{E}_{s\sim\rho,a\sim\widetilde{\pi}(\cdot|s),\widetilde{a}\sim\pi_{\text{ref}}(\cdot|s)}\Big[\Big|\sqrt{\sigma(r^*(s,a)-r^*(s,\widetilde{a}))}-\sqrt{\sigma(r(s,a)-r(s,\widetilde{a}))}\Big|^2\Big] \quad ((x-y)^2\leq 4(x+y)(\sqrt{x}-\sqrt{y})^2)$$

$$\leq 8\mathbb{E}_{s\sim\rho,a\sim\widetilde{\pi}(\cdot|s),\widetilde{a}\sim\pi_{\text{ref}}(\cdot|s)}\Big[\mathbb{H}^2(\mathbb{P}_r(\cdot|s,a,\widetilde{a})\|\mathbb{P}_{r^*}(\cdot|s,a,\widetilde{a}))\Big]. \tag{27}$$

By plugging into Eq. (26), we finish the proof. $\square$

**Lemma H.4** (Sigmoid Function). *Given* $x,y\in[-C,C]$ *for some* $C>0$,

$$|x-y|\leq 4\exp(C)|\sigma(x)-\sigma(y)|.$$

*Proof.* Without loss of generality, we assume $x\geq y$. Because $\sigma(\cdot)$ is a monotonically increasing function and it is continuous, we know there exists $z\in[y,x]$ s.t.

$$\frac{\sigma(x)-\sigma(y)}{x-y}=\sigma'(z)=\sigma(z)(1-\sigma(z))=\sigma(z)\sigma(-z).$$

Because $x,y\in[-C,C]$, we have $\sigma(-C)\leq\sigma(z)\leq\sigma(C)$. Note that the axis of symmetry of function $f(a)=a(1-a)$ is $1/2$ and $\frac{1}{2}-\sigma(C)=\sigma(-C)-\frac{1}{2}$. Therefore,

$$|x-y|=\frac{1}{\sigma'(z)}|\sigma(x)-\sigma(y)|\leq(1+\exp(C))(1+\exp(-C))\cdot|\sigma(x)-\sigma(y)|$$

$$\leq 2(1+\exp(C))|\sigma(x)-\sigma(y)|\leq 4\exp(C)|\sigma(x)-\sigma(y)|$$

$\square$

# I. Experiment Details and Additional Results

## I.1. Details in Experiment Setup

**Setup of** $r^*$  Due to the high cost of collecting real human feedback, we use preferences generated by Llama3-8B (Dubey et al., 2024) to simulate the ground-truth human annotations. More concretely, we adopt `sfairXC/FsfairX-LLaMA3-RM-v0.1` (Dong et al., 2024) as the true reward model $r^*$, which is distilled from `meta-llama/Meta-Llama-3-8B-Instruct`. This reward model can be queried with prompt-response pairs and returns reward scores for each of them.

**Best-of-N as an Approximation of** $\pi_{r^w}^*$  We recall that we consider 4 source tasks for transfer learning, including, (a) ROUGE-Lsum score (Lin, 2004), (b) BERTScore (Zhang et al., 2019), (c) T5-base (250M) $\pi_{\text{base}}$, (d) T5-large (770M) $\pi_{\text{large}}$.

To reduce computational complexity, instead of explicitly training the optimal policies $\{\pi_{r^w}^*\}_{w\in[W]}$ associated with each source reward model, we use Best-of-N (BoN) approach, a.k.a. rejection sampling, to approximate the responses generated by $\pi_{r^w}^*$. Specifically, we generate N[7] responses by the online learning policy $\pi_{\text{OL}}^k$ in Alg. 3, rank them according to the reward model, and select the top-ranked ones.

Furthermore, even for source LLM policies (3) and (4), we find that transferring from BoN-selected responses generated by $\pi_{\text{OL}}^k$ actually outperforms directly using the responses generated by T5-base/large. We hypothesize that it is because the responses by T5-base/large usually have quite low probability of being generated by the online learning policy, leading to a

---

[7]To distinguish with $N$ used to denote the block size in Alg. 3, we use N to denote the size of BoN.

distribution shift that complicates learning. In contrast, BoN-selected responses maintain non-trivial probability of being sampled, without significant distributional mismatch.

Next, we elaborate the BoN process with more details. In our experiment, we choose $N = 32$. For source reward models (1) and (2), we generate $N$ responses, and we compute the ROUGE-Lsum/BERTScore between the generated response and the human-provided summary in XSum dataset as the reward value. For source policies (3) and (4), motivated by the closed-form solution in Eq. (2), given any prompt $s$ and response $a$, we infer the log-probability of T5-base/large predicting $a$ given $s$ as the reward score, i.e. $\log \pi_{\text{base}}(a|s)$ or $\log \pi_{\text{large}}(a|s)$. In another word, we interpret T5-base/large as the optimal policies fine-tuned from uniform distribution to align with some reward models, which we treated as source rewards for transfer learning.

**Training Details**   Our training setup is based on and adapted from (Xie et al., 2024; Xiong et al., 2024). We run for 3 iterations ($K = 3$), and in each iteration, we sample a training dataset of size 10k (i.e., the block size $N =$10k). For each prompt, we collect 8 responses as follows. Firstly, we generate $N + 4$ responses by the online learning policy $\pi_{\text{OL}}^k$. The initial $N$ responses are used for Best-of-N (BoN) selection, where we choose a source reward model via the UCB strategy in Alg. 3, and then pick the top 4 from $N$ responses with the highest source rewards. These 4 responses are merged with the remaining 4 responses and we get a total of 8 responses. After that, we query $r^*$ to label the reward for those responses, and record the ones with the highest and lowest rewards to serve as positive and negative samples for DPO training.

In contrast to the procedure presented in Alg. 3, we utilize 8 responses for each prompt. Therefore, the win rates are computed in a relatively different way. Specially, we set $y^{k,n} = 1$ if the response achieving the highest reward comes from the 4 responses selected by the BoN step, and $y^{k,n} = 0$ otherwise. The updates of the win rates estimation and the computation of UCB bonus terms align with Alg. 3, except that we set $\widehat{\text{WR}}^{\pi_{\text{OL}}^k} = 0.55$ instead of 0.5. This adjustment establishes a higher threshold for enabling transfer learning, requiring source tasks to outperform the baseline policy $\pi_{\text{OL}}^k$ by a larger margin before being considered. We believe it enhances overall performance.

Regarding other hyperparameters during the training, the learning rate is 5e-5 with a cosine annealing schedule. Training is conducted on 4 H-100 GPUs with total batch size 64. We set the constant parts in the UCB bonus $c\sqrt{\log \frac{1}{\delta}} = 1.0$ in practice, considering the value range of win rates is [0, 1].

**Evaluation Details**   During the evaluation phase, we randomly sample 10k prompts from XSum test dataset without repetition. For each prompt, we generate one response for each of the policies being compared, and query their reward values from $r^*$ (i.e., the `sfairXC/FsfairX-LLaMA3-RM-v0.1` reward model). The win rate is then estimated as the frequency of that one policy generates a response with higher rewards than the other across the 10k prompts.

## I.2. Additional Experiment Results

**Results under Other Choices for Alg$_{\text{PO}}$ in Alg. 3**   In the following, we report the results with two alternative instaniations of Alg$_{\text{PO}}$: by optimizing the XPO loss (Xie et al., 2024) or the IPO loss (Azar et al., 2024). All the training setups are the same as the experiments where Alg$_{\text{PO}}$ is DPO, except that we choose a smaller learning rate 1e-5 for Alg$_{\text{PO}}$ is IPO.

*Remark* I.1.  Different from DPO and IPO, XPO is an online algorithm itself and in their original design, the pairs of online data are generated by an online exploration policy and another fixed base policy, respectively. However, empirically, Xie et al. (2024) follow the iterative-DPO and utilize the same online learning policy to generate pairs of online data. This exactly aligns with the no transfer baseline we compete with—instantiating Alg$_{\text{PO}}$ with XPO in Alg. 3 and setting $W = 0$, which we refer as iterative-XPO in this paper.

**Investigation on Source Task Selection**   Fig. 2 provides further investigations on the source task selection process. For each iteration $k = 1, 2, 3$, we count the number of times that $\pi^{k,n}$ is occupied by different transfer policies $\{\pi_{r^w}^*\}_{w \in [W]}$ or the online learning policy $\pi_{\text{OL}}^k$ (i.e. without transfer), and provide the results on the top sub-figure in Fig. 2. Besides, in the bottom sub-figure, we report the win rates $\mathbb{P}_{r^*}(\pi \succ \pi_{\text{OL}}^k)$ for all $\pi \in \{\pi_{r^w}^*\}_{w \in [W]} \cup \{\pi_{\text{OL}}^k\}$. As illustrated, the UCB sub-routine efficiently explores and identify the source task with the highest win rates against the learning policy $\pi_{\text{OL}}^k$.

Notably, as the improvement of $\pi_{\text{OL}}^k$ over the three iterations, we can observe the transition from transfer learning by leveraging high-quality source tasks to standard online learning. In other words, our method can automatically switch back to online learning and avoid being restricted by source reward models.

|         | Without Transfer | Purely Exploit ROUGE-Lsum | Purely Exploit T5-Large |
| ------- | ---------------- | ------------------------- | ----------------------- |
| Iter 1  | $52.3 \pm 1.0$   | $50.4 \pm 1.6$            | $49.9 \pm 0.4$          |
| Iter 2  | $55.2 \pm 1.4$   | $52.3 \pm 0.3$            | $50.1 \pm 0.3$          |
| Iter 3  | $55.3 \pm 1.1$   | $51.8 \pm 0.5$            | $50.3 \pm 0.5$          |

(a) IPO as $\text{Alg}_{\text{PO}}$ in Alg. 3

|         | Without Transfer | Purely Exploit ROUGE-Lsum | Purely Exploit T5-Large |
| ------- | ---------------- | ------------------------- | ----------------------- |
| Iter 1  | $52.3 \pm 1.1$   | $53.4 \pm 0.8$            | $50.2 \pm 0.3$          |
| Iter 2  | $51.6 \pm 1.3$   | $54.7 \pm 1.6$            | $49.1 \pm 1.3$          |
| Iter 3  | $52.2 \pm 1.6$   | $53.8 \pm 2.9$            | $49.2 \pm 1.1$          |

(b) XPO as $\text{Alg}_{\text{PO}}$ in Alg. 3

*Table 2.* Similar to Table 1, we report the win rates (%) of the policies trained by empirical TPO (Alg. 3) competed with 3 baselines, presented across 3 columns. Baseline (I): without transfer, i.e., iterative-IPO or iterative-XPO. Baseline (II): purely utilizing ROUGE-LSum (the lowest-quality source task) in transfer learning. Baseline (III): purely utilizing T5-Large (the highest-quality source task) in transfer learning. Results are averaged with 3 random seeds and 95% confidence levels are reported.

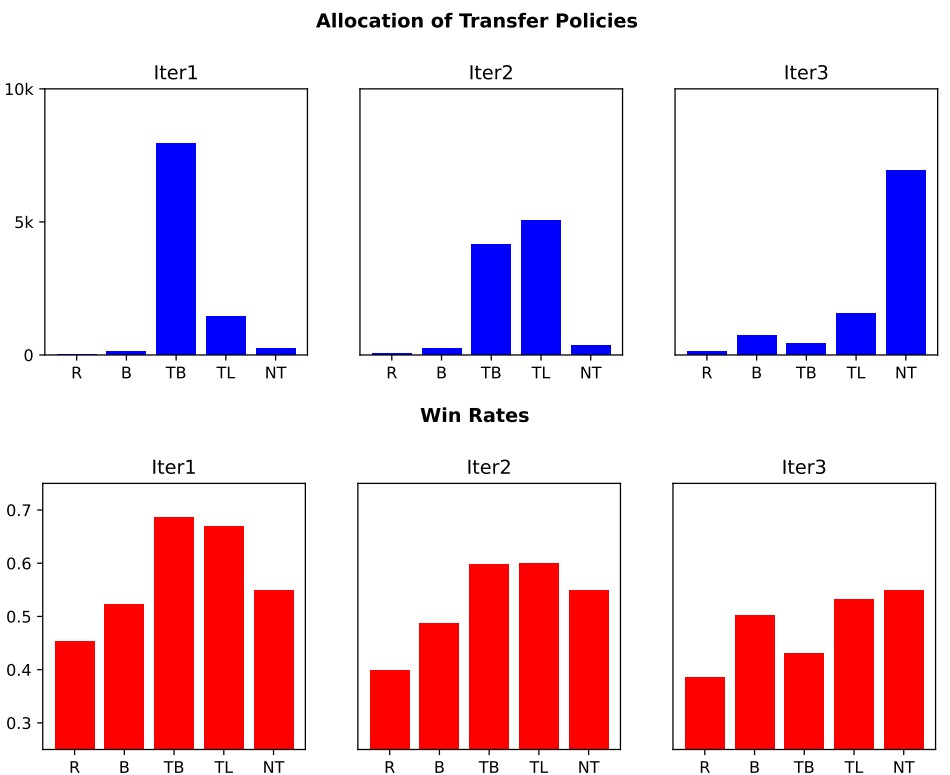

*Figure 2.* Deeper investigation on the source reward models selection process. We report the allocation of transfer budgets on each source tasks averaged over 3 trials (top figure) and the win rates $\mathbb{P}_{r^*}(\cdot \succ \pi_{\text{OL}}^k)$ (bottom figure) for iterations $k = 1, 2, 3$. Due to space limit, we use abbreviation rather than the full name of source tasks. R, B, TB, TL and NT stand for ROUGE-Lsum, BERTScore, T5-Base, T5-Large and No Transfer, respectively.

