# OpenReview forum: "Can RLHF be More Efficient with Imperfect Reward Models? A Policy Coverage Perspective"
_ICML.cc/2025/Conference — ICML 2025 poster_

### Official Review · Reviewer_yCvD · 2025-03-05

**Overall Recommendation:** 4

**Summary:**

This paper investigates reward transfer in the context of online active RLHF. Existing investigations (based on active preference collection using on-policy sampling) have regret bounds proportional to instance-dependent properties, such as the cardinality of the action space. This investigation assumes access to imperfect reward models, and makes a novel connection between the coverage of the optimal policy due to the policies induced by these reward models, and their sub-optimality gaps with respect to the optimal policy. Leveraging this insight, the authors propose a policy selection routine to speed up convergence in the early stages. This subroutine works by forming estimates of the policy value induced by the imperfect reward models, and prescribing the policy which has the largest value. In terms of performance, the proposed algorithm is shown to exhibit sublinear regret which *does not depend on structural complexity measures*.

**Claims And Evidence:**

The claims are adequately substantiated.

**Essential References Not Discussed:**

N/A

**Experimental Designs Or Analyses:**

The experiment section is fairly limited. I have discussed my concerns about the baselines under "Methods And Evaluation Criteria".

**Methods And Evaluation Criteria:**

I am not entirely convinced by the choice of baselines for comparing the proposed algorithm in empirical evaluations.

- Since the paper shows improved regret compared against on-policy algorithms, why not compare against one of them (e.g. XPO)?
- Why only choose summarization task as a baseline? What about other baselines (e.g., the ones considered in [1])?

[1] Ji, K., He, J., & Gu, Q. (2024). Reinforcement learning from human feedback with active queries. arXiv preprint arXiv:2402.09401.

**Other Comments Or Suggestions:**

N/A

**Other Strengths And Weaknesses:**

These have been discussed.

**Questions For Authors:**

I have the following questions for the authors:

1. Since we do not have access to the optimal reward $r^\star$, we may not obtain the value o the imperfect reward models with respect to the optimal policy. To circumvent this, the authors propose to estimate the value gap $J_{\beta}(\pi^\star_{r_w}) - J_{\beta}(\pi_{\rm ref})$. Instead, what if we use the an estimate of the value function $J_{\beta}(\pi^\star_{r_w})$ for policy evaluation? More specifically, form an optimistic estimate of $\widehat r$, compute its value under $\pi^\star_{r_w}$, and select the source model with the largest estimated value. Why do we require to introduce $\pi_{\rm ref}$?

2. In (8), the second term is contributed by the imperfection of the source reward models. However, if the gaps are small (near perfect source models), I wonder why it hurts the regret. Imagine having $\Delta_{\min}$ non-zero, but vanishingly small. In that case, the second term in (8) explodes. Is this a limitation of the analysis?

**Relation To Broader Scientific Literature:**

The contributions of this paper are very relevant to the RLHF community. To the best of my knowledge, reward transfer has not been investigated in the RLHF context. Furthermore, this paper makes some important theoretical observations, for example, relating the coverage of the optimal policy with respect to any policy, with the sub-optimality gap of the policy. Furthermore, the proposed algorithms improve upon existing regret bounds by getting rid of instance-dependence, which is also an important contribution. These strengths contribute towards my decision to lean towards acceptance of this paper, even though the simulations can be improved.

**Theoretical Claims:**

I did not go over the appendices, but the claims in the paper look correct.

---

> ### Author Rebuttal · Authors · 2025-03-29
>
> We thank the reviewer for the positive feedback and insightful suggestions! We address your comments as follows.
> ## 1. Methods And Evaluation Criteria & Experimental Designs Or Analyses
> ### 1.1 Comparison with other online algorithms
> Regarding online RLHF methods, we first point out that other existing methods (XPO, IPO [1] etc) cannot handle transfer settings, hence they are not directly comparable. Besides, the core technique in empirical TPO is the "win rate-based source policy selection via UCB", which allow us to adapt to the best source RMs and switch back to normal online learning if no benefits in transfer, *without prior knowledge on task quality*. Therefore, **we view other online methods (e.g. XPO) not as competing baselines, but rather as complementary approaches that can be enhanced with our transfer learning techniques**.
>
> To make it clear, in our next revision, we will replace “DPO” (line 11, Alg. 3) with $Alg_{PO}$, which serves as a placeholder for any **P**olicy **O**ptimization oracle (e.g. DPO, XPO, IPO etc). This change aligns with the usage of generic placeholder $Alg_{OL}$ in TPO (Alg. 1).
>
> To support this point, similar to Table 1 in paper, below we report win rates comparisons when we instantiate $Alg_{PO}$ by optimizing XPO and IPO loss, respectively, keeping other settings the same as Sec. 6 (except using learning rate 1e-5 in IPO). These results demonstrate that our transfer learning techniques can be effectively combined with different policy learners, leading to consistent performance improvements. We believe this highlights the modularity and generality of our framework, and we thank the reviewer again for prompting this valuable point.
> * **Empirical TPO (Alg. 3) by replacing DPO (line 11) with XPO**
> ||Without Transfer|Purely Exploit ROUGE|Purely Exploit T5-Large|
> |:-: |:-:|:-:|:-:|
> |Iter 1|52.3$\pm$1.1|53.4$\pm$0.8|50.2$\pm$0.3|
> |Iter 2|51.6$\pm$1.3|54.7$\pm$1.6|49.1$\pm$1.3|
> |Iter 3|52.2$\pm$1.6|53.8$\pm$2.9|49.2$\pm$1.1|
> * **Empirical TPO (Alg. 3) by replacing DPO (line 11) with IPO**
> ||Without Transfer|Purely Exploit ROUGE|Purely Exploit T5-Large|
> |:-: |:-:|:-:|:-:|
> |Iter 1|52.3$\pm$1.0|50.4$\pm$1.6|49.9$\pm$0.4|
> |Iter 2|55.2$\pm$1.4|52.3$\pm$0.3|50.1$\pm$0.3|
> |Iter 3|55.3$\pm$1.1|51.8$\pm$0.5|50.3$\pm$0.5|
> ### 1.2 Additional benchmarks
>
> We consider summarization task for the following reasons:
> 1. Summarization task is important and it is also widely used in RLHF [2, 3].
> 2. Summarization task is quite suitable for our reward model transfer setup. There are various choices for additional reward models with different qualities, such as similarity scores (ROUGE, BERTScore) with human expert summaries, advanced LLMs, etc.
>
> We believe it is an interesting direction to evaluate our algorithms with other LLMs and benchmarks (e.g. those in [4] as suggested). Given that our experiments already effectively demonstrate the advantage of our proposed approach and due to limited computational resources, we leave further evaluation to the future works.
> ## 2. Questions For Authors
> ### 2.1 Why not estimate policy value directly
>
> The main reason is that under the Bradley-Terry assumption, the preference distribution $\mathbb{P}_{r^*}(\cdot|s,a,\tilde{a}) = \sigma(r^*(s,a)-r^*(s,\tilde{a}))$ is invariant under the transformation that $r^*(s,a)\rightarrow r^*(s,a)+b(s)$, where $b$ is an arbitrary state-dependent function. As a result, we can at best identify the true reward $r^*(s,a)$ up to a state-dependent shift, which can largely bias the policy evaluation and make it unreliable.
>
> In contrast, introducing $J_\beta(\pi_{\text{ref}})$ as the baseline can cancel out the bias term. This enables consistent estimation for the value difference $J_\beta(\pi^*_{r^w})-J_\beta(\pi_{\text{ref}})$.
> ### 2.2 Clarification on the second term in Eq. (8)
>
> Firstly, notice that the second term of Eq. (8) takes the minimum over $\sum_w1/\Delta(w)$ and $\sqrt{Wt}$. When $\Delta(w)$ is very small, $\sqrt{Wt}$ avoids the upper bound being extremely large.
>
> Secondly, as in most regret bounds in online learning, our result characterizes the worst-case behavior. Analogous to multi-armed bandit (MAB) settings, for fixed $T$, in the worst case we may have $\Delta(w)=\sqrt{T/W}$, resulting in regret that matches our bound.
>
> Moreover, there is a simple refinement can be done: additionally taking the minimum over the current bound and $\sum_w\Delta(w)N(w,T)=O(\sum_w\Delta(w) T)$, where $N(w,T)$ denotes the number of times we transfer from model $r^w$ in $T$ iterations. This may align more closely with the reviewer’s intuition. In fact, this basic regret bound was actually the starting point of deriving Thm. 4.3.
>
> [1] A General Theoretical Paradigm to Understand Learning from Human Preferences
>
> [2] Conditional Language Policy: A General Framework for Steerable Multi-Objective Finetuning.
>
> [3] BOND: Aligning LLMs with Best-of-N Distillation
>
> [4] Reinforcement learning from human feedback with active queries

---

> > ### Comment · Reviewer_yCvD · 2025-04-04
> >
> > I would like to thank the author(s) for the detailed response to my queries. My concerns have been addressed. I was optimistic in my score, and with all my queries addressed, I would like to maintain my evaluation of the paper. Good luck!

---

> > > ### Author Response · Authors · 2025-04-04
> > >
> > > Thank you for recognizing the contributions of our work and the constructive feedback!

---

### Official Review · Reviewer_kNcQ · 2025-03-13

**Overall Recommendation:** 4

**Summary:**

This paper studies the provably benefits of transferring knowledge from imperfect reward models (RMs) in online reinforcement learning from human feedback (RLHF). First, this paper identifies an important property specific to KL-regularized RLHF: the coverability for the optimal policy can be upper bounded by the policy value gap. This implies in order to obtain a dataset with good coverage, it is sufficient to rollout the policy with high value. Guided by this principle, this paper proposes a “self-transfer learning” procedure, which first runs a online no-regret method to generate a dataset, followed by employing an offline method to output the final policy. This paper proves that this procedure enjoys a sub-optimality bound of $\mathcal{T^{-1/2}}$, which improves previous results by removing the dependence of certain complexity measure in the dominating term.

Again guided by the above principle, this paper further proposes a transfer learning method TPO. TPO first runs certain iterations of online algorithm and then switches to a transfer policy selection (TPS) procedure, which selects the policy with the highest optimistic estimate of the value. This paper provides regret bound of TPO, demonstrating that 1). in the early stage, the regret of TPO is reduced by leveraging the imperfect RMs; 2). after finite iterations, the regret bound almost reduces to the one of self-transfer learning.

Finally, this paper proposes an empirical TPO method, which selects the policy based on the wining rate. The effectiveness of TPO is validated on a simple summarization task.

**Claims And Evidence:**

I feel main claims in this paper are well supported by clear evidence. I only have some minor questions.

1. In the discussion below Theorem 3.2, the paper emphasizes that the suboptimality does not depend on |S|, |A| or other complexity measure. But if I understand correctly, the suboptimality is of $\mathcal{O} (T^{-1/2} + Cov (|\Pi|)T^{-1} )$, which still depends on the complexity measure.
2. In the first paragraph in Section 2.2, the paper states that $\Delta_{\min}=0$ implies the realizability of the reward class. This is not very rigorous because a reward with a constant shift also ensures the same optimal policy. The same issues also happens in “there is a one-to-one correspondence XXX”.

**Essential References Not Discussed:**

To the best of my knowledge, this paper provides a sufficiently thorough discussion of all closely related works.

**Experimental Designs Or Analyses:**

The experimental designs are adequate. However, the empirical results have a weak connection to the theory. This is primarily because empirical TPO uses the winning rate to select policies. Unlike the policy value, the winning rate cannot provide an upper bound for the coverage coefficient—a crucial element in the theoretical framework established in this paper.

**Methods And Evaluation Criteria:**

The methodology of this paper builds on the policy coverage perspective commonly used in existing offline RLHF theory, along with a novel structure property induced by KL regularization. I find this methodological approach well-reasoned.

Besides, the experiments are conducted on a standard summarization task, which looks solid to me.

**Other Comments Or Suggestions:**

The proposed self-transfer learning process employs two existing RLHF methods, which may not be computationally efficient since it involves two separate policy optimization procedures. Could we leverage the structural property to design a new online RLHF method with a single policy optimization procedure, while achieving an improved regret bound?

**Other Strengths And Weaknesses:**

Please see my comments in the above part.

**Questions For Authors:**

Please see my questions in the above part.

**Relation To Broader Scientific Literature:**

This paper studies online RLHF for LLM alignment and may inspire advancements towards sample-efficient methods for training LLMs.

**Theoretical Claims:**

The results appear to be mathematically sound, though I have a question about Theorem 4.3, which contains a seemingly counterintuitive element. Specifically, the second term in Eq.(8) contains a $\sum_{w} 1/\Delta (w)$ quantity that increases as the error of source RMs decreases. This seems to contradict the intuition that higher-quality RMs should lead to lower regret.

---

> ### Author Rebuttal · Authors · 2025-03-29
>
> We thank the reviewer for the positive feedback and constructive suggestions! We address your specific comments in the following.
>
> ## 1. Claims And Evidence
>
> ### 1.1 About Theorem 3.2
>
> As correctly pointed out, the sub-optimality gap is $\tilde{O}(T^{-1/2} + Cov(\Pi) T^{-1})$, and that’s why we claim the suboptimality/regret does not depend on complexity measure **after finite time**. As long as $T \geq \Omega(Cov(\Pi)^2)$, we have $\tilde{O}(T^{-1/2} + Cov(\Pi) T^{-1}) = \tilde{O}(T^{-1/2})$, and $T$ becomes the only dominating term. We will make this point clear in our revision.
>
> ### 1.2 A few rigorousness Issue
>
> Thanks for pointing them out! We will revise our statements to improve their rigor as recommended.
>
>
> ## 2. Theoretical Claims
> Firstly, notice that the second term of Eq.(8) takes the minimum over $\sum_w 1/\Delta(w)$ and $\sqrt{Wt}$. When $\Delta(w)$ is very small, $\sqrt{Wt}$ avoids the upper bound being extremely large.
>
> Secondly, as in most regret bounds in online learning, our result characterizes the worst-case behavior. Analogous to multi-armed bandit (MAB) settings, for fixed $T$, in the worst case we may have $\Delta(w) = \sqrt{T/W}$, resulting in regret that matches our bound.
>
> Moreover, there is a simple refinement can be done: additionally taking the minimum over the current bound and $\sum_w \Delta(w) N(w,T) = O(\sum_w \Delta(w) T)$, where $N(w,T)$ denotes the number of times we transfer from model $\pi^*_{r^w}$ in $T$ iterations. This may align more closely with the reviewer’s intuition. In fact, this basic regret bound was the starting point of deriving Theorem 4.3.
>
> ## 3. Experimental Designs Or Analyses
>
> Thanks for raising this valid point. Despite difference in design, both TPO and its empirical version are grounded in the same core theoretical insight—**transfer from policy with better coverage for $\pi^{\*}\_{r^{\*}}$ (i.e. low $Cov^{\pi^{\*}\_{r^{\*}}|.}$)**. The value estimation is preferable from a theoretical standpoint, but it is often computationally expensive in practice. To address this, our empirical TPO uses win rate as a proxy, because it is more scalable and characterizes the lower bound for $Cov^{\pi^{\*}\_{r^{\*}}|.}$ (Lem. 5.1). We view this as a reasonable trade-off between theoretical rigor and empirical scalability.
>
>
> ## 4. Other Comments Or Suggestions
> > Could we leverage...while achieving an improved regret bound?
>
> This is a very interesting question. We conjecture it is possible to design more elegant algorithms with better regret bound, and we believe it is a highly valuable direction. We hope our work can serve as an initial step and inspire future developments, and we leave this for future work.

---

### Official Review · Reviewer_UGku · 2025-03-25

**Overall Recommendation:** 1

**Summary:**

The paper proposes a transfer learning algorithm that utilizes offline and online preference-based policy learning methods for RLHF. They provide a policy selection algorithm in each step where a new policy is selected based on a set of imperfect reward models and is used to further augment the training dataset. They provide theoretical evidence, a regret bound on the learned policy, for their proposed method and an emprical computationally efficient algorithm as a practical alternative for their method.
The motivation and flow of the paper is well-written and the principles explained about the concept of coverage and its use in the construction of the algorithm gives very clear picture of the whole idea of the paper.
They provide a regret bound that is independent from size of the state and action space and the complexity of the policy space.

**Claims And Evidence:**

The main claims of the paper include,
1. The design of an algorithm for RLHF (TPO) with proved regret bound.
2. Proposition of the computationally efficient version of TPO
3. The importance of policy value as a criterion for selecting the policy to generate training data.
4. The proposition of a policy learned by offline data, that is proved to have a regret bound of $O(1/\sqrt{T})$ with no dependence on the action and state spaces size or any complexity measures on the policy space.

The first three claims are well justified, but the fourth claim, which I assume is the main contribution of the paper, is not correctly justified. Theorem 3.2 is proposed to provide a regret bound on the offline policy and is claimed to be independent of **any complexity measure on policy class**. But actually, when looking at the complete term of the bound in the Appendix, it **depends** on $|\Pi|$, which violates the authors' claim.

Another problem with this claim that affects the whole idea of the paper is that we use the term **Offline** when we don't have access to the environment anymore and have to teach the model with the prepared, fixed training dataset. This is one of the core applications of offline methods that work even when we can't interact with the environment anymore. Here, to train the offline model, we require a no-regret online method to generate the dataset, so we have to access the environment continuously for the value difference term in the bound to vanish, hence the so-called offline policy is no longer offline, it requires access to the environment to generate a continuously improving dataset to learn from.

Altogether, the contribution of the paper compared to many algorithms in this field like RPO and XPO is not clear. The method requires access to the environment (so it's not offline) and it uses the same data sample many times during the training (so it's not online), and also the proved bound doesn't seem to have any advantage over RPO or XPO in terms of T, or other factors.

**Essential References Not Discussed:**

I didn't recognize any missed essential references.

**Experimental Designs Or Analyses:**

Experiments are not complete. The complete algorithm that automatically selects the best source model is missed in the main paper results, only fixed source selection methods are reported and tested. Also the only compared baseline is iterative-DPO, and algorithms like RPO, XPO, IPO, and SimPO are missed as baseline.
The evaluation criterion is not justifiable, because it is only the win rate of different policies compared to each other.
The experiments section seems more like a small ablation study section rather than the main experiment section. Even the appendix doesn't cover the lack of experiments and insufficient experimental evidence. It should at least contain the absolute performance of the complete method on difference preference datasets and comparison with online and offline RLHF methods like RPO and XPO.

**Methods And Evaluation Criteria:**

The theoretical evaluation is the conventional regret bound, which is standard and reasonable.
The experimental evaluation is not very convincing, as it is the win rate of the policies compared to each other. An absolute measure of performance, like accuracy in the preference dataset, could be a better metric, as it can be compared with any baseline or SOTA method.

**Other Comments Or Suggestions:**

I have no other comments.

**Other Strengths And Weaknesses:**

The whole paper fails to convince its contribution to the big literature in RLHF,
Moreover, the provided empirical algorithm, differs significantly from the main algorithm, in the very important part of the source selection. This will make the validity of the theoretical evidence questionable for the empirical algorithm. It is common to have an empirical algorithm that is different from the theoretically justified method because of some estimations required in practice, but in this work, the difference is not just because of an estimation, the source selection algorithm is totally changed.

**Questions For Authors:**

I don't have any questions.

**Relation To Broader Scientific Literature:**

The paper's main goal is to solve the RLHF task by generating a good training dataset and train a model on the generated dataset. It is linked to both offline and online policy learning methods that use human-annotated preference data. However the focus of the paper is not generating data from human feedback, but from a set of already available reward models.

**Theoretical Claims:**

As mentioned in the claims section, theorem 3.2 provides a regret bound on the offline policy that is claimed to be independent of policy class complexity, but it's not.
There is another serious problem with the theoretical claims, and it is the removal of the best policy coverage term for the offline policy. For offline learning, coverage is necessary because we don't have control over the offline dataset. If we can control the offline dataset, we can trivially generate customized datasets that don't have a high coverage issue. Removal of the coverage term in the offline learning literature is not a contribution, as the coverage term makes the bound instance dependent and tighter, based on the quality of the dataset and algorithms that don't have the coverage term, often have looser bound, because the bound should work on **any dataset**.

Also, the assumption of a bounded policy ratio is very restrictive and not realistic in most of the applications.

---

> ### Author Rebuttal · Authors · 2025-03-29
>
> We thank the reviewer for the feedback. It seems there may be some misunderstandings regarding our setting and several our key claims. To clarify, we start with a general remark, followed by detailed point-to-point responses. We hope our replies improve the clarity of our submission and help the reviewer evaluate our paper.
> ## General Remark for Clarification
> As stated in Sec. 1&2 and Fig. 1, **we target at improving sample efficiency in online RLHF by transferring from imperfect source RMs**, i.e. learning $\pi^\*_{r^\*}$  from online human feedbacks associated with $r^\*$, while leveraging auxiliary RMs $r^1, \dots, r^W$.
> * Our theoretical method TPO (Alg 1&2) **is not offline but instead an online reward transfer algorithm**, and we just use an offline subroutine (RPO) to compute a transfer candidate (Lines 6–7 in Alg. 2). Besides, **we did not claim Thm. 3.2 is a contribution for offline literature**. Instead, we claim it improves the existing convergence rate of online RLHF methods, which motivates "self-transfer learning". **Our core theoretical contribution lies in analyzing the benefits of transfer learning in online RLHF**. As stated in Thm. 4.3 and Sec. 4.2, TPO improves the online RLHF results given good source RMs and self-transfer learning.
> * Empirical TPO (Alg. 3) is closely aligned with the theoretical insights behind TPO—**transfer from policy with better coverage for $\pi^{\*}\_{r{\*}}$ (i.e., low $Cov^{\pi^{\*}\_{r^{\*}}|.}$)**. TPO follows it by selecting policy with its value gap (upper bound of $Cov^{\pi^*_{r^*}|.}$ by Lem. 3.1), while empirical TPO utilizes win rates (lower bound of $Cov^{\pi^*_{r^*}|\cdot}$ by Lem. 5.1, but more scalable).
> ## 1. Claims and Evidence
> >Theorem 3.2...violates the authors' claim.
>
> In Thm 3.2, we only omit $\log|\Pi|$—the log covering number. We apologize that our wording is not precise enough, and will revise to "no dependence…up to log-covering number factors". However, such log terms are standard even in supervised learning, and removing any policy coverage terms in previous online RLHF bounds is a significant improvement.
>
> We also clarify that Thm. 3.2 is about convergence rate, not regret bound, and it is only part of the contributions.
> >The method...not offline…not online…
>
> We apologize for the confusion. We study reward transfer in **online setting**, and the term "offline" appears in paper just because we use an offline method (RPO) to compute a policy. We will reword "offline" to "distilled" to avoid confusion.
> >the contribution...to…RPO and XPO is not clear
>
> We reply to it together in point 2 below.
> ## 2. Methods And Evaluation Criteria & Experimental Designs Or Analyses
>
> Given that we study the online setting, offline methods (e.g. RPO) operate under different assumptions and objectives and are naturally not comparable.
>
> Regarding online RLHF methods, note that other existing methods (XPO, IPO, etc) cannot handle transfer settings, hence they are not directly comparable. Besides, the core technique in empirical TPO is the "win rate-based source policy selection via UCB", which adapts to the best source RMs and switches back to normal online learning if no benefits in transfer, *without prior knowledge on task quality*. Therefore, **we view other online methods (e.g. XPO, IPO) not as competing baselines, but rather as complementary approaches that can be enhanced with our transfer learning techniques**. For limited space, we refer to "1.1 Comparison with other online algorithms" in our response to Reviewer yCvD for additional discussion and experimental support.
>
> Our main goal in experiments is to show the effectiveness of empirical TPO, and our experiment results clearly demonstrate these. Besides, win rate is a standard metric in evaluating RLHF methods. We note that accuracy is not typically used as a standard metric for summarization tasks.
> ## 3. Theoretical Claims
> >​​Removal of...in the offline learning literature is not a contribution…
>
> There is a misinterpretation of our contribution. We study the online setting and our main contribution is to **improve sample efficiency in online RLHF by transfer learning**.
> >...bounded policy ratio is very restrictive…
>
> Such an assumption is standard in online RLHF literature [1,2]. Besides, as stated in Footnote 1 (Page 3), it is not essentially an assumption, but an additional preprocessing step given a realizable policy class, because $r^* \in [0, R]$ implies $||\log \pi^*_{r^*}/\pi_{ref}||_\infty \leq R/\beta$.
>
> ## 4. Other Strengths And Weaknesses
> >...empirical algorithm, differs significantly from the main algorithm…the source selection algorithm is totally changed
>
> We respectfully disagree with it. Both our TPO and empirical TPO share the same insight (see second bullet in general remark).
>
> [1] Exploratory preference optimization: Harnessing implicit q*-approximation for sample-efficient rlhf
>
> [2] Self-Exploring Language Models: Active Preference Elicitation for Online Alignment

---

> > ### Comment · Reviewer_UGku · 2025-04-07
> >
> > There may be a misunderstanding about the contribution of the paper. I understand that the method is a transfer learning approach to utilize a set of imperfect reward models. Now let's compare the proposed method with an existing online method like XPO. Both methods are trying to solve the RLHF task, with access to the environment (online). I understand that the approaches are different, and the proposed method uses transfer learning, yet both are finally solving the same task, as an end-to-end system.
> >
> > Now, we have the theoretical and practical contributions. I may be wrong and have misunderstood the contributions, so I ask the authors to clarify.
> >
> > Theoretical contribution:
> >
> > XPO's bound depends on $\sqrt{Cov_\infty(\Pi)}$, but TPO (in infinity) depends on $\log(|\Pi|)$. Is there any theorem to compare the two values and state that the first term is significantly larger (in order or in practice)?. Are there any other theoretical contributions compared with XPO? I seek a precise mathematical statement that theoretically, the proposed method beats already available RLHF methods, e.g., XPO.
> >
> > Practical contribution:
> >
> > Do we have any experiment that computes the win-rate of a TPO-trained policy over an XPO-trained policy?

---

> > > ### Author Response · Authors · 2025-04-07
> > >
> > > We sincerely thank the reviewer’s further questions and appreciate the chance to clarify it!
> > >
> > > ## 1. Theoretical Contribution
> > >
> > > Briefly speaking, both XPO and our results depend on $\log|\Pi|$, while our results are **strictly better in that we eliminate the coverage term $Cov_\infty(\Pi)$** (after finite time). For clarity, in the following big-O notations, we only omit constant terms and $\log T$.
> > >
> > > ### 1.1 Comparison in Regret Bounds
> > > * **XPO**: As stated in Thm. 3.1 and its proof in [1], w.p. $1-\delta$, running XPO for $T$ steps yields regret bound:
> > > $
> > > \tilde{O}(\sqrt{Cov_\infty(\Pi) T \log\frac{|\Pi|}{\delta}}).
> > > $
> > >
> > > * **TPO**: As discussed in our Sec. 4.2, w.p. $1-\delta$, with appropriate choice of $\alpha$ (e.g. $\alpha = e^{-\frac{R}{\beta}}$), the regret of TPO is
> > >
> > >   * $\tilde{O}(W\sqrt{T \log\frac{|\Pi|}{\delta}})$ when $T\leq \frac{W^2}{\Delta_{\min}^2}$; improving from $\sqrt{Cov_\infty(\Pi)}$ to $W$—the number of source tasks, which is usually small.
> > >   * $\tilde{O}(\sqrt{T  \log\frac{|\Pi|}{\delta}})$ when $T > \frac{W^2}{\Delta_{\min}^2}$ and large enough; improving from $\sqrt{Cov_\infty(\Pi)}$ to $O(1)$.
> > >
> > >   The improvements in the above two stages are contributed by existence of good source tasks and self-transfer learning, respectively.
> > >
> > > ### 1.1 Comparison in Convergence Rates
> > > * **XPO**: [1] reports a convergence rate by outputting the uniform mixture policy
> > > $$
> > > 	\tilde{O}(\sqrt{\frac{Cov_\infty(\Pi)} {T} \log\frac{|\Pi|}{\delta}}),
> > > $$
> > >
> > > * **TPO**: Similarly, the regret bound of TPO implies the following convergence rate (after finite time):
> > > $$
> > > 	\tilde{O}(\sqrt{\frac1T\log\frac{|\Pi|}{\delta}}).
> > > $$
> > >
> > > ## 2. Practical Contribution
> > >
> > > As we mentioned in our rebuttal, **we view other online methods (e.g. XPO) not as competing baselines, but rather as complementary approaches that can be enhanced with our transfer learning techniques**. We decided to replace “DPO” (line 11, Alg. 3) with $Alg_{PO}$, which serves as a placeholder for any Policy Optimization oracle (e.g. DPO, XPO, IPO etc).
> > >
> > > To support this claim, in our response to Reviewer yCvD, we consider instantiating $Alg_{PO}$ by optimizing XPO or IPO loss, and report the win rates (similar to Table 1 in paper) between the policies produced by TPO and other baselines. For convenience, we re-report those results below. All experimental settings remain the same as in Sec. 6, except for using a smaller learning rate of 1e-5 in the IPO experiments.
> > >
> > > **Empirical TPO (Alg. 3) by replacing DPO (line 11) with XPO**
> > >
> > > ||Without Transfer|Purely Exploit ROUGE|Purely Exploit T5-Large|
> > > |:-:|:-:|:-:|:-:|
> > > |Iter 1|52.3$\pm$1.1|53.4$\pm$0.8|50.2$\pm$0.3|
> > > |Iter 2|51.6$\pm$1.3|54.7$\pm$1.6|49.1$\pm$1.3|
> > > |Iter 3|52.2$\pm$1.6|53.8$\pm$2.9|49.2$\pm$1.1|
> > >
> > >
> > > Here the “without transfer” baseline (i.e., when $W = 0$) **is exactly the empirical XPO** in [1] (see “Implementation details” in Appx. E in [1]). The advantage of win rates demonstrates that our transfer learning techniques can further enhance performance of XPO when good source tasks exist. This highlights not only the effectiveness but also the modularity of our approach.
> > >
> > > **Empirical TPO (Alg. 3) by replacing DPO (line 11) with IPO**
> > >
> > > ||Without Transfer|Purely Exploit ROUGE|Purely Exploit T5-Large|
> > > |:-:|:-:|:-:|:-:|
> > > |Iter 1|52.3$\pm$1.0|50.4$\pm$1.6|49.9$\pm$0.4|
> > > |Iter 2|55.2$\pm$1.4|52.3$\pm$0.3|50.1$\pm$0.3|
> > > |Iter 3|55.3$\pm$1.1|51.8$\pm$0.5|50.3$\pm$0.5|
> > >
> > >
> > > [1] Exploratory Preference Optimization: Harnessing Implicit Q*-Approximation for Sample-Efficient

---

### Official Review · Reviewer_WBNK · 2025-03-27

**Overall Recommendation:** 4

**Summary:**

This paper studies RLHF under the contextual bandit setting with KL
regularization. In usual bandit problems, there is a need to balance exploration
and exploitation. However, they show that there is a "blessing of
regularization" in which the a policy that has low policy value gap will also be
a good exploration policy. In particular, this means that due to the presence of
regularization, the two goals of exploration and exploitation are aligned, and
do not require additional assumptions that are usually made about the problem to
avoid negative transfer. Using this idea, the authors design the Transfer Policy
Optimization (TPO) algorithm for this problem, prove a offline policy value gap,
prove a regret bound, propose an alternative algorithm that is more
computationally efficient, and perform some empirical validation.

**Claims And Evidence:**

1. While I understand the claim that Theorem 3.2 shows that the offline policy
   converges at an asymptotic rate that does not depend on any structural
   complexity measure (that you are treating everything except $T$ as a
   constant), I think it may be clearer to say more explicitly that there are
   two phases to the convergence rate. In the first phase, convergence rate
   decreases at a rate of $O(e^{1/\beta}\mathcal{C}(\Pi)(\beta T)^{-1})$, while
   in the second, it decreases at a rate of $O(T^{-1/2})$. In particular, if
   this complexity measure $\mathcal{C}(\Pi)$ used in Corollary E.6 is infinite,
   then the second phase that is independent of $\mathcal{C}(\Pi)$ never happens
   (or, if it is allowed to grow with $T$).

   Preserving $\beta$ and $\mathcal{C}(\Pi)$ in the bound makes it clear that
   they do matter in convergence, but only in the first phase. This is a pretty
   interesting phenomenon, and maybe it would be nice to expand further on this
   discussion. For example, do you have any intuition/justification for why the
   second phase is also independent of $\beta$?

2. I thought Section 4 was a bit confusing to read. I think the last two
   paragraphs of Section 3 are quite important to understand Section 4. But
   somehow, it is not obvious until one understands the TPO algorithm why they
   are important. Is there a way to write Sections 3 and 4 so that it is easier
   to grasp what the algorithm? Part of the problem is that the way that Theorem
   3.2 relates to the whole paper and TPO is also not initially clear. I like
   the line near the end of Section 5 "learn from an expert until surpassing
   it". Perhaps that idea could be more explicit earlier in the paper and could
   be a guiding thought figure throughout the paper.

**Essential References Not Discussed:**

N/A

**Experimental Designs Or Analyses:**

As I mentioned before, the experiments were somewhat confusing. It would be nice
if there were more justification for why these experiments were performed.

**Methods And Evaluation Criteria:**

1. The experiments were a bit confusing to me. A lot of details were missing
   from the main paper and are in the appendix. It is also not clear to me
   whether Table 1 is a 'good' empirical result. These numbers do not really
   seem to say that the proposed empirical method beats the baselines
   (everything seem quite close to 50% win rate). Or, is the point that only
   three iterations were needed? Would the trend in improvement continue with
   additional iterations?

**Other Comments Or Suggestions:**

The review above is written by me. Below, I asked chatgpt to help summarize it. I think it captured my overall impression quite well:

```[ChatGPT summary of my review]: This paper presents a theoretically motivated and timely contribution to RLHF by showing that KL regularization aligns exploration and exploitation, enabling safe and effective transfer from both external and self-generated policies. I found the core idea of "self-transfer" particularly compelling, and the theoretical results are thoughtfully developed. However, I believe the presentation could be improved—the connection between Sections 3 and 4 is not immediately clear, and the empirical results, while suggestive, are limited in scope and detail. Additionally, I think the claim of complexity-independent convergence could be more carefully nuanced to reflect the two-phase nature of the bound. Despite these issues, the paper makes a valuable contribution to understanding transfer in regularized bandit-style RLHF, and I recommend it for acceptance with revisions focused on clarity and empirical depth.```

Finally, though I do believe I understood the paper, I don't work directly in RL or RLHF, so my confidence score is somewhat low because I am not very familiar about how this paper is situated within the field.

**Other Strengths And Weaknesses:**

1. In Section 2, Additional Notation, you write that "$\mathcal{R}^\Pi$ denotes
   the reward class converted from $\Pi$." I don’t believe "converted" is
   standard terminology in RL. If what’s meant is the reward class induced by
   $\Pi$ via the one-to-one mapping in Eq. (2), it may be helpful to say that
   directly to avoid confusion.

2. I think overall the writing is not bad, but the overall structure could do
   with a bit more thought. It was not trivial to understand the structure of
   the paper and required very non-linear reading. Perhaps this is due to my
   lack of familiarity with the area, however.

**Questions For Authors:**

N/A

**Relation To Broader Scientific Literature:**

I found the high-level ideas of this paper quite interesting, especially the
idea of self-transfer. I almost wish there more of a focus on this and transfer
for KL-regularized bandits, and less RLHF. In other words, are the same ideas be
present even in the very basic bandit setting? In terms of exposition, I think I
would've really appreciated learning the basic ideas in a much simpler setting.
Although, I understand that it is also good to connect to the current RLHF/LLM
research interests across the community.

**Theoretical Claims:**

I did not have time to check proofs and for the most part only read the main
paper.

---

> ### Author Rebuttal · Authors · 2025-03-29
>
> We thank the reviewer for the positive feedback and constructive suggestions! We address your specific comments in the following.
>
> ## 1. Claims And Evidence
>
> ### 1.1 About Claim in Thm. 3.2
>
> Thank you for the suggestion. We will follow it and clarify our claim in our next revision.
>
> > ...why the second phase is also independent of $\beta$?
>
> As predicted by the offline learning theory, the offline policy converges to $\pi^*_{r^*}$ at a rate of $\tilde{O}(Cov^{\pi^*_{r^*}|\pi_{mix}^T} T^{-1/2})$, where $\pi_{mix}^T := \frac{1}{T}\sum_{t=1}^T \pi^t$ is the uniform mixture of the policies collecting data. Technically, in the second phase, $\pi_{mix}^T$ is very close to $\pi^*_{r^*}$, and as implied by Lem. 3.1, $Cov^{\pi^*_{r^*}|\pi_{mix}^T}$ is at constant level, which results in the $\tilde{O}(T^{-1/2})$ rate. From this view, $\beta$ only matters in how fast $Cov^{\pi^*_{r^*}|\pi_{mix}^T}$ reduces to constant as $T$ grows.
>
>
> ### 1.2 About structure of Section 3 and 4
>
> Thank you for pointing it out, and we will try our best to make it more understandable. The core insight behind our discussion in Sec. 3 and 4 (and also empirical TPO in Sec. 5) is that **"select and transfer from the policy with the best coverage of $\pi^{\*}_{r^{\*}}$"**. It may help the reader to grasp the main algorithm if we highlight this statement earlier (e.g. Sec. 3.2).
>
> ## 2. Methods And Evaluation Criteria & Experimental Designs Or Analyses
>
> > It is also not clear to me whether Table 1 is a 'good' empirical result…
>
> Win rate is a common metric in evaluating the LLM performance. A "(50+$p$)%" win rate against no-transfer baseline roughly means that, per 100 prompts, the LLM fine-tuned by empirical TPO is preferable on $50+p$ out of 100, which is $2p$ more than no-transfer baseline ($50 - p$ out of 100).
>
> Besides, the performance gap may be further enlarged by increasing training batch size and training epochs, incorporating better source reward models, adjusting test-time decoding temperature, etc. Notably, the quality of the inital policy also affects the magtitude of the performance gap---**the more near-optimal the initial policy is, the smaller the maximal possible performance gap can be**, although empirical TPO can still outperform the no-transfer baseline.
>
> Therefore, our primary goal in experiments is to demonstrate the potential of our transfer learning techniques in improving sample efficiency, especially, reward transfer remains an underexplored area in online RLHF.
>
> > ...is the point that only three iterations were needed?
>
> Here we follow the related literature [1,2], where running for three iterations is a common choice in algorithm evaluation (due to limited computational resources).
>
> > Would the trend in improvement continue with additional iterations?
>
> As reported in Fig. 2 in Appx. I, in the 3rd iteration, empirical TPO already switches back to online learning without transfer, as the online policy can already outperform the best source policy in win rates. If we continue training for more iterations, the comparison with the no-transfer baseline effectively reduces to running the same online algorithm but with a better initial policy. Overall, we can expect the advantage of empirical TPO to persist, although the performance gap may decrease as both of them are converging to the optimal policy.
>
> ## 3. Relation To Broader Scientific Literature
>
> To our knowledge, most existing transfer learning literature, especially those theoretical ones, focuses on pure-reward maximization setup. We are the first to identify special structure (Lem. 3.1) induced by KL regularization, and propose novel transfer learning principles specialized in this setting.
>
> ## 4. Other Strengths And Weaknesses
> Thank you for your suggestions! We will take them into consideration in our next revision.
>
> [1] Iterative Preference Learning from Human Feedback: Bridging Theory and Practice for RLHF under KL-constraint
>
> [2] Exploratory Preference Optimization: Harnessing Implicit Q*-Approximation for Sample-Efficient RLHF

---

### Decision · Program_Chairs · 2025-05-01

**Decision:**

Accept (poster)

**Comment:**

The reviewers appreciated the paper revealing the insight that due to reguarization, small policy value gap implies policy covering the optimal \pi_{r^*}^*, which can be vieweed as a way to perform self-transfer: using previously-trained policy to collect data can yield good coverage. They also appreciate the algorithmic insight that in the presence of imperfect reward models, these models can be used until the learned policy surpasses it. They also pointed out some weaknesses of the paper:

- the theoretical algorithm and the empirical one were somewhat disconnected, in the difference in the way the data collection policy is chosen

- comparison of the theroetical guarantee of TPO and baselines can be clarified more; the bound in Theorem 4.3 looks fairly complicated; does it provide guidance on how to choose \alpha? The paper had some discussing on choosing \alpha as a sequence, but it is not clear how to choose it apriori if the quality of the reward models were unknown ahead of time.

Due to these considerations this paper is on the borderline.